# Evaluating the Arabian Sea as a regional source of atmospheric $CO_2$: seasonal variability and drivers

Alain de Verneil[1], Zouhair Lachkar[1], Shafer Smith[2], and Marina Lévy[3]

[1]Center for Protoype Climate Modeling, New York University Abu Dhabi, Abu Dhabi, UAE
[2]Courant Institute of Mathematical Sciences, New York University, New York, USA
[3]Sorbonne Université (CNRS/IRD/MNHN), LOCEAN-IPSL, Paris, France

**Correspondence:** Alain de Verneil, (ajd11@nyu.edu)

**Abstract.** The Arabian Sea (AS) was confirmed to be a net emitter of $CO_2$ to the atmosphere during the international Joint Global Ocean Flux Study program of the 1990s, but since then little *in situ* data has been collected, leaving data-based methods to calculate air-sea exchange with fewer data and potentially out-of-date. Additionally, coarse-resolution models under-estimate $CO_2$ flux compared to other approaches. To address these shortcomings, we employ a high-resolution ($1/24^o$) regional model to quantify the seasonal cycle of air-sea $CO_2$ exchange in the AS by focusing on two main contributing factors, $pCO_2$ and winds. We compare the model to available *in situ* $pCO_2$ data and find that uncertainties in dissolved inorganic carbon (DIC) and total alkalinity (TA) lead to the greatest discrepancies. Nevertheless, the model is more successful than neural network approaches in replicating the large variability in summertime $pCO_2$ because it captures the AS's intense monsoon dynamics. In the seasonal $pCO_2$ cycle, temperature plays the major role in determining surface $pCO_2$, except where DIC delivery is important in summer upwelling areas. Since seasonal temperature forcing is relatively uniform, $pCO_2$ differences between the AS's sub-regions are mostly caused by geographic DIC gradients. We find that primary productivity during both summer and winter monsoon blooms, but also generally, is insufficient to off-set the physical delivery of DIC to the surface, resulting in limited biological control of $CO_2$ release. The most intense air-sea $CO_2$ exchange occurs during the summer monsoon where outgassing rates reach $\sim$**6** $molCm^{-2}yr^{-1}$ in the upwelling regions of Oman and Somalia, but the entire AS contributes $CO_2$ to the atmosphere. Despite a regional spring maximum of $pCO_2$ driven by surface heating, $CO_2$ exchange rates peak in summer due to winds, which account for $\sim$**90%** of the summer $CO_2$ flux variability versus **6%** for $pCO_2$. In comparison with other estimates, we find that the AS emits $\sim$**160**$TgCyr^{-1}$, slightly higher than previously reported. Altogether, there is **2x** variability in annual flux magnitude across methodologies considered. Future attempts to reduce the variability in estimates will likely require more *in situ* carbon data. Since summer monsoon winds are critical in determining flux both directly and indirectly through temperature, DIC, TA, mixing, and primary production effects on $pCO_2$, studies looking to predict $CO_2$ emissions in the AS with ongoing climate change will need to correctly resolve their timing, strength, and upwelling dynamics.

## 1 Introduction

The global ocean represents a major reservoir of inorganic carbon on the planet's surface (40x atmosphere), and up to the present has on average acted to uptake $\sim$23% of the 11Gt excess anthropogenic carbon (Friedlingstein et al., 2020; Ciais et al.,

2013; Khatiwala et al., 2009). The Arabian Sea (AS) is a region of the ocean that has been found to naturally release $CO_2$ to the atmosphere ($\sim 90 \mathrm{MtCyr}^{-1}$ Sarma et al., 1998), mitigating the ocean's role in moderating atmospheric $CO_2$ accumulation. While the AS as a regional basin is considered too small to greatly impact global budgets of air-sea $CO_2$ exchange (Naqvi et al., 2005), it attracts attention because high rates of air-sea $CO_2$ flux 7-33 $\mathrm{molCm}^{-2}\mathrm{yr}^{-1}$ and values >700 µatm of partial pressure of $CO_2$, or $pCO_2$, have been observed there, in addition to unique features such as the world's thickest oxygen minimum zone

(OMZ) (Morrison et al., 1999; Acharya and Panigrahi, 2016; Lachkar et al., 2016) and corresponding Carbon Maximum Zone (CMZ) (Paulmier et al., 2011).

The role of the AS as a region of net $CO_2$ emission, while suspected for decades (Keeling, 1968; Naqvi et al., 1993), was more firmly established with observations conducted under the international collaborative efforts of the Joint Global Ocean Flux Study (JGOFS) program during the 1990s (Sarma et al., 1998; Millero et al., 1998a; Goyet et al., 1998b; Naqvi et al.,

2005); see Smith (2005) and the accompanying Progress in Oceanography special issue for greater context. Conducted over several years, a major focus was to sample over the particularly strong seasonal monsoon cycle present in the AS, complete with surface current reversals, coastal upwelling, and intense phytoplankton blooms (Schott and McCreary Jr, 2001; Kumar et al., 2001; Lévy et al., 2007). JGOFS carbon data were first used to create linear statistical models, which were then extrapolated over a greater region of the AS to produce larger-scale estimates of seasonal $CO_2$ flux showing emission to the atmosphere

(Sabine et al., 2000; Sarma, 2003; Bates et al., 2006). JGOFS data still represent the greatest source of data for current de facto standard global products, such as Takahashi et al. (2009) (hereafter TK09), who produced a global climatology of $pCO_2$ and $CO_2$ flux gridded onto a $4^o$ x $5^o$ grid using a horizontal advection-diffusion scheme. In recent years, neural networks have been applied instead of simpler statistical models to likewise produce global climatologies, such as Landschützer et al. (2015) (hereafter L15) on an increased-resolution $1^o$ x $1^o$ grid. All these different methodologies, although of differing sophistication,

still rely on the availability of *in situ* data.

The wealth of information provided by the JGOFS expeditions has been invaluable for understanding the AS, but there has been little subsequent *in situ* sampling in the region, as has been previously remarked (Hood et al., 2016). For example, in the Global Ocean Data Analysis Project v2 (GLODAP; Olsen et al., 2019) database, there are no reported observations in the AS of two important carbon variables, dissolved inorganic carbon (DIC) and total alkalinity (TA), more recent than 1998, with

a similar >98% of data predating 2000 for $pCO_2$. Thus, the global products of TK09 and L15 are based upon conditions in the AS from 20 years ago. Since quantities like surface $pCO_2$ concurrently trend with rising atmospheric $CO_2$ concentration (Tjiputra et al., 2014), the dearth of recent sampling means that uncertainty in the AS's carbon system will only grow with time. The gap in data collection also means that the AS is proportionally under-represented in global datasets: whereas the AS is 2% of the ocean surface, DIC and TA measurements in the AS are <1% of the GLODAP ensemble, which is also the case

with $pCO_2$ reported in the Surface Ocean Carbon ATlas (SOCAT; Bakker et al., 2016; Pfeil et al., 2013).

Where data are sparse in the AS, numerical circulation models have been used to complement the lack of spatiotemporal coverage. These models fill the domain with their own estimates of carbon variables, such as $pCO_2$, while also providing detailed information on the factors affecting them, e.g. DIC, temperature, biological productivity, etc. For example, in the wake of the JGOFS expeditions, the synthesis study of Sarma et al. (2003) used a numerical model to examine biological and

chemical aspects of the annual carbon budget in the central and eastern AS. Further studies focus on other aspects over different timescales, such as intraseasonal $pCO_2$ variability due to temperature versus DIC (Valsala and Murtugudde, 2015), or decadal trends in pH (Sreeush et al., 2019a). These approaches, without more *in situ* data, are the best estimates we have of the current AS carbon system's behavior. Therefore, it is incumbent that these models are vigorously validated against what precious few data exist. The need to reduce uncertainty is further emphasized when modeled carbon chemistry quantities are utilized as a proxy for other things. For example, a recent modeling study in the AS found that $pCO_2$ could be used to indicate community compensation depth, which reflects the complicated balance between primary production and respiration in the water column (Sreeush et al., 2019b). As a result, the possibility exists to propagate uncertainties beyond carbon chemistry. However, these AS modeling studies compare output to established climatologies, such as TK09, which are coarse in spatial resolution and smooth out unique features of the AS such as coastal upwelling, although some studies have begun using ARGO float profiles for model validation (Chakraborty et al., 2018).

Despite the wealth of information that models provide, they have their own weaknesses. In a review of $CO_2$ flux estimates from various independent methodologies, Sarma et al. (2013) found that coupled ocean biogeochemical models underestimated the air-sea $CO_2$ flux in the AS. The underestimate was attributed to poor resolution of monsoonal currents, specifically near the coasts of Oman and Somalia. The need for sufficient resolution of monsoon and upwelling currents is underscored by the roles that small-scale horizontal (Mahadevan et al., 2004) and vertical (Mahadevan et al., 2011; Resplandy et al., 2019) currents can play in advecting carbon. Additionally, Sarma et al. (2013) found that the peak of air-sea $CO_2$ flux observed in boreal summer occurred slightly out of phase, with models leading observations by over a month in the AS. Finally, the modeled $pCO_2$ in the AS found a springtime maximum not seen in the observations based on the data from TK09. Clearly, an effort must be made to establish whether these discrepancies are residual effects of low resolution, endemic to models generally, or indicative of a real pattern that suggests future concerted *in situ* sampling.

Considering the challenges specific to studying the AS carbon cycle, in this paper we aim to put into context the role of the AS as a $CO_2$ source by quantifying air-sea $CO_2$ flux with a targeted approach. First, by employing a higher-resolution regional numerical model of the AS carbon system, monsoonal and upwelling currents will be sufficiently resolved. Furthermore, model validation will use raw data, not a smoothed climatological product, to evaluate the model air-sea $CO_2$ flux. Quantification of seasonal air-sea $CO_2$ flux will focus on the contributing factors of $\Delta pCO_2$, the difference in seawater and atmospheric $pCO_2$, and wind. In particular, the role of sea surface temperature (SST), sea surface salinity (SSS), DIC, and TA in determining the seasonal cycle of $pCO_2$ will be investigated for the entire domain of the AS as well as its spatial heterogeneity within the AS. A further budget analysis of surface DIC compares the physical and biological mechanisms governing carbon sources and sinks, such as advection and mixing versus biological production and respiration, among others. The relative impact of $pCO_2$ and winds upon the seasonal cycle of $CO_2$ flux are also compared, culminating in a meta-analysis of the model's $CO_2$ flux estimates relative to alternative approaches.

For this study, we choose to focus on the seasonal cycle due to the strength of the monsoon in the AS and because it is resolved by the *in situ* data, although models suggest interannual (Valsala and Maksyutov, 2013; Valsala et al., 2020) and intraseasonal (Valsala and Murtugudde, 2015) variability exists. The study begins with a description of $pCO_2$ datasets used,

along with the model configuration and methods of analysis in Section 2. Following this in Section 3 is a description of the model validation and results, with discussion in Section 4. We conclude in Section 5 with perspectives and recommendations regarding future studies of pCO$_2$ and air-sea CO$_2$ flux in the AS.

## 2 Methods

### 2.1 pCO$_2$ data

In this study, sea surface pCO$_2$ is used as the primary *in situ* data for model validation. Whereas models favor DIC and TA (Wolf-Gladrow et al., 2007), shipboard pCO$_2$ can be measured underway and hence there are more observations available. Additionally, since model pCO$_2$ is calculated from DIC and TA (see Sect. 2.2), pCO$_2$ measurements act as an independent dataset. Here, pCO$_2$ validation stems from *in situ* un-gridded data merged from SOCAT v. 2019 (downloaded from https://www.socat.info/index.php/version-2019/ September 2019) and the Lamont-Doherty Earth Observatory (LDEO) surface pCO$_2$
database (Takahashi et al., 2019). Both databases aggregate all available *in situ* surface pCO$_2$ data, including JGOFS. SOCAT and LDEO contain >180,000 and ∼90,000 data in the AS, respectively. SOCAT has more data because it includes multiple methodologies. As a result, SOCAT data are preferred, and LDEO observations are included for the years 1980-81 where SOCAT data are unreported. SOCAT fugacity ($f$CO$_2$) values are converted to pCO$_2$ and mole fraction (xCO$_2$) using reported SST and SSS data included in the products using routines from the CO2SYS software package (Van Heuven et al., 2011). The
anthropogenic effect of increasing surface pCO$_2$ is removed by calculating a fit linear trend of 2 $\mathrm{\mu atm\ yr^{-1}}$, slightly higher than ≈1.5 seen in Tjiputra et al. (2014). pCO$_2$ values are calibrated to the year 2005, the representative year used for the model's atmospheric xCO$_2$. The year 2005 is chosen for the model's xCO$_2$ concentration because it is the end of the historical period for the Intergovernmental Panel of Climate Change (IPCC) models used in its 5th report published 2014. The earliest SOCAT data comes from 1962, and different databases used in this study stem from similarly different timespans. As a result,
we assume there is a baseline seasonal cycle of pCO$_2$ and air-sea CO$_2$ flux which has held stable over the past decades.

Alternative pCO$_2$ products are used for comparison purposes. A complete list of these datasets and their characteristics is provided in Table 1. For all the comparison datasets, air-sea CO$_2$ flux is calculated from monthly values. ∆pCO$_2$ values are calculated using Keeling curve data (downloaded from https://www.esrl.noaa.gov/gmd/ccgg/trends/gl_data.html, downloaded September 2019) of atmospheric xCO$_2$ for the respective calibrated year of each data set. The same climatological winds
as used in the model (Sect. 2.2) are applied to the pCO$_2$ products. The gridded product TK09 is chosen because previous modeling studies in the AS use it as validation (see Introduction). The L15 climatology, while based upon the same *in situ* data mentioned above, represent different processing methodologies, and as a high-resolution, global pCO$_2$ dataset, also serves to provide independent context to the model validation. pCO$_2$ is also calculated from DIC and TA provided by the statistical fits to JGOFS data by Sarma (2003) and to the gridded GLODAP climatological product. The statistical fits of Sarma (2003)
are used twice, first using model SST,SSS, and Chl-*a*, and second with World Ocean Atlas (WOA) 2009 SST, SSS with SeaWifs Chl-*a*. GLODAP-derived pCO$_2$ also uses WOA2009 SST, SSS applied to the annual DIC, TA values. Calculations of pCO$_2$ are performed using the CO2SYS software package (Van Heuven et al., 2011). Since all calculations are conducted

at the near-surface, differences between this software suite and Orr and Epitalon (2015) are minimal. Furthermore, for air-sea $CO_2$ flux intercomparison purposes, all $pCO_2$ values except for TK09 are interpolated to the same $1^o x1^o$ grid already shared by GLODAP, WOA, and L15. Due to the model's higher resolution, the re-gridding process reduces the area covered, consequently lowering the total model $CO_2$ flux quoted in later sections of this study.

## 2.2 Model details and set-up

The model we use is the Regional Ocean Modeling System-Adaptive Grid Refinement In Fortran (ROMS-AGRIF) version 3.1.1. Shchepetkin and McWilliams (2005). Previously used in the AS by (Lachkar et al., 2016), the model is a free-surface primitive equation model, with a sigma and curvilinear grid for the vertical and horizontal dimensions, respectively. ROMS implements a forward-backward time-stepping alogrithm with split baroclinic and barotropic modes. The advection of tracers uses a rotated-split 3rd order upstream biased algorithm to reduce spurious mixing (Marchesiello et al., 2009). The K-profile parameterization (KPP; Large et al., 1994) for vertical mixing is used. The model domain spans from $5.3^oS$ to $30.5^oN$, and from $33^oE$ to $78.1^oE$ (Fig. 1). For the sake of comparison with Sarma et al. (2013), we will present the region north of the equator, and exclude the Red Sea and Arabian Gulf. The model's horizontal resolution is $1/24^o$, resulting in $\sim$5km horizontal grid spacing.

Coupled to the hydrodynamic model is a nitrogen-based biogeochemical model with two components for nutrients, nitrate and ammonium, with one phytoplankton, zooplankton, and two detrital pools (Gruber et al., 2006). Biological parameters for the model are the same as those used in Gruber et al. (2011). A carbon module is also applied to the model with the state variables of DIC, TA, and calcium carbonate ($CaCO_3$) (Gruber et al., 2012; Hauri et al., 2013; Lachkar and Gruber, 2013). In addition to usual physical transport and mixing, $CaCO_3$ is allowed to vertically sink at $20 \ mday^{-1}$. The chosen sinking rate is a simplification in that it does not include the faster rates observed for foraminifera shells (Curry et al., 1992), which as a biological group are not resolved by the biological model due to numerical constraints. Organic carbon is linked to organic nitrogen through the Redfield ratio 106:16. DIC is altered by air-sea $CO_2$ flux, primary production, respiration/remineralization, and dissolution/precipitation of $CaCO_3$. TA changes with the removal and creation of nitrate ($NO_3$), including nitrification and denitrification, as well as dissolution/precipitation of $CaCO_3$. The amount of $CaCO_3$ precipitation is linked to primary production through a constant ratio of 0.07, meaning 0.07 moles of $CaCO_3$ are produced for each mole of organic carbon. The dissolution rate is a constant $0.0057 \ day^{-1}$ in the water column and $0.002 \ day^{-1}$ in the sediments. Surface fluxes of DIC and TA due to evaporation, precipitation, and river input are included as virtual fluxes proportional to SSS forcing. Inside the module, surface carbon chemistry is calculated using routines from the Ocean Carbon-Cycle Model Intercomparison Project (OCMIP) carbonate chemistry routines (http://ocmip5.ipsl.jussieu.fr/OCMIP/phase3/simulations/). Carbon chemistry coefficients used here include $K_1$ and $K_2$ $CO_2$ dissociation from Millero (1995), original data from Mehrbach et al. (1973) and refit by Dickson and Millero (1987). A summary of the biological parameters used in the biogeochemical model is provided in Table 2.

The model is run with 360-day years and interpolated, climatologically averaged monthly forcing. The different climatological products derive from datasets spanning slightly different periods, and so here we assume that the dynamics represented within them have not changed in the time since. Heat flux, evaporation and precipitation, and restoring SSS are provided by

the Comprehensive Ocean-Atmosphere Data Set (COADS; da Silva et al., 1994). SST forcing is provided by a monthly climatology of Pathfinder data from 1985-1997 (Casey and Cornillon, 1999). Wind stress is produced using the QuikSCAT/SCOW monthly climatology from 1999-2009 (Risien and Chelton, 2008). Tracer values for the initial conditions and the boundaries are given by WOA 2009 for temperature, salinity, $NO_3$, and oxygen. Horizontal velocities u,v for initial and boundary conditions derive from the Simple Ocean Data Assimilation (SODA) analysis (Carton and Giese, 2008). Initial and boundary conditions for DIC and TA come from GLODAP from 300m down to the bottom. Surface TA was calculated using the relations from Lee et al. (2006), and the corresponding DIC was calculated using WOA phosphate, silicate, T, and S values along with L15 $pCO_2$. DIC and TA values between the surface and 300m are calculated using density-weighting. The model is spun up for 30 years, with 5 additional years for analysis. Atmospheric $xCO_2$ values are set to 380ppm, equivalent to 2005 levels, with an annual sinusoidal perturbation of 2.9ppm.

## 2.3 Domains of Analysis

In this study we focus on 6 distinct regions (Fig. 1). The first, the entire analysis domain, is the AS north of the equator. The upwelling regions of the Oman and Somalian coasts are included separately to focus on the summer monsoon impact of enhanced DIC but also enhanced biological productivity (Schott and McCreary Jr, 2001). The Oman region begins at the coast and extends 300km outward. The Somalia region begins near $3.8^oN$ and extends north to the tip of the Horn of Africa, with an eastern extension to $58.6^oE$ so as to encompass the region known as the Great Whirl (Vic et al., 2014), shown to be important for air-sea exchange in previous studies (Valsala and Murtugudde, 2015). The North region is defined by a rectangle from $59.4^oE$, $21^oN$ to $69.5^oE$, $26.5^oN$, encompassing the northern part of the AS where the winter monsoon's primary productivity is most intense (Kumar et al., 2001). An oligotrophic region representing the central AS, which has less productivity and chlorophyll-a on average (Fig. 1), is defined by a rectangle from $61.31^oE$, $3.3^oN$ to $70.8^oE$, $17^oN$. The last region, covering the western coast of India, extends from the coastline 100km offshore.

## 2.4 Analysis of air-sea $CO_2$ flux, $pCO_2$, and DIC variability

### 2.4.1 Air-sea $CO_2$ variability

The air-sea flux in the model is calculated using

$$F_{CO_2} = K_0\,\alpha\,(pCO_2^{sea} - pCO_2^{air})$$
$$= K_0\,\alpha\,\Delta pCO_2 \tag{1}$$

where $K_0$ is the solubility determined by temperature and salinity (Weiss, 1974), $\alpha$ is the $CO_2$ piston velocity with a quadratic wind speed dependence (Wanninkhof, 1992), and the difference in ocean and atmosphere $pCO_2$, $\Delta pCO_2$, is arranged so that the flux convention is positive outward from the ocean. The choice of Wanninkhof (1992) for the solubility parameterization is for direct comparison with previous modeling studies (see Introduction), despite the fact that more recent formulations are available, such as Wanninkhof (2014). The objective being to characterize seasonal anomalies of air-sea $CO_2$ flux, here

we use a Reynolds decomposition. Briefly, a Reynolds decomposition takes a timeseries and divides it into a temporal mean and fluctuating component. When applied correctly, multiple terms can be produced in isolation showing their fluctuating contribution to the total. Noting that temperature effects upon solubility ($K_0$) and piston velocity ($\alpha$) approximately cancel, meaning that their product mostly reflects wind forcing, we have the following arrangement for the decomposition of flux anomalies (Doney et al., 2009b):

$$F'_{CO_2} = \underbrace{(K_0\,\alpha)'\overline{\Delta pCO_2}}_{\text{wind}} + \underbrace{\overline{(K_0\,\alpha)}\Delta pCO'_2}_{\text{pCO}_2} + \underbrace{\left((K_0\,\alpha)'(\Delta pCO_2)' - \overline{(K_0\,\alpha)'\Delta pCO'_2}\right)}_{\text{cross terms}}, \tag{2}$$

where $'$ indicates an anomaly and $\overline{x}$ is a five-year average of variable $x$, which are calculated at each grid point. The five-year average is necessary for exact closure in the Reynolds decomposition. $F'_{CO_2}$ is the seasonal flux anomaly, with groupings based on wind anomalies $(K_0\,\alpha)'$, $\Delta pCO'_2$ anomalies, and cross-terms involving both.

The winds in this study are prescribed, so uncertainty in air-sea flux stems from $pCO_2$. The SOCAT protocol assigns a minimum uncertainty of 2µatm to observations. Using the average SST and SSS from the SOCAT observations, the solubility change is $2.68 \cdot 10^{-2}\,\text{mmolCm}^{-3}\mu\text{atm}^{-1}$. Wind speeds of 1, 5, and $10\,\text{ms}^{-1}$ will then produce a shift of 0.0018, 0.0443, and $0.177\,\text{molCm}^{-2}\text{yr}^{-1}$, respectively. The model presents a median value of $1.28\,\text{molCm}^{-2}\text{yr}^{-1}$ with median winds of $5\,\text{ms}^{-1}$, so therefore the baseline uncertainty in air-sea $CO_2$ is $\sim$3.5%.

### 2.4.2 pCO$_2$ variability

The proximate variables that affect $pCO_2$ change in the model are DIC, TA, SST, and SSS. Following previous studies (Lovenduski et al., 2007; Turi et al., 2014), we use a first-order Taylor expansion to decompose $pCO_2$ into contributions from these four, neglecting contributions from nutrients (phosphate and silicate). Initially, the decomposition would follow the form

$$\Delta pCO_2 \approx \frac{\partial pCO_2}{\partial DIC}\Delta DIC + \frac{\partial pCO_2}{\partial TA}\Delta TA + \frac{\partial pCO_2}{\partial SST}\Delta SST + \frac{\partial pCO_2}{\partial SSS}\Delta SSS \tag{3}$$

where $\Delta pCO_2$ is the perturbation of $pCO_2$ from a mean value, and the $\Delta$ terms for DIC, TA, SST, and SSS likewise express deviations from a prescribed value depending on whether the deviations are spatial or temporal in nature (see below). The coefficients of the $\Delta$ terms are partial derivatives of $pCO_2$ with respect to these variables, namely DIC, TA, SST, and SSS, and are calculated via centered differences described below. However, in order to control for salinity effects on DIC and TA (Keeling et al., 2004), we normalize DIC and TA by the salinity $S_0$=35 psu, to create the variables

$$DIC^s = S_0 \frac{DIC}{SSS} \ \ and \ \ TA^s = S_0 \frac{TA}{SSS}. \tag{4}$$

Substituting these terms into Eqn. (3), we can expand to produce, for example with DIC, the following (Lovenduski et al., 2007):

$$\begin{aligned}
\frac{\partial pCO_2}{\partial DIC}\Delta DIC &= \frac{\partial pCO_2}{\partial(SSS/S_0DIC^s)}\Delta(SSS/S_0DIC^s)\\
&= \frac{DIC^s}{S_0}\frac{\partial pCO_2}{\partial DIC}\Delta SSS + \frac{S}{S_0}\frac{pCO_2}{\partial DIC}\Delta DIC^s.
\end{aligned} \tag{5}$$

Collectively, the $\Delta$SSS term in Eqn. (5) and its counterpart in TA can be added to the original $\Delta$SSS term in Eqn. (3) to represent all salinity effects in a "freshwater" term, so that we now have (Turi et al., 2014):

$$\Delta pCO_2 \approx \underbrace{\frac{\partial pCO_2}{\partial DIC^S}\Delta DIC^S}_{\Delta pCO_2^{DIC^s}} + \underbrace{\frac{\partial pCO_2}{\partial TA^S}\Delta TA^S}_{\Delta pCO_2^{TA^s}} + \underbrace{\frac{\partial pCO_2}{\partial T}\Delta T}_{\Delta pCO_2^{T}} + \underbrace{\frac{\partial pCO_2}{\partial SSS}\Delta SSS}_{\Delta pCO_2^{SSS}}. \tag{6}$$

For the remainder of this paper, when discussing the results of the Taylor series decomposition method, it will be understood that DIC and TA refer to $DIC^s$ and $TA^s$, and SSS will refer to the combined term.

The contributions of DIC, TA, SST, and SSS to $pCO_2$ variability are used to construct maps and timeseries of $pCO_2$ anomalies. In order to calculate the anomaly $\Delta pCO_2$ requires calculating both the $\Delta$ deviations of DIC, TA, T, and SSS, as well as partial derivatives. In this study, we calculate both temporal and spatial anomalies. To consider spatial variability, starting with annual means of $pCO_2$, DIC, TA, SST, and SSS, an average value for the whole domain is calculated and removed from each grid point's annual mean to get a $\Delta$ perturbation, or anomaly. Similarly, for temporal variability, with the monthly values of

$pCO_2$, DIC, TA, SST, and SSS at each grid point, the annual average at that grid point is removed to produce the monthly $\Delta$ perturbation/anomaly. Partial derivatives are approximated via centered differences. These are obtained by calculating $pCO_2$ with slight deviations of DIC, TA, SST, and SSS from the mean value. Both positive and negative deviations are used to construct centered differences, with deviation magnitude determined by Orr et al. (2018). For example to calculate the monthly $pCO_2$ anomaly due to SST for a gridpoint with annual mean $pCO_2$ of 430$\mu$atm, annual mean SST of 24$^o$C, and monthly SST

of 26$^o$C:

$$\Delta pCO_2 \approx \frac{\partial pCO_2}{\partial SST}\Delta SST + ... \approx \frac{pCO_2(24 + 1\cdot 10^{-4},...) - pCO_2(24 - 1\cdot 10^{-4})}{2\cdot 1\cdot 10^{-4}} \cdot (26 - 24) + ... \tag{7}$$

where $1\cdot 10^{-4}$ is the recommended SST deviation.

### 2.4.3 DIC budget

Whereas the state variables of DIC, TA, SST, and SSS provide the chemical context which determines carbon availability to

240 potential air-sea flux via $pCO_2$, tracking the overall inventory of inorganic carbon (i.e. DIC), allows for the parsing of numerous source and sink processes governing the total amount of carbon reaching the surface. Beyond the biological processes impacting DIC as outlined in Sect. 2.2, the physical processes impacting DIC are air-sea $CO_2$ flux, surface evaporation and precipitation, horizontal and vertical advection, and horizontal and vertical mixing. In order to diagnose the relative importance of these terms (i.e. to weigh competition between upwelling circulation-source and biological drawdown-sink), we calculate the budget

$I_{DIC}$ in a 3D volume by integrating:

$$I_{DIC} = \iint\limits_{\mathbf{A}} \int\limits_{-z(\sigma)}^{\eta} J(x,y,z)\, d\mathbf{A}\, dz \tag{8}$$

with

$$J = \underbrace{-PP_{New+Reg} - CaCO_{3prec-remin} + Zoo_{resp} + Det_{remin}}_{Biology} - \underbrace{F_{AS}}_{Air-Sea}$$

$$+ \underbrace{Adv_x + Adv_y + Mix_x + Mix_y}_{Horz.\,Circ} + \underbrace{Adv_z + Mix_z}_{Vert.\,Circ} + \underbrace{Evap - Precip}_{Forc}, \tag{9}$$

which is the volume-specific flux $J$ of DIC in a given grid cell. $PP_{New+Reg}$ is net community primary production scaled by
the Redfield ratio, $CaCO_{3remin-prec}$ is net $CaCO_3$ precipitation and remineralization, $Zoo_{resp}$ is zooplankton respiration,
and $Det_{remin}$ is remineralization of both detrital pools. All these terms are grouped together into $Biology$ because they
represent all biological processes. $F_{AS}$ is air-sea flux, with a sign convention of positive outward. $Adv_x$ is advective flux in the
x-direction, with corresponding y and z components. $Mix_x$ is the x-component of mixing flux, again with y and z components.
All x and y components of both advective and mixing DIC fluxes are grouped into horizontal circulation, with a similar
grouping for vertical circulation in the z-direction. $Evap - Precip$ is the forced virtual flux from evaporation and precipitation
at the surface. **A** is the two-dimensional horizontal area to be considered, which in our study includes the entire domain but also
the sub-regions of analysis. The bottom boundary of integration, $-z(\sigma)$, is the sigma-layer depth at which integration starts,
moving up to the free-moving surface $\eta$. We chose to integrate the top five sigma layers of the model, corresponding to $\sim$20m
depth. This level was chosen because below this depth, annual cycles of $I_{DIC}$ begin to deviate from the surface DIC, which is
our focus in this study of air-sea $CO_2$ flux.

## 3   Results

### 3.1   Model validation and pCO$_2$ data-model comparisons

The implementation of ROMS-AGRIF presented here has been used in previous studies of the AS (Lachkar et al., 2016).
Model output of net primary productivity (NPP) captures the summer monsoon highs near the upwelling regions of Oman and
Somalia (model >400 vs data >500gCm$^{-2}$yr$^{-1}$), with enhanced NPP in the North during the winter monsoon (model $\sim$300
vs data >400gCm$^{-2}$yr$^{-1}$) (Fig. 1). The model also captures the vertical distributions of temperature and salinity (Fig. S1-2)
with deviations from WOA around 1$^{\circ}$C and 0.2psu. Depth profiles of nitrate, oxygen, DIC, and TA are similarly conserved
(Fig. S3-S6). Nitrate, DIC, and TA all show their usual nutrient-like profiles, while oxygen is its minimum within the OMZ.
The deviations seen between *in situ* data and model output are greatest at depths less than 500m. Deviations in near-surface
NO$_3$ (Fig. S3) can be large for intermediate values (5-20μM) but overall do not show a systematic bias. DIC (Fig. S5) also
has large deviations ($\sim$50μM) in the top 500m and with a slight positive bias. It is in TA (Fig. S6) that deviations, while
similarly $\sim$50μM $-$ eq, show a consistent near-surface underestimation. The surface currents in the model also demonstrate
the monsoonal shifts and reversals seen in the AS (Fig. S7).

Regarding pCO$_2$, *in situ* data from the merged SOCAT/LDEO database shows that $\sim$90% of $\Delta$pCO$_2$ values in the AS
are positive (Fig. 2a, inset), indicating a positive flux to the atmosphere that is applicable geographically (Fig. 2a). Sampling

dates for $pCO_2$ (Fig. 2b) show that ∼70% are from the summer monsoon months (June-September, JJAS). Most observations similarly date from the 1990s, with 1995 and 1997 alone accounting for 96%.

Seasonal $pCO_2$ distributions from both data and the model are shown in Fig. 3. During the winter monsoon, $pCO_2$ values are at their lowest (range: 348-455 µatm; Fig. 3a). Spring intermonsoon (Fig. 3d) finds $pCO_2$ values similar to the winter (range: 354-451 µatm), with data coverage improving in the western AS. Summer monsoon, with best data coverage (Fig. 3g), has $pCO_2$ peaking at 773 µatm. In contrast, the fall intermonsoon (Fig. 3j) has very little data coverage, with $pCO_2$ ranging from 311-485 µatm. Similar to the data, model $pCO_2$ (Fig. 3b) is at its lowest during the winter. However, in the spring (Fig. 3e) open-ocean $pCO_2$ finds its peak with a domain-average of 439µatm, which is not reflected in the *in situ* data set (Fig. 3d,e). Maximum model $pCO_2$ is found in the summer monsoon near upwelling regions (Fig. 3h), with values attaining >800µatm in Oman. Fall model $pCO_2$ (Fig. 3k) still has elevated values averaging 427 µatm, but less than the summer period. Certain regions in the model show persistent maxima in $pCO_2$, such as the Gulf of Oman and the Strait of Hormuz, which are not reflected in the few data collected there. Model $pCO_2$ values in the Gulf of Aden increase during spring and then peak during the summer, a pattern which is unclear from the data. Annual and seasonal $pCO_2$ means, with standard deviations in parentheses, are displayed in Table 3 for both the data and model. Differences from interpolated model output and *in situ* data are shown on the right column of Figure 3 (Fig. 3c,f,i,l). Most differences show that model output is higher in value than the data, averaging 24.6, 48.4, and 33.7 µatm higher for the winter, spring, and fall seasons, respectively.

A Taylor diagram (Taylor, 2001) comparing *in situ* $pCO_2$ data with model output shows the model's relative performance (Fig. 4). The distance from the origin is model variability normalized by standard deviation of the *in situ* data. The angle created from the y-axis is the Pearson correlation coefficient between the model and *in situ* data. If the model were to perfectly reproduce the data, it would appear at the position (1,0), equivalent to a normalized standard deviation of 1, and correlation coefficient of 1. For the entire dataset, as well as for the spring and summer seasons, the model's correlation with data is ∼0.5. Winter and fall have lower values at 0.2 and 0.06, respectively. Variability expressed as normalized standard deviation shows that overall, and during spring and summer periods, the model under-estimates data variability (∼0.5µatm). During the winter and fall, however, the model over-estimates variability (1.1 and 1.6, respectively). For all periods apart from summer, model $pCO_2$ has a positive bias (9.1, 24.6, 48.4, and 33.7 µatm for the annual, winter, spring, and fall, respectively). During the summer, the model has a negative bias of -3.1 µatm.

The source of bias in $pCO_2$ is linked to the four state variables SST, SSS, DIC, and TA. Comparisons with the model are made with SST and SSS from the merged LDEO/SOCAT database, while DIC and TA come from the ungridded GLODAP product (Fig. S8). In this case, model SST and SSS (Fig. S8a,b) largely overlap with a 1:1 relationship, but with slight positive biases of ∼0.4°C and 0.3psu. Removing these biases from the model results in a $pCO_2$ shift of -6.8 and -3.5 µatm for SST and SSS, respectively. These deviations are close in magnitude to the best-case measurement error of ∼2µatm. Taylor diagrams for SST and SSS (Fig. S9) further show the seasonal performance of these two variables. The model performs best for SST (Fig.S9a) during the winter, with correlation of 0.93 and normalized standard deviation of 0.97. The other seasons have lower correlations (0.74-0.81), and reduced standard deviations (0.63-0.8), except for the fall with standard deviation of 1. SSS (Fig. S9b) has lower correlations and standard deviations than SST, with all seasons demonstrating a positive bias (0.02-0.39psu).

Correlation is best in the winter at 0.89, and worst in the fall at 0.46. Model variability in SSS is also less than the data, with standard deviations ranging from 0.33 to 0.72. Lower variability is most likely due to the raw nature of the *in situ* data used here, in opposition to the monthly averaged climatological forcing and initial conditions of the model.

Ungridded DIC and TA data from GLODAP, though more sparse (n=334 data points with both DIC and TA at depth $\leq 50$m), show more deviation from the 1:1 line (Fig. S8c,d) with overall negative biases of -15.8 $\mu molkg^{-1}$ and -30.0 $\mu mol - eqkg^{-1}$ for DIC and TA. These biases result in $pCO_2$ perturbations of -33.8 and +45.7 $\mu atm$, respectively, when accounted for individually. Since the buffering capacity of seawater is related to the ratio of TA and DIC, when both biases are considered average $pCO_2$ shifts +16.7 $\mu atm$. As a result, while the DIC model bias lowers $pCO_2$, the stronger bias in TA is the most likely cause for the model's overall positive $pCO_2$ bias, which may in part be due to the unresolved fast sinking rates of foraminifera in the model.

Direct comparisons between the *in situ* and model output demonstrate the positive bias and middling correlations of the model with respect to the data, as well as the model's tendency to under-represent variability. As a result, it is necessary to investigate how these shortcomings compare with alternative $pCO_2$ estimates in the AS. Figure 5 shows monthly comparisons of the $pCO_2$ probability distribution functions from *in situ* data, model output, and L15. For most of the year, the data (Fig. 5a) stays within a relatively narrow range (375-425 $\mu atm$), except for the summer monsoon where values can exceed 500 $\mu atm$ and the median value has its peak. In the model (Fig. 5b), $pCO_2$ is almost entirely above 400 $\mu atm$, with the median value increasing during spring inter-monsoon and peaking in June (453$\mu atm$). Similar to the data, the upper bound variability in $pCO_2$ peaks in August. L15 (Fig. 5c), by contrast, has a tighter envelope of variability, with 5-95 percentile values never going beyond the range of 368-434 $\mu atm$. Median $pCO_2$ in L15 peaks in the summer like the data at 402$\mu atm$, but there is no large increase in upper bound variability, with the 95% upper bound in L15 reaching 434$\mu atm$ in September.

In summary, the survey of available data and comparing it to the model output produces a few distinct features: 1) available *in situ* data shows that the majority of observations are skewed towards the summer monsoon during the years 1995 and 1997; 2) most *in situ* data show $CO_2$ out-gassing in the AS; 3) the model has a net positive bias in surface $pCO_2$, driven by a joint DIC-TA bias which is slightly stronger in TA; and 4) the model captures the high summer monsoon $pCO_2$ values better than the alternative L15 climatology.

## 3.2 Air-sea $CO_2$ flux, drivers of seasonal variability, and flux intercomparison

Modeled annual mean atmospheric flux of $CO_2$ (Fig. 6a) shows outgassing (positive, red) throughout the entire domain, producing an average annual $CO_2$ flux density rate of 1.9 mol C $m^{-2}yr^{-1}$ and a total of 162.6TgC $yr^{-1}$. Similar to $pCO_2$, several hotspots appear in the geographic distribution. Near the coast of Oman, the average flux density is 2.7, with 3.2 in Somalia and 2.4 along the coast of India, producing a flux of 11.4, 32.9, and 4.9 $TgCyr^{-1}$, respectively. The other regions, the North AS and oligotrophic central AS, have average densities of 2.0 and 1.5 mol C $m^{-2}yr^{-1}$, with total fluxes of 10.5 and 28.6 $TgCyr^{-1}$. The seasonal air-sea flux (Fig 6b-e) has minima during fall and winter, with an increase in spring and a strong maximum during summer monsoon. Oman and Somalia flux densities during summer monsoon are 5.8 and 5.9 mol C $m^{-2}yr^{-1}$, respectively. The distribution of enhanced summer air-sea $CO_2$ flux coincides with the southwest monsoon winds, (Fig. S10) as well as the band of cooler temperatures impacting spatial $pCO_2$ anomalies (see Sect. 3.3.1). The entire domain fluxes 32.0, 26.6, 90.9, and

13.1 TgCyr$^{-1}$ for the winter, spring, summer, and fall periods, respectively, each contributing 19.7, 16.3, 55.9, and 8.1% of the annual total.

The variability in air-sea CO$_2$ flux can be attributed to the contributions of winds, $\Delta$pCO$_2$, and interacting cross-terms, as described in Eqn. (2). The temporal anomalies for the summer monsoon, the period with strongest CO$_2$ flux signal, are presented in Figure 7. Most of the domain has positive but variable strength anomalies in air-sea flux (Fig. 7a), averaging 1.3molCm$^{-2}$yr$^{-1}$ with a standard deviation of 1.35. The wind contribution to flux variability, $\kappa\alpha$ (Fig. 7b), is also positive in most of the domain except the Gulf of Aden and the south-eastern corner of the domain. The wind anomaly's magnitude and distribution closely match the total anomaly in Fig. 7a, with mean flux anomaly of 1.18molCm$^{-2}$yr$^{-1}$ and 0.96 standard deviation. The $\Delta$pCO$_2$ contribution to seasonal flux anomaly (Fig. 7c) has a lower magnitude effect overall (mean flux anomaly 0.1, deviation 0.5, maximum 6.2 molCm$^{-2}$yr$^{-1}$), with positive values north of $10^o$N and slightly negative to the south. The maxima approaching 6.2 molCm$^{-2}$yr$^{-1}$ are in the upwelling centers of Oman, Somalia, and the Indian coast. Second-order cross-term values (Fig. 7d) are almost all positive, with maxima also occurring near upwelling centers similar to the $\Delta$pCO$_2$ term, but weaker in magnitude with average 0.04molCm$^{-2}$yr$^{-1}$.

The seasonal flux anomalies for all regions are displayed in Fig. 8. The summer monsoon flux is so strong that it makes the anomalies (black lines) for all the other seasons negative, except for May in the spring. During the winter months DJFM, both wind and pCO$_2$ terms produce negative flux anomalies (ranging to -0.78 and -0.38 in the domain for wind and pCO$_2$, respectively; Fig. 8a), indicating the relative lack of winds and minimum pCO$_2$ values. In winter, while the negative wind term is universally strongest, within the upwelling regions the pCO$_2$ term is 58% (Fig. 8b) of the wind term's magnitude, and 49% for the entire domain. The spring intermonsoon, where many regions such as Somalia and the central oligotrophic AS (Fig. 8d-e) experience their pCO$_2$ maximum, shows a positive pCO$_2$ effect on flux anomaly that is as large as or larger than the negative wind effect (Somalia May pCO$_2$ anomaly of 1.1molCm$^{-2}$yr$^{-1}$, wind anomaly of 0.1). Summer monsoon winds represent the majority contribution to CO$_2$ flux variability, with a minimum 64.7% contribution relative to the total anomaly in India, a maximum of 112.8% in the oligotrophic AS, and 90.8% for the whole domain. By contrast, summer pCO$_2$ and cross-terms contribute 6.0% and 3.1% to the domain's anomaly, respectively. Fall inter-monsoon months resemble the winter monsoon, with negative wind anomalies contributing most with small or negative pCO$_2$ contributions. In most scenarios, pCO$_2$ contributes in the same direction as the winds or little at all, with the notable exceptions of Oman, oligotrophic AS, Somalia, and the domain during spring inter-monsoon.

While strong monsoon winds dominate the timing of air-sea CO$_2$ flux, and the AS is always a source of CO$_2$ due to positive $\Delta$pCO$_2$, differences in pCO$_2$ between independent sources can still result in a wide range of overall magnitudes. In the AS, CO$_2$ outgassing estimates vary from 7 TgCyr$^{-1}$ (Goyet et al., 1998b) to >90 TgCyr$^{-1}$ (Sarma, 2003), and everything in between (Somasundar et al., 1990), with each study using their own pCO$_2$ data and wind parameterizations. Considering the important seasonal role of winds, the best way to investigate the role of pCO$_2$ variability is to keep winds (and their flux parameterization) constant. Towards this end, we use multiple pCO$_2$ products to calculate CO$_2$ flux with the same wind and parameterization as the model (Fig. 9). As summarized in Table 1, pCO$_2$ from TK09, L15, GLODAP data and Sarma (2003), interpolated to the WOA $1^o$x$1^o$ grid, are used in these calculations (except for TK09 where the coarse resolution reduced

coverage). The original applicability of the Sarma (2003) model is north of $10^o$N, and so flux is calculated for this region, as well.

All calculations have their peak $CO_2$ flux sometime in the summer, confirming the role of winds in $CO_2$ flux timing. This study's model consistently produces one of the higher estimates with $120 TgCyr^{-1}$ (reduced from 162.6 due re-gridding) and $57 \ TgCyr^{-1}$ north of $10^o$N. The only estimate higher than the model is GLODAP data in the region north of $10^o$N with 65

$TgCyr^{-1}$ possibly driven by summer monsoon sampling bias. The high model estimate is perhaps unsurprising, considering the $pCO_2$ bias. The range in estimates of total $CO_2$ flux is 57-120 $TgCyr^{-1}$, resulting in a ratio of 2.1x variability. In the reduced domain of the AS north of $10^o$N, estimates range from 12.3 to 65.6, resulting in 5.3x variability. The 5.3x ratio is quite high, and is in part driven by the low estimates from the Sarma (2003) model, which are 12.3 and 17.6 using tracer data from WOA and ROMS, respectively. Indeed, the Sarma (2003) model estimates have negative $CO_2$ flux for some months, which is

not observed in the original publication, and the total fluxes are quite smaller than the 70 $TgCyr^{-1}$ reported. If the two lower estimates are removed, the range in air-sea $CO_2$ flux in the domain north of $10^o$N is 41-65 $TgCyr^{-1}$, providing a ratio of 1.6 similar to 2.1 for the whole domain. Even considering the model's $pCO_2$ bias, as previously mentioned the GLODAP estimate supersedes it in the region north of $10^o$N, as does the original Sarma (2003) estimate of 70 $TgCyr^{-1}$. Thus, while we may think the model over-estimates flux, it is still within the range of previous studies in the AS.

## 3.3 $pCO_2$ distribution, seasonal cycle, and underlying contributors

### 3.3.1 Spatial $pCO_2$ distribution

Spatial $pCO_2$ anomalies calculated from the annual mean highlight the geographic hotspots of $pCO_2$ inside the domain (Fig 10a). $pCO_2$ anomalies range from -89 to +415µatm, indicative of a positive skew in the distribution. Within the regions of analysis prescribed in this study, it is clear that Oman, the Indian coast, and the North AS host enhanced $pCO_2$, with average

positive anomalies of 8.6, 21.5, and 49µatm, respectively. In contrast, both the oligotrophic central AS and Somalia regions have negative $pCO_2$ anomalies (-13.7 and -2.9µatm, respectively). The contributing factors to these $pCO_2$ anomalies, SST, DIC, TA, and SSS, display differing distributions. SST (Fig. 10c) contributes toward negative $pCO_2$ anomalies in a southwest-to-northeast band along the coasts of east Africa and the Arabian peninsula, up to the coasts of Pakistan and the northern coast of India near Gujarat. The cold SST structure contributes a -20µatm effect on $pCO_2$, and largely overlaps the stronger

summer monsoon winds (Fig. S10). The opposite trend is found in the central oligotrophic and Indian regions, where the average temperature contribution to $pCO_2$ is 20µatm despite upwelling along the southern Indian coast. The distribution of DIC-induced anomalies (Fig. 10d) shows a positive influence near coastal regions and the western AS off the coast of Somalia (+25µatm), whereas a strong minimum is found in an oval region encompassing the central, open-ocean AS (-36.6µatm). TA effects (Fig. 10e) show a north-south gradient similar to SSS, with positive contributions to $pCO_2$ of +20µatm occurring in the

north and -20µatm towards the south, resulting in magnitudes similar to SST contributions. SSS contributions (Fig. 10f) show a similar distribution as TA, but weaker in magnitude ($\pm$10µatm).

### 3.3.2 Seasonal pCO$_2$ cycle

The previous section outlines the geographic regions within the AS that have overall high or low pCO$_2$ values, but in order to investigate the strong seasonal monsoon cycle in the AS, the decomposition of variables affecting monthly pCO$_2$ values is calculated at each model grid point and averaged into each analysis region (Fig. 11). Regarding the whole domain (Fig. 11a), pCO$_2$ variability is similar to that seen in Fig. 5b, with a spring pCO$_2$ anomaly peak (20 μatm) and minimum during fall and winter (-9.4 μatm). Temperature effects largely mirror the overall pCO$_2$ cycle (May peak 30, January minimum -17μatm). Change in pCO$_2$ associated with DIC acts in opposition to temperature but with lower magnitude (16 in February, -8 in June). Both TA and SSS effects are negative for the first half of the year before becoming slightly positive in the second half, never reaching 10μatm in magnitude.

Different pCO$_2$ anomaly cycles can be found in the upwelling regions of Oman, Somalia and India (Fig. 11b,e,f). Here, a positive temperature peak appears in the spring (27-45μatm), which is then supplanted by a positive DIC peak during the summer monsoon (41-81μatm). In both Oman and India, the summertime DIC peak is strong enough to contribute to the annual pCO$_2$ peak despite cooler temperatures. In Somalia, the summertime DIC peak is not sufficiently stronger than temperature (41 vs -34μatm) such that in sum with the other terms maximum pCO$_2$ is found in the spring, not the summer, similar to the whole domain and oligotrophic regions. Both TA and SSS effects in these three regions are lower in magnitude (never exceeding 18.4, 7.3μatm for TA and SSS, respectively) and generally run counter to DIC.

A completely different regime occurs in the North AS (Fig. 11c). Here, while temperature effects (49μatm in June) create a similar spring-summertime peak in pCO$_2$ (15.9μatm) somewhat counter-acted by DIC (-40μatm), during the winter monsoon temperature and DIC effects are both maximal and in opposing amplitudes (-49.5 and 51.4μatm for SST and DIC, respectively). This occurs due to the convective mixing that occurs during winter in the North AS, where cooling temperatures lower pCO$_2$ but subsurface water introduces more DIC, resulting in a near-balance.

The oligotrophic central region (Fig. 11d), the largest in area, has similar pCO$_2$ and temperature impacts as the whole domain, with the two largely overlapping. DIC, TA, and SSS impacts also follow similar patterns, but have slightly higher magnitudes in the central AS, with DIC reaching 32μatm.

### 3.4 Near-surface DIC budgets and cycling

SST's effect on pCO$_2$ reflects physical processes like surface heating and cooling, mixing and advection. DIC, by contrast, reflects both physical and biological processes because in addition it is also impacted by photosynthesis, CaCO$_3$ shell formation and dissolution, zooplankton respiration, detritus remineralization (bacterial respiration), and air-sea exchange. Budgets of DIC fluxes in the upper 20 m (Fig. 12; see Fig. S11 for a volume-specific DIC flux) show that two major processes dominate, vertical circulation (light blue lines) and net biological processes (magenta lines). In the entire domain and all sub-regions, and for all months, vertical circulation (advection and mixing) acts as a source of DIC, with the sum of all biological processes acting as a sink (n.b. the top 20 m does not constitute the entire euphotic zone, so respiration and remineralization at depth is not included). Maximum magnitudes of both vertical circulation and biological flux occur during the summer monsoon for all regions, except

for the North AS where they occur during the winter monsoon bloom (Fig. 12c). The maximum DIC flux in the domain due to vertical circulation is $1.76 \mathrm{PgCyr}^{-1}$, whereas biological flux peaks at $-1.0 \mathrm{PgCyr}^{-1}$. Biological fluxes are nearly phase-matched with vertical circulation, though peaks in summer biological flux lag vertical circulation by a month (Fig. 12d,e,f). Comparing the two flux terms, after normalizing biological flux by vertical circulation flux, the relative strength of biological processes versus vertical sources of DIC becomes apparent. In the whole domain, biological flux ranges from -90% to -34.5% of vertical

flux, similar to Rixen et al. (2005). As a result, biological fixation of carbon is generally weaker than physical vertical delivery of DIC.

Air-sea flux (red lines) is always negative due to the high $pCO_2$ values, peaking during the summer monsoon. DIC flux due to atmospheric escape, while reaching its maximum magnitude of $\sim 0.32 \mathrm{PgCyr}^{-1}$ in June and July for the whole domain (Fig. 12a), only surpasses biological flux in May, when $0.23 \mathrm{PgCyr}^{-1}$ is releasing to the atmosphere compared to $0.15 \mathrm{PgCyr}^{-1}$

in biological processes. Evaporation and precipitation (brown lines) results in higher DIC for most of the year in the entire domain and upwelling regions (i.e. net evaporation, averaging $0.07 \mathrm{PgCyr}^{-1}$ in the domain) , except India where it is negative (net precipitation, averaging $-4.8 \cdot 10^{-3} \mathrm{PgCyr}^{-1}$). The oligotrophic region's evaporation and precipitation flux (Fig. 12d) oscillates from being either positive or negative four times during the year, with magnitudes rivaling air-sea flux at times ($5 \cdot 10^{-2} \mathrm{PgCyr}^{-1}$). Horizontal advection (dark blue lines) is negative on average for the whole domain ($-0.2 \mathrm{PgCyr}^{-1}$), denot-

ing net export (Fig. 12a). The same pattern occurs for all sub-regions except India with net horizontal import of surface DIC (Fig. 12f; $2.9 \cdot 10^{-3} \mathrm{PgCyr}^{-1}$). The Oman upwelling region and the oligotrophic region experience positive peaks of horizontal import during the summer monsoon (27 and $56 \mathrm{TgCyr}^{-1}$ for Oman and oligotrophic regions, respectively), though for Somalia this period is the maximum DIC export, peaking at $220 \mathrm{TgCyr}^{-1}$ in July.

## 4   Discussion

### 4.1   Model pCO$_2$ vs. data

The $pCO_2$ output from the model has a positive bias with respect to the *in situ* data, as is clear from Fig. 3-5. The question becomes whether the model bias precludes its use in acquiring a reasonable air-sea $CO_2$ flux estimate. Regarding the direction of $CO_2$ flux (positive outgassing or negative uptake), since most *in situ* $\Delta pCO_2$ data are already positive (Fig. 2), an additional positive bias will not impact flux direction, reaffirming the previous findings of Sarma et al. (1998) and subsequent work

demonstrating that the AS is a source of $CO_2$ to the atmosphere. A positive model bias in $pCO_2$ has been noted in previous modeling studies. For instance, in the global data assimilation study of Valsala and Maksyutov (2010), they found an overall positive bias in the North Indian ocean, $\sim +5\text{-}15 \mathrm{\mu atm}$ above TK09 (compared to our -3.1 to $+48.4 \mathrm{\mu atm}$ with respect to *in situ data*). Additionally, that study found a similar underestimate near the upwelling regions (summer negative bias in the model) of the AS and overestimate elsewhere (their Figures 3 and 4). In Sreeush et al. (2019a), ROMS resulted in systematic positive

$pCO_2$ bias, whereas the offline Ocean Transport Tracer Model (OTTM) produced negative bias in $pCO_2$ in comparison to TK09.

The search for the model bias source is hindered by the lack of *in situ* data in the region. As already noted, GLODAP has 334 locations with DIC and TA in the top 50m. The few available *in situ* data that *do* exist in the AS have a number of deficiencies for the purpose of validating model output. First, the data available are both old and concentrated around the years 1995 and 1997.

While the JGOFS studies were quintessential in diagnosing the seasonal cycle of $pCO_2$, they preclude being able to decipher the secular trend in surface $pCO_2$ due to increasing atmospheric $CO_2$ concentrations. In our analysis, we estimated a $+2\mu atm\ yr^{-1}$ trend, close to that of Tjiputra et al. (2014), though finding an inter-annual linear trend requires more data at regular intervals. Second, due to the nature of strong upwelling in the AS, previous cruise sampling also biases not only the summer months ($\approx$70% of data), but also in the vicinity of the Oman coast (Fig. 3g). As a result, it is difficult to determine to what extent the

data are representative of the entire AS. Consider that in the model, flux intensities are lower in the central, oligotrophic region (Fig.6), but due to its surface area the total flux ($28.6\ TgCyr^{-1}$) was close to that of Somalia ($32.9\ TgCyr^{-1}$), an observation also made by Lendt et al. (2003). Determining to what extent the model over- or under-estimates $CO_2$ flux due to $pCO_2$ bias would require more *in situ* sampling, which would need to be designed around solving the problems of areal coverage (outside of Oman and upwelling zones) and temporal coverage (off-summer months and recurrent over multiple years).

The distribution of model $pCO_2$ is both similar to and different from previous data-based and modeling studies. Apart from the aforementioned bias leading to heightened absolute values (though Bates et al. (2006) has $>400\ \mu atm$ for large parts of the AS), the relatively enhanced $pCO_2$ values near Oman, along the west coast of India, and in the Gulf of Aden have already been observed (Sabine et al., 2000; Bates et al., 2006; Sarma et al., 2000; Körtzinger et al., 1997). These same studies, however, note a minimum of $pCO_2$ outside of the summer monsoon near the south-west coast of India due to freshwater influx, which is

not replicated well in the model. Additionally, elevated $pCO_2$ near the equator is not observed (Sabine et al., 2000; Bates et al., 2006), although it can appear in other models (Valsala and Murtugudde, 2015). The model's seasonal $pCO_2$ minimum during the winter monsoon is also not reflective of results found elsewhere (Goyet et al. (1998a, b); Bates et al. (2006); though many studies highlight the North AS, where minimum model $pCO_2$ occurs during the spring). Instead, these papers state $pCO_2$ is minimal during the fall inter-monsoon. Likewise, the large-scale spring maximum of $pCO_2$ seen in the model is not found in

these studies, except for in Louanchi et al. (1996), though this result is somewhat anomalous since that study showed a $pCO_2$ minimum during summer monsoon. Thus, while the model agrees with previous work insofar as the coastal regions impacted by upwelling show enhanced $pCO_2$, mismatches do appear in the seasonal timing of maxima and minima, especially within certain sub-regions.

Despite the model's limitations, its advantages are also clear. Beyond the obvious increase in spatio-temporal coverage,

capturing the monsoon's strong seasonal dynamics helps the model where other approaches fall short. This is especially illustrated in Fig. 5. Since upwelling regions are limited in geographic extent near the coast, capturing their high $pCO_2$ values can be difficult for other approaches, such as TK09 with its coarse grid. Even the L15 product, with its finer grid, is unable to produce the higher $pCO_2$ values seen during the summer. Judging from these comparisons, the trade-off appears to be that the model currently may produce less accurate $pCO_2$ values outside of summer, but the explicit resolving of upwelling allows for

enhanced $pCO_2$ values during the summer monsoon, the peak of $CO_2$ flux.

## 4.2 Spatial distribution of air-sea $CO_2$ flux and $pCO_2$

The model results both affirm the conclusions of previous studies in terms of $CO_2$ flux direction and seasonality, yet find difference in magnitudes. As previously stated, the AS is a atmospheric $CO_2$ source, with most flux occurring (56%) during the summer monsoon (Fig. 6). In our results, however, there is no region during any of the seasons where $CO_2$ uptake takes place. While somewhat expected, this is still in disagreement with some of the other $pCO_2$ datasets previously considered, such as in Sarma (2003), where negative $\Delta pCO_2$ values appear, such as during winter monsoon near the south coast of India. The model's positive $pCO_2$ bias may be to blame for this, making it so that no negative $\Delta pCO_2$ appears. Despite the positive $pCO_2$ bias, a few other patterns are clear in comparison to other $CO_2$ flux estimates. Sabine et al. (2000) and Sarma (2003) both find the maximum flux occurring during the summer monsoon centered around the upwelling regions, which is also quite visible in the model results (Fig. 6d). However, Bates et al. (2006) found that a secondary maximum of flux occurs during the winter monsoon, though due to the color scale in their figure 6 it is difficult to ascertain much beyond $CO_2$ outgassing from the AS during all months of the year. Their secondary max in flux may be partly attributable to higher wintertime $pCO_2$, as well.

The spatial decomposition of factors influencing $pCO_2$ (Fig. 10) highlights how geographically DIC can be the strongest factor, with SST and TA taking secondary roles and SSS being a weak contributor. Since DIC and TA can co-vary with salinity, when they are not normalized their distribution in the AS mirrors the north-south salinity gradient (see figures 2,3 in Bates et al. (2006)). Once corrected for salinity, it is clear that the upwelling region of Oman still has elevated DIC whereas the central, oligotrophic AS shows a DIC deficit. By contrast, the onshore-offshore gradient in TA is weaker. Differences between coastal and offshore normalized DIC and TA in the AS have been previously observed (Millero et al., 1998b; Lendt et al., 2003), but the stronger relative absence of DIC in the central AS and its role in affecting $pCO_2$ has not been emphasized. A similar analysis in the California Current upwelling system (Turi et al., 2014) indicates near-compensation of DIC and temperature in opposing directions, nearly overlapping each other. In that scenario, DIC overpowers temperature at the coast, with TA and SSS being secondary. For the AS, while the upwelling regions of Oman and Somalia show temperature and DIC working against each other, they are not as well compensated. Furthermore, the gradients of positive/negative $pCO_2$ contributions from temperature and DIC do not overlap, leading to the curious scenario where temperature and DIC both contribute positively to the $pCO_2$ anomaly along the Indian coast. The positioning of these gradients and the surprising negative influence of DIC away from upwelling regions perhaps underscores how the AS is rather unique, where strong seasonal upwelling winds mingle with strong tropical heating and the influence of outflows from marginal seas (Prasad et al., 2001; l'Hegaret et al., 2015).

## 4.3 Seasonality of air-sea $CO_2$ flux, $pCO_2$, and DIC

### 4.3.1 Air-sea $CO_2$ Flux

The fact that model $CO_2$ flux for the entire domain peaks in summer despite a spring peak in $pCO_2$ for the domain as a whole (along with the Somalia and oligotrophic regions) is the first sign that perhaps $pCO_2$ is not the primary driver in determining flux timing. The Reynolds decomposition of $CO_2$ flux terms (Fig. 8) clearly shows that a large proportion of the summer flux is due to the arrival of the strong SW summer monsoon winds. The positive contributions due to $pCO_2$ occur in the usual

upwelling regions, though their contribution in magnitude is relatively muted, and negative in the southern portion of the AS. Cross-terms, while non-zero, are inconsequential in determining the overall anomaly in summer flux intensity, as has been seen elsewhere (Doney et al., 2009b). Indeed, in a scenario where the cross-term contribution is at its maximum amplitude, the Omani upwelling region during summer, the cross-term is not stroung enough to sway the direction of the flux anomaly.

The summer flux signal is such that in nearly all the regions outside of summer, the anomaly is negative. Furthermore, the contribution of winds in particular is so strong, it is the largest factor all year except for the spring intermonsoon, where peak $pCO_2$ is important relative to the effects of wind (or lack thereof) in the central oligotrophic AS, Somalia, and the averaged domain. This suggests that, on first order, winds are the most important factor in determining the seasonal air-sea flux cycle in the AS. We should keep in mind, however, that these results conflict with the analysis of Roobaert et al. (2019). In their global study of coastal waters, while seasonal $CO_2$ flux variability in the AS is relatively high compared to other regions (their Figure 6), the largest contributions come from $\Delta pCO_2$ and cross-terms (their Figure 7), especially near the Horn of Africa. As a result, further work should be conducted to reduce uncertainty in sea surface $pCO_2$ values to determine whether winds, $\Delta pCO_2$, or cross-terms are significant drivers of air-sea flux. Additionally, when considering the inconsistencies of models in estimating air-sea $CO_2$ flux (Sarma et al., 2013), uncertainties from incomplete representation of winds and the various parameterizations of piston velocity must be considered in addition to $pCO_2$, especially in light of recent work in the field (Ho et al., 2006; Wanninkhof, 2014; Roobaert et al., 2018).

Wind parameterizations notwithstanding, once winds are controlled in our metanalysis (Fig. 9) it appears that on balance: 1) gridded data-based $pCO_2$ products will under-estimate the upwelling zone maxima of $pCO_2$ and $CO_2$ flux during the summer, 2) the model over-estimates $pCO_2$ the rest of the year, eventually contributing to a possible over-estimate of $CO_2$ flux, and 3) this leaves reality somewhere in between. The only way to rectify these differences and arrive at a more accurate estimate will be to conduct sufficient *in situ* sampling of DIC, TA, and $pCO_2$ in more regions than the upwelling zones, and preferably outside of the summer and over the course of multiple years. With the advent of ARGO floats with pH sensors, and the advancement of technology for other variables such as TA, the possibility emerges of using autonomous sampling platforms to expand beyond the limitations of ship-board measurements to fill the data gap in the AS carbon system.

### 4.3.2   $pCO_2$ seasonality

Decomposition of seasonal $pCO_2$ anomalies within regions portrays a slightly different picture where temperature is the dominant force, with DIC countervailing in the upwelling regions. Not only is this seasonal cycle more akin to that seen in the California Current (Turi et al., 2014), the dueling role of these two forces is also reflected in a similar analysis by Sreeush et al. (2019a) for pH instead of $pCO_2$ in the AS. Interestingly, in that study both ROMS and OTTM were compared side-by-side, and in OTTM, TA played a larger role than in ROMS. Similarly, in Valsala and Maksyutov (2013), TA played an important role in regulating inter-annual $pCO_2$ variability in the AS. A preliminary TA budget of the model (Fig. S12) shows that while vertical circulation and biological processes dominate the seasonal cycle of near-surface DIC, TA has multiple forces influencing its time evolution. However, the magnitude of the fluxes are $\sim \frac{1}{5}$ those of DIC, indicating that TA is less seasonally variable than

DIC (reflected also in Fig. 11). These results, from another model as well as the low variability in this model's TA, raise the possibility that TA's importance is under-estimated in the current study.

Zooming out from the upwelling regions and looking at the whole AS, the dominance of temperature on the seasonal $pCO_2$ cycle is clear. In the domain average, temperature effects nearly overlap with the overall $pCO_2$ anomaly. This result brings back into focus the seasonal timing of $pCO_2$ minima/maxima in the model vis à vis previous work. In the earlier studies, which either use data directly or build statistical models from those data, there is no spring intermonsoon $pCO_2$ maximum driven by heating. Indeed, Sabine et al. (2000) noted that $pCO_2$ in the spring was much lower than would be expected given the SST, but attributed this to drawdown due to biological production. The model, however, indicates that this is precisely the season where biological production is at its lowest. The presence of these springtime maxima can be seen in other models, visible in the results of Valsala and Maksyutov (2010) and a synthesis by Sarma et al. (2013). Since the model indicates temperature is producing the maxima, it reduces the concern that erroneous DIC or TA values in the model are driving this signal. The model SST matches well with the *in situ* data (Fig. S8-S9), and the forcing datasets for SST and heat flux correspond to data that predate or include the $pCO_2$ sampling period (i.e. before 2000), so a climate change bias is unlikely. What might be more likely, then, is a sampling bias towards summertime Oman, one of the few areas in the AS with a summertime instead of springtime $pCO_2$ max. Such a bias could possibly obscure what is happening in the rest of the AS. Regardless, the discrepancy between models and observations during the spring period can be added as yet another reason to conduct more *in situ* sampling to either confirm or disavow whether the model results are spurious.

### 4.3.3  DIC seasonality

The potential for biological control in setting $pCO_2$ has been found in Sri Lanka near the AS (Chakraborty et al., 2018). In this study, it was found that the source water in Sri Lanka was sufficiently low in DIC relative to inorganic nutrients that upwelling actually reduced surface $pCO_2$. In a similar vein, Takahashi et al. (2002) found, using a metric comparing temperature and "biological" effects (i.e. everything else), that the AS's $pCO_2$ is reduced more by biological production than temperature effects. Conducting this analysis on the model output (Fig. S13), it appears that "biological" control appears dominant over the upwelling areas (Oman coast, coast of Somalia, India) and near the equator east of $60^o$E, but for the majority of the AS temperature dominates. This cursory analysis aside, as is evident in the results of Chakraborty et al. (2018), the more useful comparison is in determining whether biological production is sufficient to outweigh DIC enhancement from subsurface water.

In summary, the results in Fig. 12 indicate that for the entire AS, DIC enhancement by vertical circulation (both advection and mixing) brings more DIC into the near-surface than is removed by net biological processes, and so no biologically-induced decrease of $pCO_2$ occurs in the final $pCO_2$ signal. The timing of biological drawdown, occurring at the same time or lagging vertical circulation, is consistent with the general phenology of blooms and similar to previous findings (Louanchi et al., 1996; Rixen et al., 2006; Sharada et al., 2008). The result that biological cycling of carbon is much larger than the air-sea flux of $CO_2$ also corroborates the results of Lendt et al. (2003), who found net community production to be $\sim$3.6 times larger than $CO_2$ emission. The relatively low impact of horizontal advection is an interesting detail to consider; in other upwelling systems, significant proportions of water and biological production are advected offshore (Nagai et al., 2015). Lendt et al. (2003) suggest

upwelled nitrate is assimilated and does not arrive in the central AS, while Resplandy et al. (2011) show that a large fraction of total nutrients in the central AS come from the upwelling zones. Thus, although water may be advected offshore, the relevant timescale for DIC cycling processes (i.e. air-sea emission, biological uptake) may be short enough so that horizontal export of enhanced DIC (keep in mind the onshore-offshore normalized DIC gradient) from the upwelling regions does not significantly
contribute to the central AS or other regions.

## 5   Conclusions

In this study, we used a regional circulation model coupled with a biogeochemical model to investigate the annual magnitude, seasonal cycle, and drivers of air-sea $CO_2$ flux in the AS, primarily winds and $\Delta pCO_2$. This effort was made to complement previous flux estimates, where limited data or insufficient model resolution have produced contrasting results. Consistent with
previous work, we find that the AS is a source of $CO_2$ to the atmosphere for the entire year, with the bulk occurring during the summer monsoon. Our estimate of flux, $\sim 160 \text{ TgCyr}^{-1}$, with concentrated flux densities up to $6 \text{ molCm}^{-2}\text{yr}^{-1}$ in the upwelling regions, is larger than most previous reports but not inconsistent with the range of other findings (Sarma, 2003; Naqvi et al., 2005; Sarma et al., 2013) . Since the AS lacks carbon data, here we subjected the model to validation with raw data instead of smoothed climatologies. The model is shown to have a positive bias in $pCO_2$, attributed to TA and DIC, with
TA bias being stronger. Despite this, $pCO_2$ variability compares favorably to alternative products in the region. The bias results in strongly positive $\Delta pCO_2$ throughout the domain year-round. While positive $\Delta pCO_2$ values have been observed before in the AS, we likely over-estimate $CO_2$ flux outside of the summer monsoon.

    The majority of flux occurs during the summer as opposed to a modeled spring $pCO_2$ maximum due to the influence of winds. A Reynolds decomposition of both $pCO_2$ and wind variability shows that the intense winds of the summer monsoon
contribute 90% of that season's flux anomaly. In fact, winds play a more important role than the increase of $pCO_2$ in the upwelling regions. Even though winds represent such a major variable in determining AS $CO_2$ flux *timing*, the variability in *total* flux due to different $pCO_2$ products leads to a 2x range in magnitude. These results suggest that in addition to the expected increase of surface ocean $pCO_2$ due to anthropogenic climate change, possible changes in the timing, location, and magnitude of monsoon winds (Lachkar et al., 2018; Praveen et al., 2020) will have downstream impacts on seasonal air-sea flux.

An important result of this modeling study is that temperature drives a springtime maximum of $pCO_2$ in the AS. This maximum has been observed in lower-resolution models, but is not found in the *in situ* data. Due to the fact that temperature is not sensitive to biological processes like DIC and TA, this discrepancy suggests that more sampling is necessary to determine whether it is an artifact of spotty sampling or an inherent problem in models unrelated to resolving coastal upwelling. Additionally, we find that spatial gradients of DIC and temperature do not overlap as they do elsewhere in the ocean. Instead,
temperature follows a southwest-northeast monsoon wind pattern, whereas DIC is enhanced nearest to the coasts. The resulting apparent deficit of normalized DIC in the central, oligotrophic AS has not been emphasized previously. Finally, we find that despite the intense biological activity in the AS, primary production by phytoplankton is insufficient to counter the increased carbon supply provided by vertical circulation during bloom periods.

Models can be used to expand spatiotemporal coverage when data is scarce. However, models' limitations often manifest when there is no new data to test their fidelity. Limitations in the spatiotemporal coverage of existing datasets stem from biases in sampling during summer monsoon, sampling close to the Oman upwelling region, and limited in scope to the years of JGOFS expeditions of the 1990s. In order to fully characterize the $pCO_2$ cycle outside of summer in the rest of the AS, as well as to determine the secular trend of surface $pCO_2$ due to anthropogenic carbon additions to the atmosphere, more *in situ* data of the carbon system (e.g. DIC, TA, $pCO_2$), from shipboard measurements or autonomous sampling platforms, are sorely needed. Finally since $\Delta pCO_2$ is generally positive in the AS, the direction of air-sea $CO_2$ exchange examined here is robust to model error, whereas other important indicators such as pH and aragonite saturation, $\Omega_a$, which at important thresholds of low values have deleterious impacts for various biological taxa (Doney et al., 2009a; Bednaršek et al., 2019, 2021) will be less so. These data are thus critical for resolving the possible responses of the carbon system in the AS to ongoing climate change, whether from changes in timing or magnitude of monsoon wind forcing, the impact of increased surface heating on stratification and vertical circulation, or changing levels of primary and fisheries productivity with altered carbonate solubility. Without this baseline information, it will be difficult to predict what the future has in store for the AS carbon system.

*Code availability.* ROMS-AGRIF is free to download at https://www.croco-ocean.org/download/roms_agrif-project/

*Author contributions.* A.D., Z.L., and M.L. conceived the study, A.D., Z.L. ran the model, A.D., Z.L., S.S., and M.L. conducted analysis, A.D. generated figures and text, Z.L., S.S., and M.L. revised figures and text.

*Competing interests.* The authors declare that they have no competing financial interests.

*Acknowledgements.* Support for this research comes from the Center for Prototype Climate Modeling (CPCM), the New York University Abu Dhabi (NYUAD) Research Institute. Computations were conducted at the High Performance cluster (HPC) at NYUAD, Dalma. We deeply thank both B. Marchand and M. Barwani for their technical support. We are also grateful for the work of two anonymous reviewers and associate editor Dr. Peter Landschützer, who greatly improved the manuscript.

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

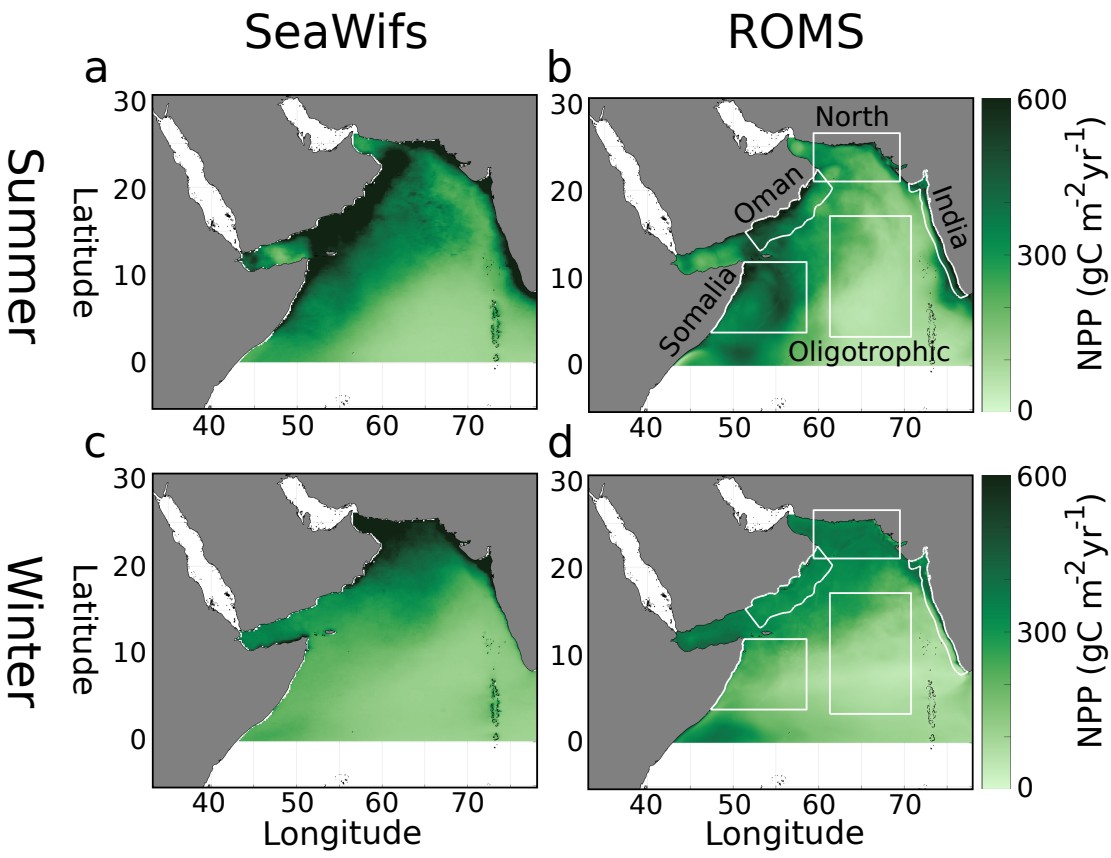

**Figure 1.** Vertically integrated net primary production in the Arabian Sea ($\mathrm{gCm^{-2}yr^{-1}}$) from the VGPM algorithm (Behrenfeld and Falkowski, 1997) for SeaWifs data (years 1997-2010) (a,c) and model output (b,d) for summer (JJAS, top) and winter (DJFM, bottom) monsoons. White boxes in (b,d) denote regions of analysis in the paper.

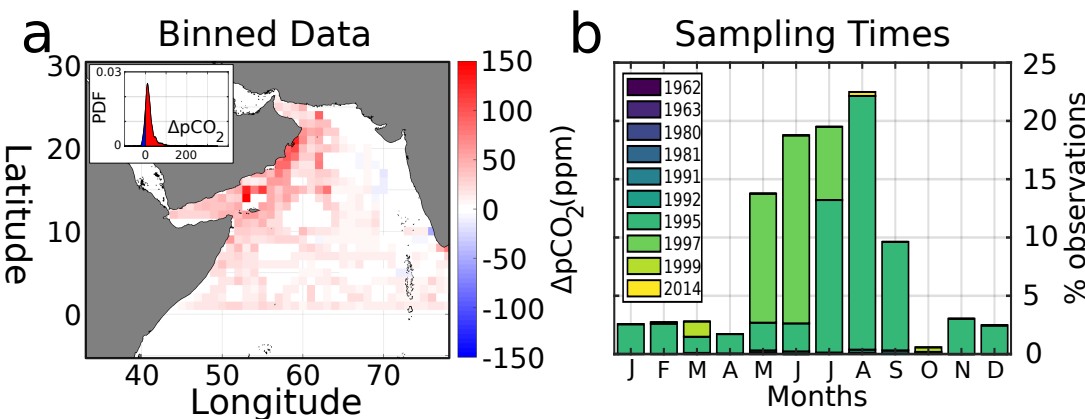

**Figure 2.** (a) Average surface *in situ* $\Delta pCO_2$ (ppm), with probability density function of all $\Delta pCO_2$ values inset. $\Delta pCO_2$ data are calculated in comparison to Keeling atmospheric $pCO_2$, then binned into a $1^o$x$1^o$ grid. (b) Monthly distribution of *in situ* data sampling times, color-coded by sampling year.

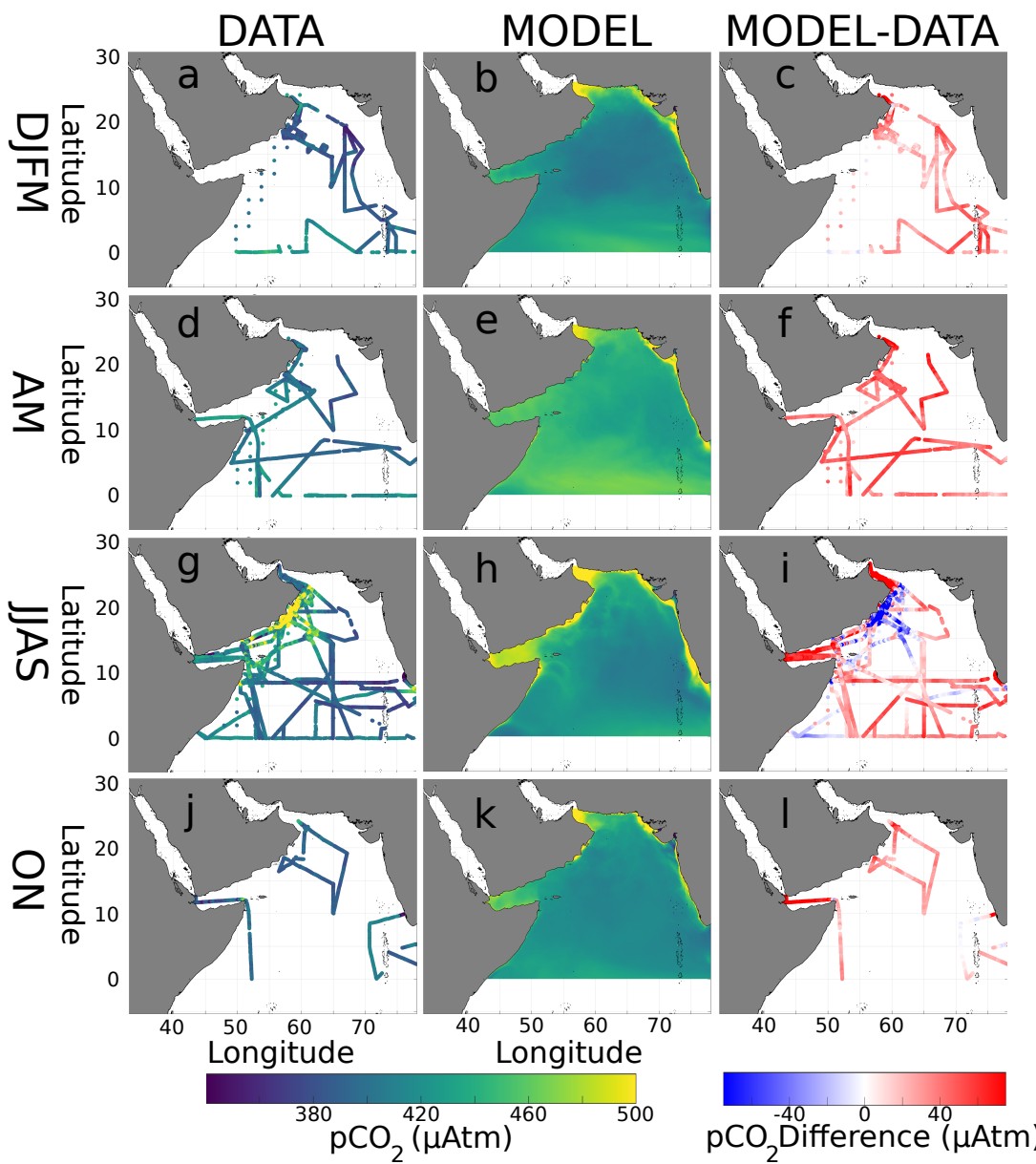

**Figure 3.** Seasonal surface pCO$_2$ (µatm) from data (left column, a,d,g,j) and the model (middle, b,e,h,k), as well as their differences (right, c,f,i,l). Plots are arranged by season: winter monsoon DJFM (a-c), spring intermonsoon AM (d-f), summer monsoon JJAS (g-i), and fall intermonsoon ON (j-l).

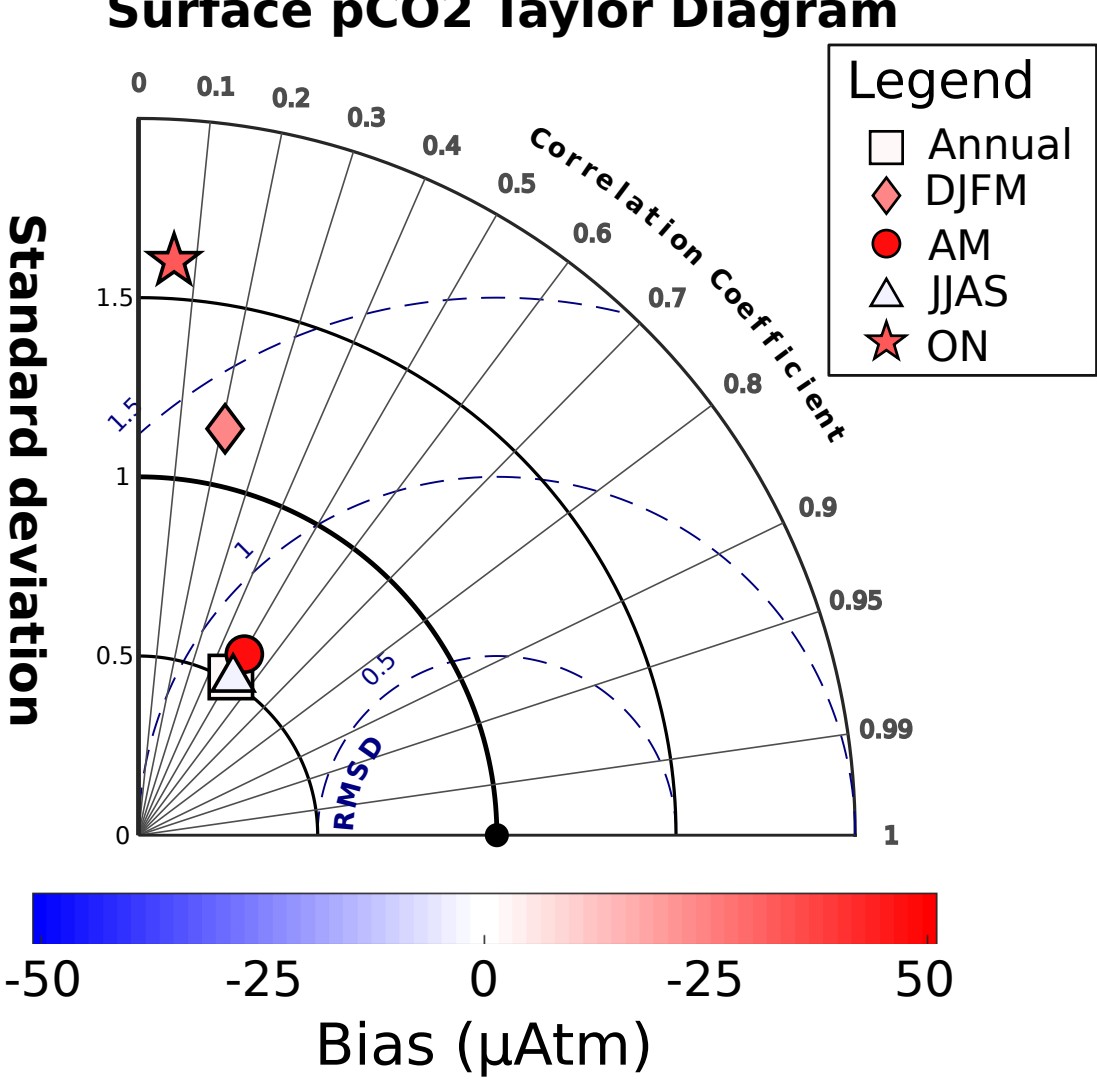

**Figure 4.** Taylor diagram of modeled vs. observed surface pCO$_2$, both annually and seasonally. Data are from merged SOCAT and LDEO databases, corrected to year 2005. Distance from origin (concentric solid lines) is normalized model standard deviation. Angle from vertical axis is Pearson correlation coefficient. Distance from observation point (black dot) is root-mean square deviation (blue dashed lines). Color of each point denotes model bias, *i.e.* positive values are overestimates.

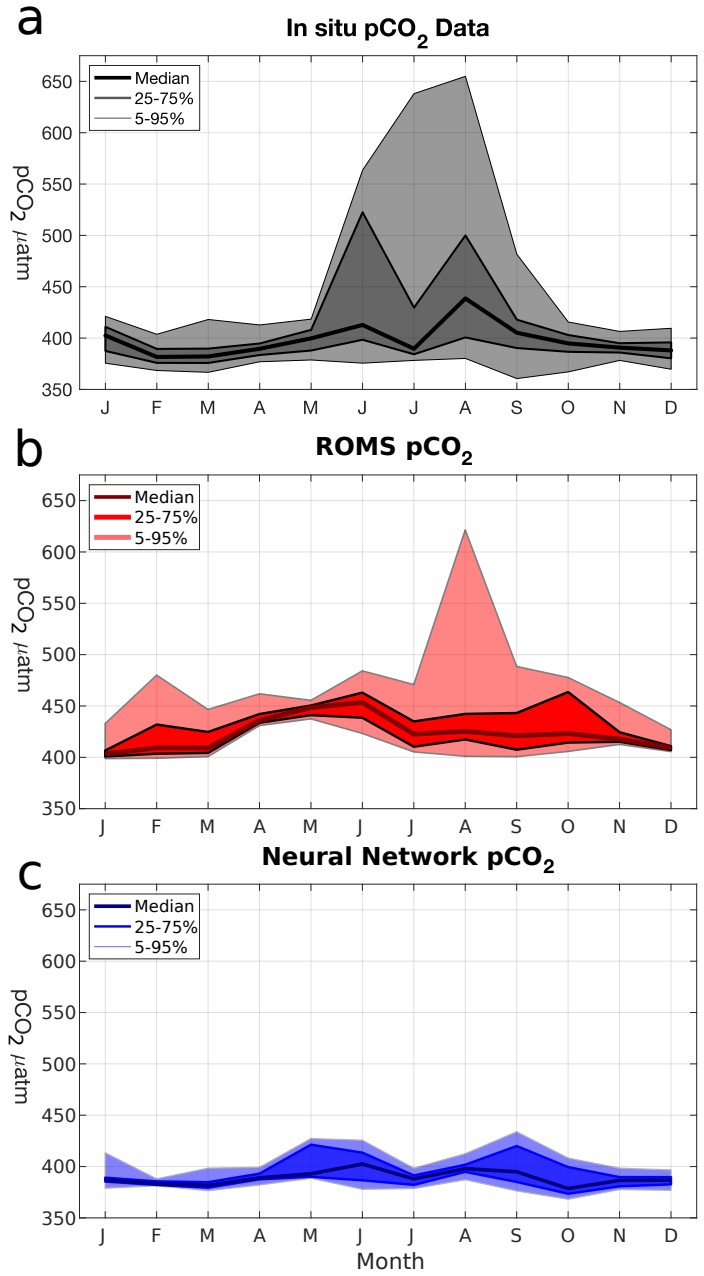

**Figure 5.** Monthly probability density distributions of surface pCO₂ (μatm) in (a) merged SOCAT/LDEO *in situ* data, (b) modeled pCO₂, and (c) L15 pCO₂ climatology.

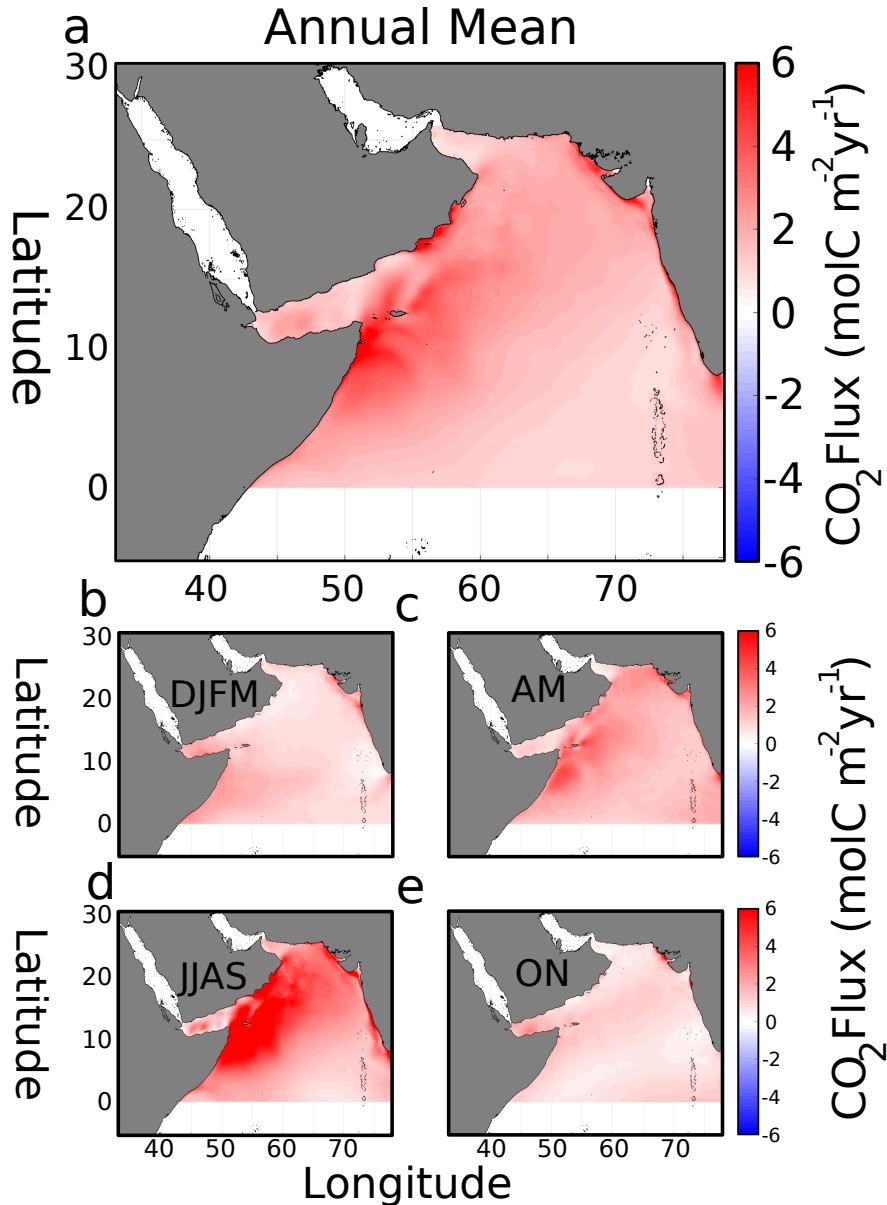

**Figure 6.** (a) Modeled annual mean air-sea $CO_2$ flux density $(molCm^{-2}yr^{-1})$. (b-e) Seasonal flux density for winter DJFM, spring AM, summer JJAS, and fall ON, respectively. Positive is flux out of the ocean.

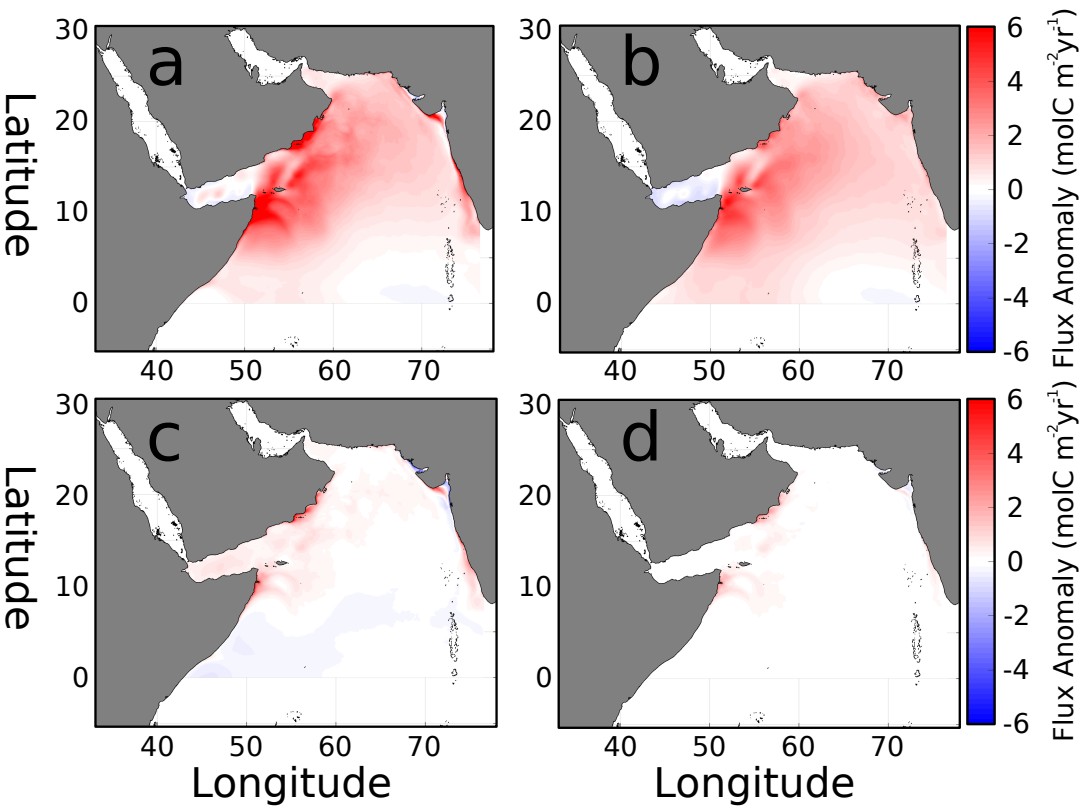

**Figure 7.** (a) Anomaly of air-sea $CO_2$ flux during summer monsoon JJAS ($molCm^{-2}yr^{-1}$). Summer flux anomaly contributions due to (b) wind, (c) $pCO_2$, and (d) cross-terms in Eqn.(2).

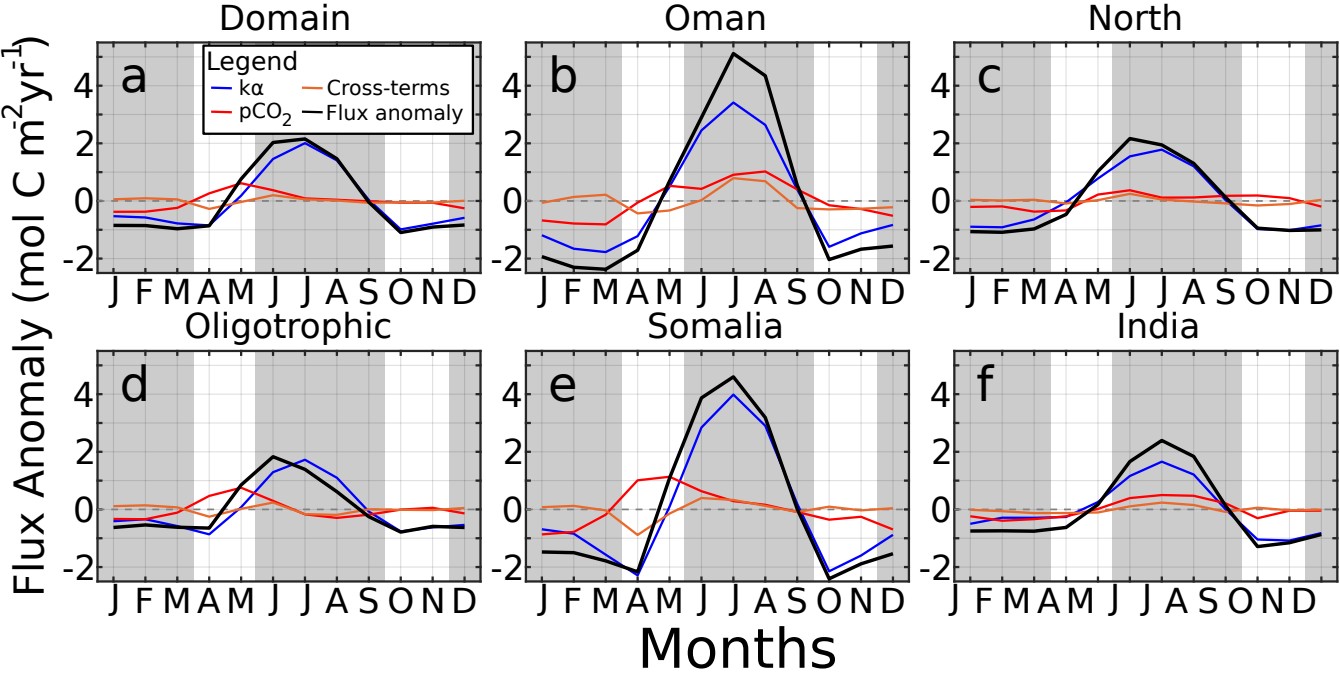

**Figure 8.** Monthly $CO_2$ air-sea flux anomaly $(molCm^{-2}yr^{-1})$ for (a) the domain, (b) Oman, (c) North AS, (d) Oligotrophic central AS, (e) Somali coast, and (f) Indian coast. Contributors to the flux are solubility/winds ($k\alpha$,blue), $pCO_2$ (red), and cross-terms (orange). Gray regions indicate winter and summer monsoons.

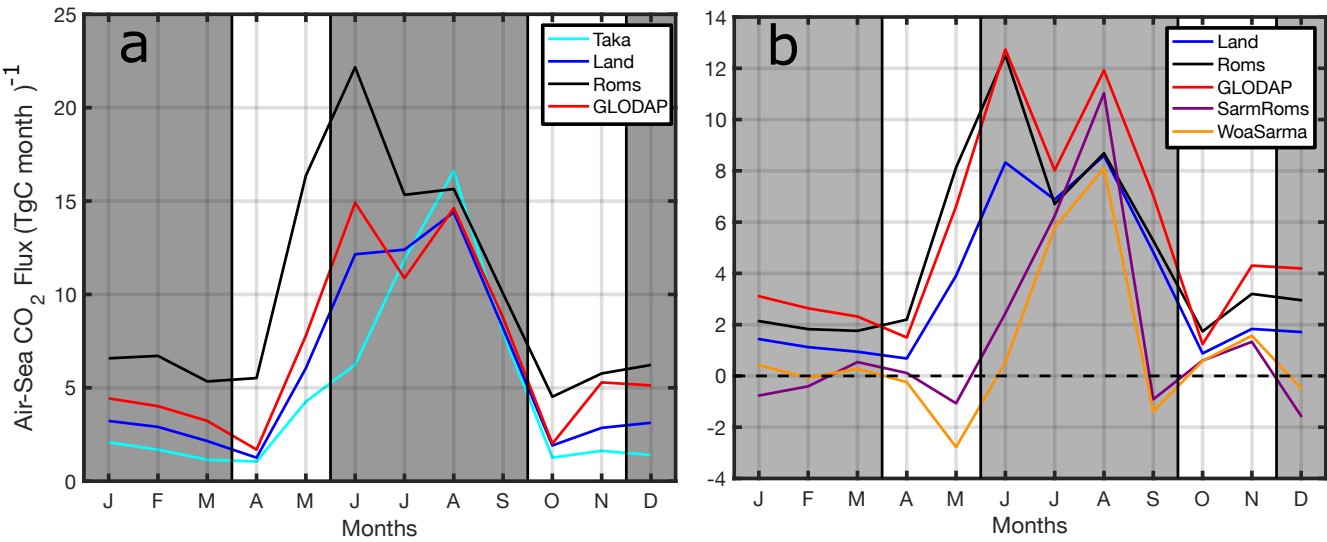

**Figure 9.** (a) Monthy $CO_2$ flux ($\mathrm{TgCmonth^{-1}}$) from the AS as calculated using $pCO_2$ from TK09 (cyan), L15 (blue), model (black), and GLODAP (red). (b) Monthly $CO_2$ flux from $10^o$N and north using $pCO_2$ from L15 (blue), model (black), GLODAP (red), Sarma using model output (purple), and Sarma using WOA data (orange). Dashed line in (b) is the zero flux axis, gray regions denote winter and summer monsoons. Positive flux is out from the ocean surface.

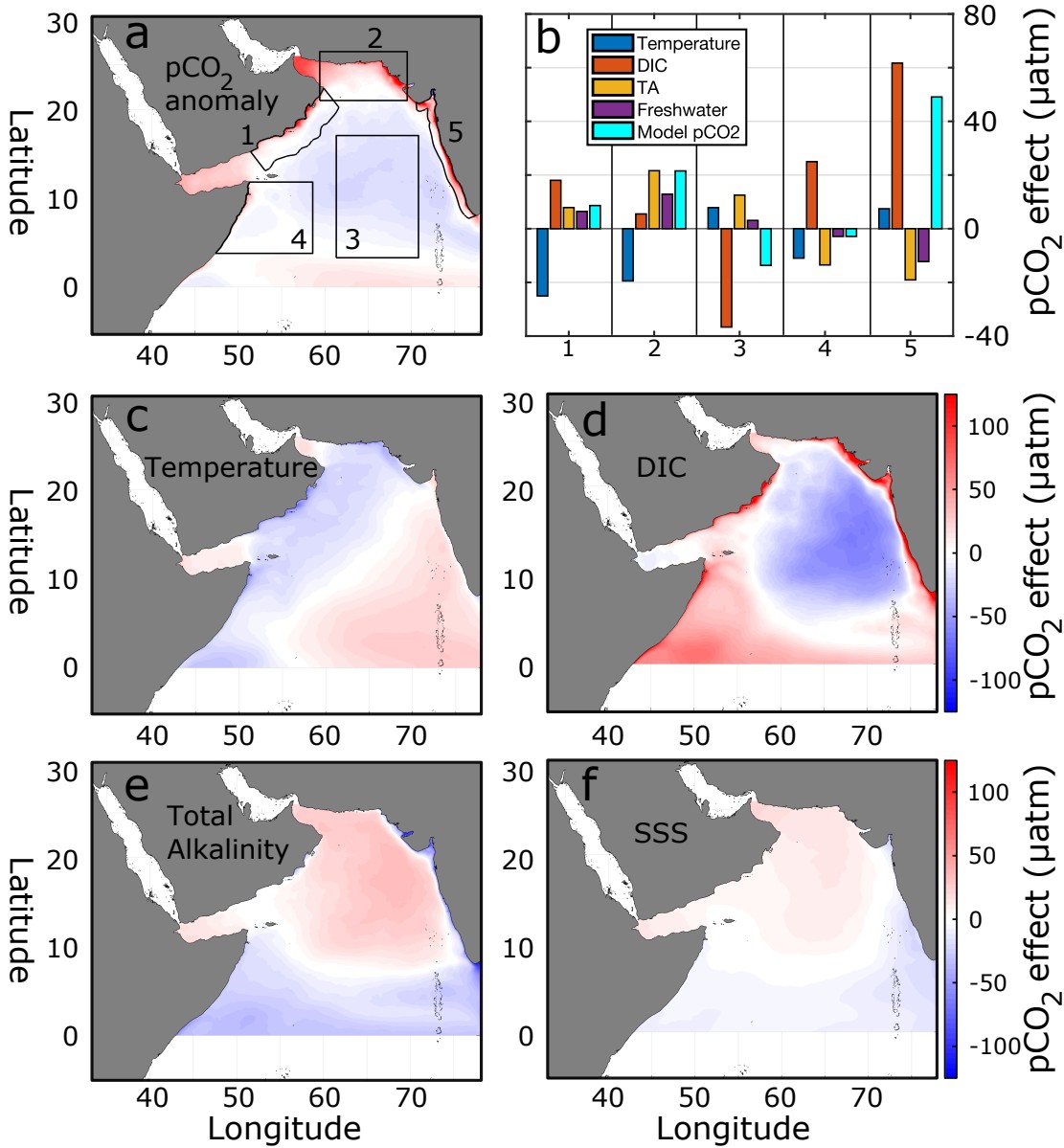

**Figure 10.** (a) Spatial anomaly of time-averaged surface pCO₂ (μatm). Black boxes represent regions of analysis used in (b) to show averaged contributions of four parameters to pCO₂ variability. The changes in pCO₂ due to these variables are shown for (c) temperature, (d) DIC, (e) TA, and (f) SSS.

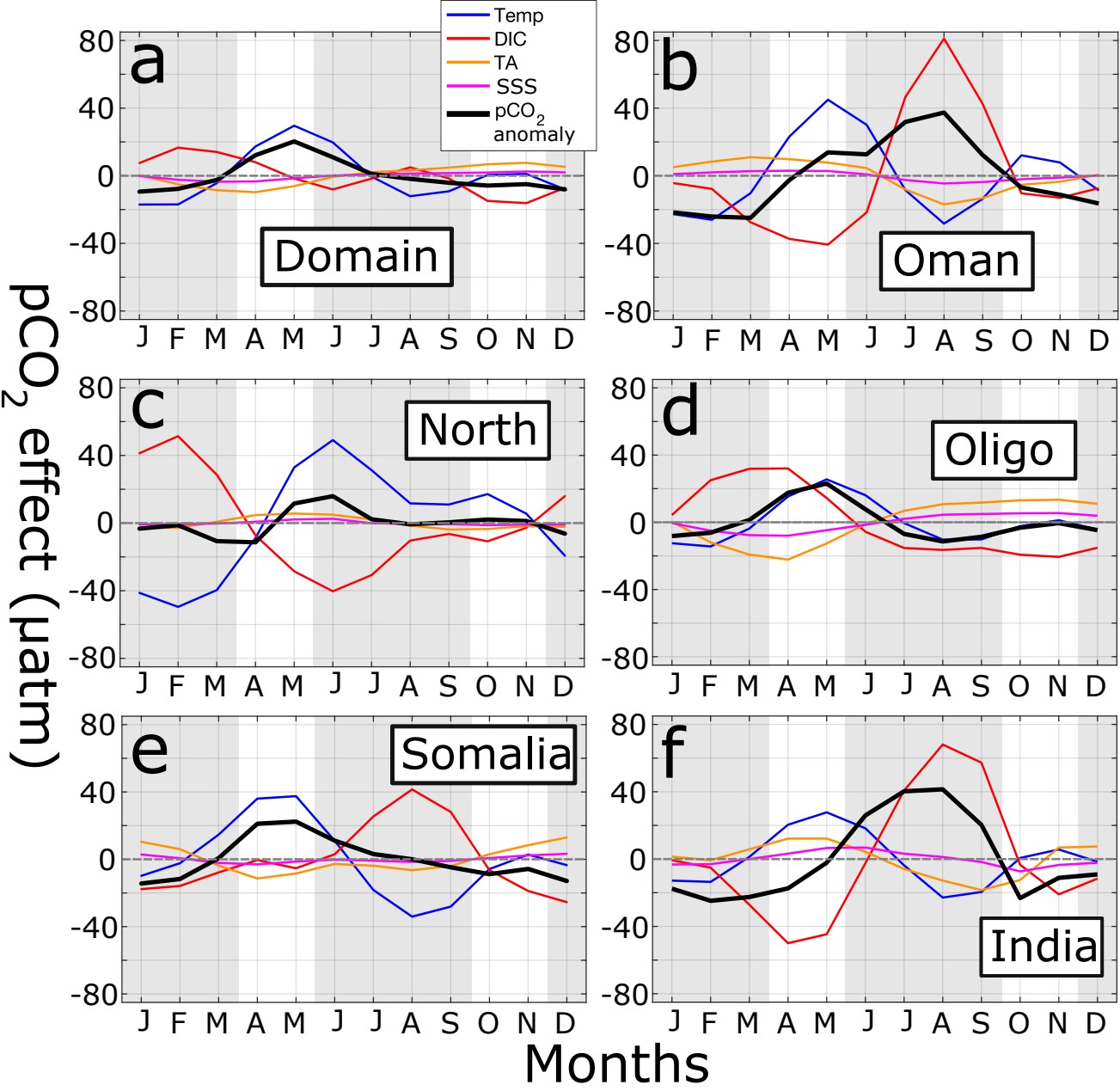

**Figure 11.** Timeseries of pCO₂ anomalies (μatm) (black lines) for (a) the entire domain, (b) Oman, (c) North AS, (d) oligotrophic central AS, (e) Somalia, and (f) India. Dashed gray lines indicates horizontal axis. Gray shading shows summer and winter monsoons. Additional lines show change in pCO₂ due to temperature (blue), DIC (red), TA (orange), and SSS (magenta).

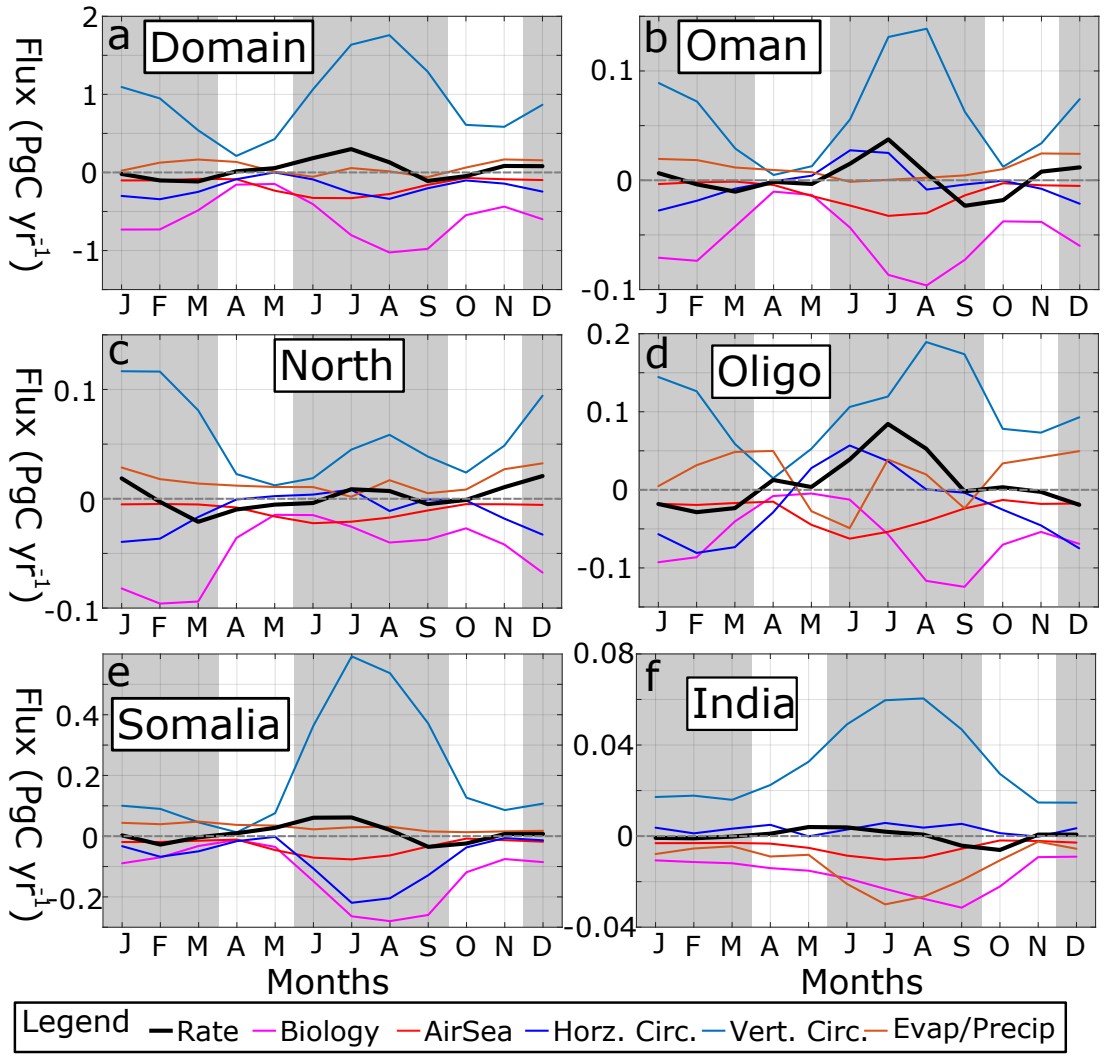

**Figure 12.** Timeseries of DIC fluxes $(\mathrm{PgCyr}^{-1})$ in the top 20 m for (a) the domain, (b) Oman, (c) North AS, (d) oligotrophic central AS, (e) Somalia, and (f) India. Dashed gray line shows x=zero axis. Gray shading denotes summer and winter monsoons.

**Table 1.** Summary of $pCO_2$ datasets used in this study. Included is whether the product is gridded, and if so, its spatial and temporal resolution. Reference year (Ref. Yr) indicates the year from which Keeling atmospheric $xCO_2$ values are used to calculate $CO_2$ flux. Purpose designates use case within the article. $pCO_2$ calculated indicates whether product provides $pCO_2$ (No) or whether $pCO_2$ was calculated using DIC, TA, temperature, salinity, and possibly Chl-$a$ (Yes).

| Dataset | Gridded (Y/N), Resolution | Ref. Year ($xCO_2$) | Domain | Purpose | $pCO_2$ calculated | Reference |
|---|---|---|---|---|---|---|
| Surface Ocean Carbon Atlas (SO-CAT) | No, N/A | 2005 | Global | Model $pCO_2$ validation | No | Bakker et al. (2016) |
| Lamont-Doherty Earth Observatory $pCO_2$ database (LDEO) | No, N/A | 2005 | Global | Model $pCO_2$ validation | No | Takahashi et al. (2019) |
| Takahashi 2009 (TK09) | Yes, $4^o$x$5^o$, monthly | 2005 | Global | Air-Sea $CO_2$ flux estimate | No | Takahashi et al. (2009) |
| Landschützer 2015 (L15) | Yes, $1^o$x$1^o$, monthly | 2001 | Global | $pCO_2$ comparison and Air-Sea $CO_2$ flux estimate | No | Landschützer et al. (2015) |
| Sarma statistical model, T/S/Chl-$a$ from model (ROMS) | Yes, $1/24^o$ (interpolated to $1^o$), seasonal | 1995 | AS north of 10°N | Air-Sea $CO_2$ flux estimate | Yes | Sarma (2003) |
| Sarma statistical model, World Ocean Atlas T/S, SeaWifs Chl-$a$ | Yes, $1^o$x$1^o$, seasonal | 1995 | AS north of 10°N | Air-Sea $CO_2$ flux estimate | Yes | Sarma (2003) |
| GLODAP DIC/TA, World Ocean Atlas T/S | Yes, $1^o$x$1^o$, annual | 2002 | Global | Air-Sea $CO_2$ flux estimate | Yes | Olsen et al. (2019) |

**Table 2.** List of parameters and their values used in the biogeochemical model.

| Parameter | Value |
|---|---|
| $K_w$, seawater light attenuation | $0.04 \text{m}^{-1}$ |
| $K_{Chl}$, Chl-$a$ light attenuation | $0.024 \text{m}^{-1}$ |
| $P_{alpha}$, initial slope of P-I curve | $1.0 \text{ Wm}^{-2}\text{d}^{-1}$ |
| C:$N_P$, carbon-to-nitrogen ratio of phytoplankton | $6.625 \text{ molCmolN}^{-1}$ |
| C:$N_Z$, carbon-to-nitrogen ratio of zooplankton | $6.625 \text{ molCmolN}^{-1}$ |
| $O_2$:$NO_3$, oxygen-to-nitrogen ratio for nitrate uptake | $9.375 \text{ molO}_2\text{molNO}_3^{-1}$ |
| $O_2$:$NH_4$, oxygen-to-nitrogen ratio for ammonium uptake | $7.375 \text{ molO}_2\text{molNO}_3^{-1}$ |
| N:$C_{den}$, nitrate-to-DIC ratio for denitrification | $0.8 \text{ molNO}_3\text{molDIC}^{-1}$ |
| $O_{2den}$, oxygen threshold for denitrification | $4.0 \text{ mmolO}_2\text{m}^{-3}$ |
| $R_CaCO3$, ratio of calcium carbonate precipitation to production | $0.07 \text{ molCaCO}_3\text{molC}^{-1}$ |
| $\Theta_m$, maximum Chl-$a$ to Carbon ratio | $1.3538 \text{ mgChlamgC}^{-1}$ |
| $K_{NO3}$, half-saturation rate for nitrate uptake | $0.75 \text{ mmolNm}^{-3}$ |
| $K_{NH4}$, half-saturation rate for ammonium uptake | $0.5 \text{ mmolNm}^{-3}$ |
| $\mu_{nitr}$, nitrification rate | $0.05 \text{ d}^{-1}$ |
| $\mu_P$, phytoplankton mortality rate | $0.072 \text{ d}^{-1}$ |
| $G_{max}$, maximum zooplankton growth rate | $0.6 \text{ d}^{-1}$ |
| $\beta$, zooplankton assimilation efficiency | $0.75$ |
| $K_{phy}$, half-saturation rate for zooplankton ingestion | $1.0 \text{ d}^{-1}$ |
| $\mu_{exc}$, zooplankton excretion rate | $0.1 \text{ d}^{-1}$ |
| $\mu_{Zmor}$, zooplankton mortality rate | $0.025 \text{ d}^{-1}$ |
| $Z_{gam}$, fraction of sloppy feeding to fecal pellets | $0.33$ |
| $\mu_{SD}$, small detritus breakdown rate to ammonium | $0.03 \text{ d}^{-1}$ |
| $\mu_{agg}$, specific aggregation rate of small detritus and phytoplankton | $0.005 \text{ mmolN}^{-1}\text{d}^{-1}$ |
| $\mu_{LD}$, large detritus breakdown | $0.01 \text{ d}^{-1}$ |
| $T_{dissol}$, water column dissolution rate of calcium carbonate | $0.0057 \text{ d}^{-1}$ |
| $T_{sedremin}$, remineralization rate in sediments | $0.003 \text{ d}^{-1}$ |
| $w_P$, phytoplankton sinking velocity | $0.5 \text{ md}^{-1}$ |
| $w_{SD}$, small detritus sinking velocity | $1.0 \text{ md}^{-1}$ |
| $w_{LD}$, large detritus sinking velocity | $10.0 \text{ md}^{-1}$ |
| $w_{CaCO3}$, vertical sinking speed of calcium carbonate | $20 \text{ md}^{-1}$ |

**Table 3.** Mean and standard deviation (in parentheses) of annual and seasonal surface pCO$_2$ ($\mu atm$) in both the merged dataset and model.

|  | Data | Model |
|---|---|---|
| **Annual** | 426 (68) | 428 (32) |
| **Winter (DJFM)** | 389 (14) | 418 (30) |
| **Spring (AM)** | 398 (13) | 439 (26) |
| **Summer (JJAS)** | 439 (77) | 433 (36) |
| **Fall (ON)** | 393 (12) | 427 (27) |