# Peer review of "Evaluating the Arabian Sea as a regional source of atmospheric CO2: seasonal variability and drivers"

_Biogeosciences, 2021_

## Referee Comment (RC1)

Review of the "Evaluating the Arabian Sea as a regional source of atmospheric CO2: seasonal variability and drivers" by Alain de Verneil1, Zouhair Lachkar1, Shafer Smith2, and Marina Lévy3

This study analyzes the spatial and seasonal distribution of the air-sea CO2 exchange in the Arabian sea. This study is relevant and original both because, on the one hand, it is carried out in the Arabian Sea; region poor in data and poorly known. On the other hand, this study covers a large panel of aspects both in space and time. They first validated their model to observational pCO2 data and climatologies in addition to other variables such as SST, SSS DIC. The spatio-temporal variability of the CO2 flux is then analyzed for the entire domain of the AS as well as for different subregions with a quantification of the wind speed and pCO2 contribution. A further details analysis of the pCO2 spatio-temporal variability is also performed with a finer decomposition into processes such as temperature, circulation (upwelling effect), biology, …

However, this study presents a lot of information that goes a bit in all directions with a lot of sentences without justifications behind their strategies (e.g., line 106 value of 2 µatm) and a lack of quantitative values to support the text in the results and discussions sections.

I recommend this publication in "Biogeosciences" after major revision.

**Lines 25-31:** clearly need quantitative values to support the text.

**Lines 25-26**: "*and up to the present has on average acted to uptake excess anthropogenic CO2 (Ciais et al., 2013; Khatiwala et al., 2009)*" Should add a value + references need to be updated (e.g., GCP)

**Line 27:** "*The Arabian Sea (AS) is a region of the ocean that has been found to naturally release CO2 to the atmosphere (Sarma et al., 1998), mitigating the ocean's role in moderating atmospheric CO2 accumulation*." Reference 1998 is very old. What about e.g., Cao et al 2020? What is the value of the AS source of Sarma? Without a value, it is difficult to evaluate the contribution of the AS.

**Line 65:** "*The need is further emphasized when quantities such as pCO2 can be utilized as a proxy for other things, such as community compensation depth (Sreeush et al., 2019b)."* Not clear, need to be developed if you mention that in the text.

**Line 66**: "*However, most recent studies…*" Should add references. And most recent "modelled" studies or "observational" studies? Not clear

**Line 71-72**: "*Sarma et al. (2013) found that coupled ocean biogeochemical models underestimated the air-sea CO2 flux.*" should add "in the AS."

**line 75:** "*peak of flux*". Which flux? Air-sea co2 exchange? Advecting carbon flux? Not clear

**Line 83**: should say "air-sea CO2 flux."

**Line 84:** "*the quantification of the air-sea flux will focus on the contributing factors of pco2 …*". Spatial contribution or seasonal contribution? Not clear

**Line 84:** should say "deltapco2 and wind" instead of pco2 and wind. What about the contribution of atmospheric pco2?

**Line 84:** should say "sea surface temperature". Same for salinity

**Line 85:** "*which of course also varies from region to region within the AS*". Should say "for the entire domain of the AS as well as its spatial heterogeneity within the AS".

**Line 87:** "*biological production/respiration*". What about calcification remineralization or Nitrification denitrification?

**Line 91:** "*resolved by the data*" Not clear which data? from which study?

**line 91** In the beginning of the introduction, you refer that data in the AS are very old (20 years ago) and here you refer to study from 2015. Not clear

**Line 92**: « The rest of the paper » should use another sentence/to be reformulate.

**2.1**

**Question:** How many data in SOCAT and LDEO database? Monthly? Daily? annual values? what period of study? What is the spatial resolution? Do you include the coastal datasets?

**Question:** paragraph from line 110: Why in addition to the Socat and LDEO observation data, do you also add the climatology of tk09 and L15? what more do they bring? And why you used the climatology of TK09 since others and more recent are available (see e.g., Rodenbeck et al., 2015 which compare different pco2 climatologies). I do not understand why you choose TK09?

**Question:** Is the coastal domain included in your definition of the AS domain? if yes (e.g., line 157 you use the word coast), the pco2 climatology of Laruelle et al 2017 and Roobaert et al (2019) for the coastal flux should be considered here? You refer to TK09 and L15 but what about the coastal regions?

**Comment:** In the text, a clarification is need on the period in which you are going to work + spatial resolution + temporal and how you have treated the different datasets to switch, for example, to seasonal. Why are some databases used only for model validation while others for the flux estimation (table 1)?

**Comment**: This manuscript needs a paragraph about the spatial and temporal pco2 and FCO2 uncertainties.

**Comment**: I suggest moving section 2.4.3 before 2.4.1

**Line 98:** "*pco2*" should specify that it is sea surface pco2

**Line 100:** "*model pCO2 is calculated from DIC and TA*" how?

**Line 101:** "*in situ un-gridded data merged from SOCAT v. 2019*" why you do not discuss about this dataset in the Introduction. In the introduction, you said that there is in the AS only observational data from 20 years ago (JGOFS) and here you discuss about LDEO and SOCAT. Not clear

**Line 103:** "*SOCAT data was preferred to LDEO observations*" why?

**Line 103:** "*large overlap*" how many data? you should be more quantitative and more precise in the words used.

**Line 105:** "*… using reported sea surface temperature (SST) and S data included in the products*" S= sea surface salinity? which equation do you use to convert fco2 to pCO2? Not clear how fugacity and partial pressure are related to SST and S

**Line 106**: why a value of 2 µatm yr-1. Should be justified.

**Line 109**: "*The gridded products*" monthly?daily?annual grid? For which period?

**Line 115**: Which products/datasets do you used to calculate the air-sea flux?

**2.2**

**Line 121:** "*System-AGRIF (ROMS-AGRIF) 3.1.1.*" version 3.1.1? what means AGRIF?

**Line122:** "*Previously used in the region (Lachkar et al., 2016), the…*" should be **by** Lachkar et al.,

**Line 125:** "*advection of tracers*" in repeat twice in the same sentence

**Line 125:** what is the K-profile?

**Line 126:** what is KPP?

**Line 127:** 33° W or E?

**Line 127:** which previous studies? Need references.

**Line 127:** sometimes you speak at present sometime in future → same comment for all text.

**Line 130:** what is the name of the OBGCM model?

**Line 131:** "*Biological parameters for the model are the same as those used in Gruber et al. (2011)*." That is to say? Which parameters? Should be enumerated.

**Line 136**: what about nitrification and denitrification?

**Line 144:** "*climatological mode*". Not clear, what is the period?

**Line 145**: "*surface salinity*". In the text before it was S. Not clear

**Line 146:** SST is from 1985-1997 but then you make a FCO2 comparison for another period. Still confused about the period of this study. What is the spatial resolution of SST?

**Line 147:** Why do you need wind stress? The period of the wind stress is 1999-2009, different than SST. Not clear again.

**Line 148**: "*T, S*" same comment than before. SST or T .. not clear

**2.3**

**Line 157-158:** need reference from the literature to support summer monsoon and enhanced biology.

**Line 162:** same comment need reference for winter monsoon NPP

**Figure 1:** what is the period of the study?

**2.4**

**2.4.1**

**Line 168:** "*The proximate variables impacting pCO2 in the*" should be "*..that affect pco2 change in the*"

**Line 170:** What is the depth of the water column where the pco2 decomposition into processes is performed in the model?

**Line 171**: In equation 1, terms on the RHS are not described in the text and not clear when you said "*deviation from an average.*"

**Line 181:** how partial derivative are calculated? Which coefficient are used?

**2.4.2**

**Line 197-198:** what about the river flux? Did you include the contributions from rivers?

**Line 202**: Eq 6 what are "New" and "Reg" and why CaCO3precip.remin has the same sign than PPnew+reg (both with a -)?

**2.4.3**

**Line 226**: "*five-year average*" why five? Not clear

**3.1**

==Question==: Why did you analyze the variables in depth in Fig. S1 and Fig. S2 and not just the spatial distribution at the sea surface as it is performed for Fig. S3? What is the reason for the choice to do in depth?

==Question==: I do not understand between figure 2 and figure 3 for the data spatial distribution. Are they the same data between both figures? It seems there is more surface covered by the data on figure 2 than when they are divided into seasons (figure 3)?

==Comment==: Lots of imprecision on the form and problems with the numbering of the panels.
e.g., Figure 3. « *Seasonal surface pCO2 (µatm) from data (a-d)* » panels are not ok between the figures and the caption.

==Comment==: the pco2 Seasonality has already been analyzed by the previous figures and compared to the SOCAT database spatially and temporally with a Taylor diagram. Why do a comparison to LA15 again? What more does this bring? not clear in the text

==Comment==: I do not understand the added value of the Taylor diagram. What more does this figure add? moreover, you do not explain the various terms of the taylor diagram such as the Pearson coefficient. Why is it not done on the other variables, especially when you go directly to line 255: Since the model successfully replicates other tracers, physics, and biological processes". A little fast, no? In addition, the seasonality of these processes is not analyzed.

==Comment:== You start with fig S1 to S3 discussing variables then you discuss about pco2 and then you discuss again with variables (line 255). Not clear. Should be first pco2 then variable or vice versa

**Line 230** "*The implementation of ROMS-AGRIF presented here has been used in previous studies (Lachkar et al., 2016).*" And? What is the conclusion of this sentence? Is it in the AS?

**Line 233**: Add panels with the mismatch (model minus data) in FIG S1 to S3 would be useful to highlight difference between model-data. Same comment for figure 3.

**Fig. S 2.** "*Transects, similar to Fig. S1, but with salinity instead of temeprature*" should be temperature.

**Lines 230-233**: e.g. "*Reproduce well the …*" need to add quantitative values.

**Line 235**: "*Regarding pCO2, in situ data from the merged SOCAT/LDEO database shows that average binned pCO2 values in the region are positive for most of the AS (Fig. 2a). The ensemble of observations show that 90% of _pCO2 observations are positive, indicating positive flux to the atmosphere (Fig. 2a, inset)."* These two sentences should be merged in one.

**Line 239:** "pCO2 value…" we do not know if it is modelled pco2 value or observational pco2 value. Maybe you could use "data-based pco2" and "modeled-based pCO2" everywhere in the text.

**Line 242:** Again, this paragraph needs quantitative values. In the pco2 model description, there is no value in the text. Use values instead of "Is lowest".

**Fig. 4** "*Taylor diagram of modeled vs. observed surface pCO2, both in total and seasonal sub-sampling. Data are from merged SOCAT and LDEO databases, corrected to year 2005. Distance **frmo** origin ..*" should be from

**Line 260**: "*close to reported measurement error*" give a value.
**Line 255:** "*tracers, physics, and biological processes*" which are?
**Line 257:** Why the SST from SOCAT is used here from the socat database while in Fig. S1 it derives from the WOA? Not clear

**Line 258:** "*sst*" sometime sst sometime T. Not clear. Same comment for every variable e.g. Fig S4, SAL instead of S or SSS. Be consistent all text and figures.

**Fig. S4**. same comment SST or temperature… not clear

**Line 267:** Figure 5, I suggest merging the three plots in one and use monthly boxplot for instance for median, 5$^{th}$ and 95th. Easier for the reader.

**Line 277:** "*competing product*" Not clear. what that mean? Observational? Model? Not clear

**3.2**
**3.2.1**

**Comment:** need to be more quantitative.
**Comment:** You should be careful when you use the term "dpco2" for change in pco2 spatial or temporally. You also use dpco2 for the air-sea CO2 gradient. Not clear e.g., Fig. 6 and Fig.2.
**Comment:** maybe specify in the text what corresponds to a positive anomaly and a negative anomaly on pco2

**Line 286:** T, S alk are variable but FW is a process. Maybe salinity instead of FW?

**Line 284:** In fig 6, maybe add numbers to regions for each box, then in panel b also add the number instead of name. Easier for the reader to say in panel b where is the regions in panel a

**Line 286:** "*The cold SST structure largely overlaps the stronger summer monsoon winds.*" Not clear

**Line 301:** "*The previous section outlines why certain geographic regions within the AS have overall high or low pCO2 values*" This does not explain why. Just how the variables change within different regions but not in terms of process the section does not explain why. It DESCRIBES the pCO2 changes associated with the variables but not processes; pay attention to the word used. Question: what are the processes that explain the spatial distribution of pco2? Not clear

**3.2.2**

**Comment:** Again, this section needs numbers to support the text. E.g., what is the typical amplitude of the respective seasonal signal?

**Question:** How do you explain that the important contribution of DIC in the upwelling regions is not seen on the seasonality of the entire domain (Fig. 7a)

**Line 302:** "*the decomposition of factors affecting monthly*" should be "variable" instead of "factors"?

**Line 303:** Figure 7. "*Timeseries of pCO2 anomalies (µatm)*" units should be µatm month-1

**Line 304:** "pCO2" should be called "pco2 anomaly"

**Line 305:** "*DIC acts*" DIC is not a process. Should be « change in pCO2 associated to DIC change»

**Line 307:** you first discuss about Fig.7a in the text for the entire domain. Then you discuss about the oligotrophic central region. Hence, Fig.7d should be Fig. 7b. The different panels in figure 7 should be in order of what they are called in the text.

**Lines 320-326** Maybe this section should be in the discussion section instead of the result section.

**Line 328:** "*is controlled by the physical processes of surface forcing, mixing and advection*" not clear what means the term "surface forcing"?

**Line 330:** Figure 8. "*Timeseries of DIC fluxes (PgCyr−1)*" why the DIC flux is in Pg C yr-1? Should be inPg C month-1? Why do you call this variable "DIC flux" instead of "delta DIC" such as "deltapco2" before?

**3.3**

**Comment:** Again, this section needs numbers and more precise.

**Line 330:** "*Budgets of DIC fluxes in the upper 20 m (Fig. 8) show that two major processes dominate, vertical circulation (light blue lines) and net biological processes (green lines).*" I'm not sure that the term "budget" is appropriate here. This figure shows the monthly decomposition of DIC change into processes which is not a budget. Also, it is not clear when you said, "the two major processes dominate". They do not dominate; they show the largest seasonal variations but they not dominate the final seasonal signal.

**Line 330**: Fig. 8 complicated to compare one region over another. All panels should be the same yaxis scale.

**Line 332**: "*acts as a source of DIC*" when? All time? Which month? Should be more precise.

**Line 341:** "*escape is mostly smaller than biological flux*," not clear. Smaller of what? In term of seasonal amplitude? For a specific month? Again, need to be more precise in the text.

**Line 351-357** Maybe this section should be in the discussion section instead of the result section.

**3.4**

==Comment==: Again, this section needs more value to support results than only a qualitative description.

==Comment:== I propose to change the order of the different sections. First, what governs the variability of FCO2, then pCO2. So, I suggest to first start with 3.1, then 3.4 then 3.2 and 3.3.

**Line 370**: what about the contribution of atmospheric pco2?

**Line 372:** Figure 10 "*(a) Anomaly of air-sea CO2 flux during summer monsoon JJAS (molCm−2yr−1. Summer flux anomaly contributions due to (b) wind, (c) pCO2, and (d) cross-terms in Eqn. (8)*". Could say what means a positive/negative anomaly. Also (molCm-2yr-1).

**Line 374**: "*The _pCO2 contribution to*" dpco2 or pco2? Not clear. What about the atmospheric CO2 contribution?

**Line 378:** Figure 11. "*Seasonal CO2 flux anomaly (purple) for winter DJFM monsoon (top-left), springAM(top-right), summer monsoon JJAS (bottomleft), and fall ON (bottom-right). Contributors to the flux are solubility/winds (k_,blue), pCO2 (red), and cross-terms (yellow).*" Results are shown for entire domain and sub-regions." Problem with panel "top left" "top right" in the caption and the figure.

**Line 378:** Figure 11. It is complicated based on this figure to see which process controls the seasonal flux variability. This figure represents the budget for each season. It would be easier to represent the seasonal flux variability into processes using the same strategy than for the seasonal pCO2 anyalsis with seasonal profiles as realized for pco2 (Fig 7 and 8).

**Line 378:** "*The seasonal flux anomalies for all regions throughout the year are displayed in Fig. 11*" should be "aggregated by season" instead of "*throughout* the year".

**Line 393-395**: Maybe this section should be in the discussion section instead of the result section.

==Comment==: Lots of repetition of results in the discussion section. Should be reduced. I also propose to reorganize the discussion section in 1) model vs data, 2) spatial analysis (FCO2 and pCO2) and processes, 3) seasonal analysis (FCO2 and pCO2) and processes and include in 3) the point 4.3

==Question==: according to your results, does the seasonal FCO2 and pCO2 variabilities and processes diverge between the open ocean and the coastal domain in the AS?

**Line 406:** what is the order of magnitude of the bias of Valsala? same order as you? Need quantitative values.

**Line 428:** "*Since upwelling regions are limited in geographic extent near the coast, capturing their high pCO2 values can be difficult for other approaches, such as TK09 with its coarse grid*". Same comment than before. Why did you use the TK09 climatology instead of others finer spatial resolution climatologies. Also, since the upwelling is close to the coast, what about the coastal climatology of Laruelle et al. (2017)?

**4.2**

**Line 434-445**: should be in section 4.1

**Line 434:** "from previous studies" observational or modelled studies? Not clear

**Line 446**: what are the consequences on this seasonal mismatch on your decomposition into processes?

**Line 462:** "*influence of outflows from marginal seas*" how?

**Line 470:** "*the dominance of temperature is clear*" spatially or seasonally?

**Line 471:** Be careful when you compared processes on different time scale (seasonal vs interannual)

**4.3**

**Line 497:** "*and so no biologically-induced decrease of pCO2 occurs*". Maybe add "in the final pCO2 signal".

**4.4**

**4.4.1**

**Line 512:** "*where uptake of CO2 takes*" should be "atmospheric CO2 uptake takes"

**Line 518:** Fig. 9d instead of Fig. 10d

**Line 521:** "*The fact that model CO2 flux peaks in summer despite a wide-ranging spring peak in pCO2*" For the entire region? Which region? Not clear

**Line 535**: need more reference for the use of different k-parametrization and wind speed on the flux calculation (e.g., wanninkhof, Ho, Roobaert, …)

**Line 521-535**: Need more comparison to other studies. e.g, you could compare your results with the study of Roobaert et al. (2019) that decompose the CO2 flux seasonality into processes in the region.

**4.4.2**

==Question==: What is the real purpose of this section 4.4.2? is the objective to calculate an uncertainty on the flux? if so, why only focused on the choice of pco2? why you chose TK09 instead of that of Rodenbeck (2014) which is finer in terms of spatial resolution and more recent. What about the uncertainty associated with the other terms involved in the flux calculation? the wind? its spatial and temporal resolution? the k-parametreization? Etc…

==Question==: This study highlights that the effect of the wind governs the seasonal variability of the flux. Why is it important to do, as it is performed here, a finer pCO2 decomposition into processes?

**Line 540:** "*Considering the important role of winds*" on the seasonality? Spatiality? Not clear

**Line 542**: Fig 12: should be "Tg C month-1" instead of "TgC"

**Line 546**: "*Despite differing pCO2 seasonality*". In this figure only LA15 is referred. Not the other products. So you can not refer as something common between all pco2 products

**Line 560**: «*1) gridded data-based pCO2 products will under-estimate the upwelling zone maxima of pCO2 and CO2 flux during the summer*" how can you conclude that since you said before that your model produces higher pCO2 values compared to other studies. What is the value/study for this conclusion? Not clear

**Line 573:** 160 while in line 550 it is between 57-120. Not clear.

**Line 574**: need references.

---

## Author Comment (AC1)

**Response to Reviewer 1's Comments**

Alain de Verneil     Zouhair Lachkar     Marina Lévy     Shafer Smith

June 16, 2021

We all would like to thank Reviewer 1 for providing their comments in reviewing this manuscript (MS). It is very clear from the detail and thoughtfulness of the comments and questions that a lot of time and effort were spent in the review, which is both fair and will help produce a better paper. Below, we reproduce the reviewer comments (lines preceded with >), with our responses interspersed within along with updated text in the MS shown in italics where necessary. Changes made to the original text show red and blue for deletions and additions, respectively.

>Review of the "Evaluating the Arabian Sea as a regional source of atmospheric CO2:
>seasonal variability and drivers" by Alain de Verneil1, Zouhair Lachkar1, Shafer
>Smith2, and Marina Lévy3

>This study analyzes the spatial and seasonal distribution of the air-sea CO2
>exchange in the Arabian sea. This study is relevant and original both because,
>on the one hand, it is carried out in the Arabian Sea; region poor in data and poorly
>known. On the other hand, this study covers a large panel of aspects both in space
>and time. They first validated their model to observational pCO2 data and
>climatologies in addition to other variables such as SST, SSS DIC. The spatio-
>temporal variability of the CO2 flux is then analyzed for the entire domain of the AS
>as well as for different subregions with a quantification of the wind speed and pCO2
>contribution. A further details analysis of the pCO2 spatio-temporal variability is
>also performed with a finer decomposition into processes such as temperature,
>circulation (upwelling effect), biology, ... However, this study presents a lot of
>information that goes a bit in all directions with a lot of sentences without
>justifications behind their strategies (e.g., line 106 value of 2 matm) and a lack of
>quantitative values to support the text in the results and discussions sections.
>I recommend this publication in "Biogeosciences" after major revision.
>1

>Lines 25-31: clearly need quantitative values to support the text.
>Lines 25-26: "and up to the present has on average acted to uptake excess
>anthropogenic CO2 (Ciais et al., 2013; Khatiwala et al., 2009)" Should add a value
>+ references need to be updated (e.g., GCP)
>Line 27: "The Arabian Sea (AS) is a region of the ocean that has been found to
>naturally release CO2 to the atmosphere (Sarma et al., 1998), mitigating the ocean's
>role in moderating atmospheric CO2 accumulation." Reference 1998 is very old.
>What about e.g., Cao et al 2020? What is the value of the AS source of Sarma?
>Without a value, it is difficult to evaluate the contribution of the AS.

We agree that including quantitative values provides better context to motivate the paper. We've added values to these lines as follows:

*The global ocean represents a major reservoir of inorganic carbon on the planet's surface (40x atmosphere), and up to the present has on average acted to uptake ∼23% of the 11Gt excess anthropogenic carbon $CO_2$ each year (Friedlingstein et al., 2020; Ciais et al., 2013; Khatiwala et al., 2009). The Arabian Sea (AS) is a region of the ocean that has been found to naturally release $CO_2$ to the atmosphere (∼$90MtCyr^{-1}$, Sarma et al., 1998), mitigating the ocean's role in moderating atmospheric $CO_2$ accumulation. While the AS as a regional basin is considered too small to greatly impact global budgets of air-sea $CO_2$ exchange (Naqvi et al., 2005), it attracts attention because some of the highest rates of air-sea $CO_2$ flux 7-33 $molCm^{-2}yr^{-1}$ and values >700 $\mu atm$ of partial pressure of $CO_2$, or $pCO_2$, have been observed there, in addition to unique features such as the world's thickest oxygen minimum zone (OMZ) (Lachkar et al., 2016) and corresponding Carbon Maximum Zone (CMZ) (Paulmier et al., 2011).*

The reference from 1998 is old, indeed. Thank you for bringing Cao et al., 2020, to our attention. It's true that the Arabian Sea is a case study in this paper, but if you look into the methods they use the same 1995 JGOFS data as essentially all other data-driven studies. In addition, since that paper focuses on the nearshore regions ≤100km of the coast (although admittedly a decent amount of the upwelling occurs there), it doesn't address the Arabian Sea as a whole.
* * *
>Line 65: "The need is further emphasized when quantities such as pCO2 can be
>utilized as a proxy for other things, such as community compensation depth
>(Sreeush et al., 2019b)." Not clear, need to be developed if you mention that in
>the text.

Yes, we included this to simply show the importance of better determining $pCO_2$ and air-sea $CO_2$ flux because these very same quantities are being used to estimate more complicated quantities related to an ecological state, compounding the uncertainty. We propose to expand upon it as follows:

*The need to reduce uncertainty is further emphasized when modeled carbon chemistry quantities such as pCO2 can be are utilized as a proxy for other things,. For example, a recent modeling study in the AS found that $pCO_2$ could be used to indicate such as community compensation depth, which reflects the complicated balance between primary production and respiration in the water column (Sreeush et al., 2019b). As a result, the possibility exists to propagate uncertainties beyond carbon chemistry.*
* * *
>Line 66: "However, most recent studies..." Should add references. And most recent
>"modelled" studies or "observational" studies? Not clear

Yes, by "recent studies" we are referring to modeling studies, since observational carbon data is lacking. The citations in mind are the three just cited in the lines above. We've added the following to clarify:

*However, most recentthese AS modeling studies compare...*
* * *
>Line 71-72: "Sarma et al. (2013) found that coupled ocean biogeochemical models
>underestimated the air-sea CO2 flux." should add "in the AS."

This change will be in the revised MS.
* * *
>line 75: "peak of flux". Which flux? Air-sea co2 exchange? Advecting carbon flux?
>Not clear

Yes, "peak of flux" refers to Air-sea $CO_2$ exchange, and will be changed to read:

*Additionally, Sarma et al. (2013) found that  peak air-sea $CO_2$  flux observed in...*
* * *
>Line 83: should say "air-sea CO2 flux."

This change will be in the revised MS.
* * *
>Line 84: "the quantification of the air-sea flux will focus on the contributing factors
>of pco2 ...". Spatial contribution or seasonal contribution? Not clear

While the focus in this MS is on the seasonal cycle, geographic differences are also highlighted, at least for $pCO_2$ and air-sea $CO_2$ flux. However, to highlight the seasonal focus, we will amend the MS to read:

*Quantification of seasonal air-sea $CO_2$ flux will focus on the contributing factors of $\Delta pCO_2$ , the difference in seawater and atmospheric $pCO_2$, and wind.*
* * *
>Line 84: should say "deltapco2 and wind" instead of pco2 and wind. What about
>the contribution of atmospheric pco2?

The change will be implemented in the MS (see directly above). Atmospheric $pCO_2$ is clearly a factor in determining $\Delta pCO_2$ and has a seasonal signal of its own. However, in this region $\Delta pCO_2$ is almost always positive (see Figure 2a inset), meaning that the magnitude of annual oscillations in atmospheric $pCO_2$ is generally not enough to reverse air-sea $CO_2$ flux direction. Thus, atmospheric $pCO_2$ variability will only impact the magnitude of flux. Considering that during the year sea surface $pCO_2$ can vary by $\sim 200 \mu$atm vs. $\sim 4$-$6 \mu$atm in the well-mixed atmosphere (range for Keeling curve), most of the variability will be driven by the sea surface.
* * *
>Line 84: should say "sea surface temperature". Same for salinity

We understand the confusion and the need for clarification, and note that the need for consistent phrasing is repeated in the rest of the MS. So, we propose the following changes:

*In particular, the role of sea surface temperature (SST), sea surface salinity (SSS), DIC, and TA in determining the seasonal cycle...*

For the rest of the MS, will use SST and SSS.
* * *
>Line 85: "which of course also varies from region to region within the AS". Should
>say "for the entire domain of the AS as well as its spatial heterogeneity within the
>AS".

The suggested changes will be implemented in the revised MS.
* * *
>Line 87: "biological production/respiration". What about calcification
>remineralization or Nitrification denitrification?

CaCO$_3$ calcification, remineralization, and nitrification/denitrification are all processes included in the biogeochemical model used here. However, by use of the phrase "such as," the intention was to simply provide an example, not all of the physical vs. biological process comparisons that could be made. We will specify this in the revised sentence:

*...such as advection and mixing versus biological production and respiration, among others.*
* * *
>Line 91: "resolved by the data" Not clear which data? from which study?

We will change this to specify:

*resolved by the in situ data*

Generally speaking, in our revised text we will try to make sure data refers to the limited, in situ data, whereas model "data" will be referred to as "output"
* * *
>line 91 In the beginning of the introduction, you refer that data in the AS are very
>old (20 years ago) and here you refer to study from 2015. Not clear

Yes, we understand the confusion. In some instances, the "old" data from the 1990s are used in contemporary studies, such as the Cao et al., 2020 reference you have provided. Additionally, many studies, such as the references in this sentence, are all model results, and as mentioned in the previous response we will rectify this to specify these are model results, not "data":

*resolved by the in situ data, although models suggest interannual...*
* * *
>Line 92: "The rest of the paper" should use another sentence/to be reformulate.

We understand that this may be too informal, the sentence will be rephrased as:

*The*  *study begins with a description...*
* * *
>2
>2.1
>Question: How many data in SOCAT and LDEO database? Monthly? Daily? annual
>values? what period of study? What is the spatial resolution? Do you include the
>coastal datasets?

The SOCAT database has 186,970 observations in the AS domain, whereas LDEO has 90,870, meaning SOCAT has roughly double the observations. The reason for this difference is that the LDEO database collects from a specific methodology for measuring pCO$_2$, whereas SOCAT merges multiple methods. Since these are both collections of data from multiple cruises, there are no regularly recurring monthly/daily/annual observations. SOCAT includes observations from the years: 1962-3, 1991-2, 1995, 1997, 1999, and 2014. LDEO has observations from the years: 1962-63, 1980-81, 1991-2, 1995, and 1999. Being ungridded data, there is also no regular spatial resolution.

The hope was that the monthly and annual distribution of sampling in the AS would be visible from Figure 2b. All the data are included, and no difference is made between open ocean vs coastal sampling locations. In order to demonstrate how both the SOCAT and LDEO datasets cover similar areas and include coastal areas, please see the plot below showing the sampling locations.

[Figure]

[Figure]

Fig. R1-1. Sampling locations for (a) SOCAT and (b) LDEO.

In the revised MS, we will include these numbers in Sect. 2.1 so that our choices regarding datasets (such as preferring SOCAT) are more transparently motivated.
* * *
>Question: paragraph from line 110: Why in addition to the Socat and LDEO
>observation data, do you also add the climatology of tk09 and L15? what more do
>they bring? And why you used the climatology of TK09 since others and more recent
>are available (see e.g., Rodenbeck et al., 2015 which compare different pco2
>climatologies). I do not understand why you choose TK09?

As mentioned in the introduction, since there are few in situ data, we made the effort to validate our model output with these data directly. The AS modeling studies cited in the introduction compare their models to TK09 as the validation step. Therefore, the comparison with SOCAT/LDEO $pCO_2$ is our attempt to be more thorough in our model validation, and the comparison with TK09 for air-sea $CO_2$ flux is an attempt to provide context that allows for comparison with previous modeling work in the AS, despite the fact that more recent climatologies are available such as Rödenbeck et al., 2015. Additionally, in Rödenbeck et al., 2015 it appears to be that only the southern portion of the AS is represented.

Regarding L15, even though it is also calibrated with available LDEO/SOCAT data, as a forward-feed neural network, we include it because it is an independent methodology to the model and TK09 with relatively high spatial resolution ($1^o$x$1^o$), and also because it is a de facto standard for global $pCO_2$ estimates, such as the GCP and IPCC reports.

Please see our response below to the question "Line 109: "The gridded products" monthly?daily?annual grid? For which period?" for our proposed text to clarify these choices of TK09 and L15, as well as to highlight why they (and the other $pCO_2$ datasets) are needed.
* * *
>Question: Is the coastal domain included in your definition of the AS domain? if yes
>(e.g., line 157 you use the word coast), the pco2 climatology of Laruelle et al 2017
>and Roobaert et al (2019) for the coastal flux should be considered here? You refer
>to TK09 and L15 but what about the coastal regions?

The AS domain does include both the open ocean and coastal regions. The spatial resolution of TK09 is indeed problematic for the coasts, whereas L15 does include values within 100km of the coast. We appreciate that some climatologies are better suited for the open oceans whereas others are specific to coastal regions. Considering how the AS is ocean-dominated as a coastal region (the conclusion of the Cao et al., 2020 reference), and that the climatology of L15 and the in situ $pCO_2$ data include the coast, adding a sub-regional air-sea $CO_2$ flux analysis specifically for the coasts, while possibly useful, would also risk further confusing the reader with the addition

of another dataset. This concern arises especially in light of the amount of reorganization and clarification necessary for the major revisions in this MS.
* * *
>Comment: In the text, a clarification is need on the period in which you are going to
>work + spatial resolution + temporal and how you have treated the different
>datasets to switch, for example, to seasonal. Why are some databases used only for
>model validation while others for the flux estimation (table 1)?

We are happy to elucidate further the period/spatial-temporal resolution in which we focus in the paper. This study is on the baseline seasonal cycle of $pCO_2$ and air-sea $CO_2$ flux. The model is initialized and forced with a mix of climatologically averaged data which do not belong to a specific time period, such as 1990-2011, apart from the time period that any data at all exist in the AS. As discussed in a later response below, the implied assumption is that the "baseline" seasonal $pCO_2$ and air-sea $CO_2$ flux has not changed considerably over the past decades. As a result, model output is designed to be representative of the AS during any of the past few decades. To this end, we will add to the MS:

*... $pCO_2$ values were calibrated to the year 2005, the representative year used for the model's atmospheric $xCO_2$. The year 2005 was chosen for the model's $xCO_2$ concentration because it is the end of the historical period for the Intergovernmental Panel of Climate Change (IPCC) models. The earliest SOCAT data comes from 1962, and different databases used in this study stem from similarly different timespans. As a result, in using them this present analysis both assumes the existence of and attempts to quantify a baseline seasonal $pCO_2$ and air-sea $CO_2$ cycle that has remained stable over the past decades.*

Regarding data treatment, all calculations and manipulations on the model output take place at the monthly level, and are then aggregated into seasons to reflect the monsoon and inter-monsoon periods. To compare in situ data to the model output, we take each observation within a given month at a given lon/lat and interpolate the model's monthly $pCO_2$ value for the position. When calculating air-sea $CO_2$ flux from other datasets, again monthly values are used and then aggregated into seasons when desirable. This treatment to the in situ data will be added to the first paragraph of Sect. 2.1, and the treatment of comparison datasets is in a response below.
* * *
>Comment: This manuscript needs a paragraph about the spatial and temporal pco2
>and FCO2 uncertainties.

We agree that more needs to be discussed on the uncertainties in these quantities. The SOCAT protocol is designed so that observations with quality control flags A and B are within $2\mu$atm. Within our dataset, this is the case for <20,000 observations, about 10%. Since wind is prescribed in our study, we assume there is no uncertainty there. Using the average SST and SSS from the A/B quality SOCAT observations, we get $2.68e^{-2}$mmolC/m$^3$/$\mu$atm, resulting in changes of 0.0018, 0.0443, and 0.177molCm$^{-2}$yr$^{-1}$ of deviation with 1, 5, and 10 ms$^{-1}$ winds, respectively. We will include these numbers in the revised MS to provide the reader some insight into uncertainty.
* * *
>Comment: I suggest moving section 2.4.3 before 2.4.1

We understand your suggestion, and those made later in this review, to re-organize the paper so that air-sea flux is presented first before $pCO_2$ variability. Granted, air-sea $CO_2$ flux is the putative focus of the paper, and so this order makes sense, and other studies such as Sarma et al. (2013) do the same. We ordered the paper this way primarily because a lot of text is dedicated to first validating the model through $pCO_2$. Moving on to air-sea $CO_2$ flux after $pCO_2$ validation, followed by revisiting $pCO_2$ seasonality, felt less continuous in the narrative sense. However, we are open to the consideration and will make an attempt in the revision to follow this order.
* * *
>Line 98: "pco2" should specify that it is sea surface pco2

This will be changed in the revised MS.
* * *
>Line 100: "model pCO2 is calculated from DIC and TA" how?

The methods used in the model are discussed in the following section, so for the curious we will add the following:

*…model $pCO_2$ is calculated from DIC and TA (see Sect. 2.2), $pCO_2$…*
* * *
>Line 101: "in situ un-gridded data merged from SOCAT v. 2019" why you do not
>discuss about this dataset in the Introduction. In the introduction, you said that
>there is in the AS only observational data from 20 years ago (JGOFS) and here you
>discuss about LDEO and SOCAT. Not clear

Both LDEO and SOCAT contain (and in the AS mostly consist of) the JGOFS data. We would hope that this is self-evident, with both products being attempts at being collections of all available in-situ data of a given methodology. This is reflected in the fact that most of the in situ data we use come from the 1990s, as is visible in Figure 2b. We briefly mention SOCAT in the introduction to make the point that the AS is under-represented in part because of the lack of follow-up in data collection following JGOFS. To clarify the point, we will add that SOCAT and LDEO both contain the JGOFS data and whatever few data there exists outside of JGOFS:

*Here, $pCO_2$ validation stems from in situ un-gridded data merged from SOCAT v. 2019 (downloaded from https://www.socat.info/index.php/version-2019/ September 2019) and the Lamont-Doherty Earth Observatory (LDEO) surface $pCO_2$ database (Takahashi et al., 2019). Both databases aggregate all available in situ surface $pCO_2$ data, including JGOFS.*
* * *
>Line 103: "SOCAT data was preferred to LDEO observations" why?

In answering your first question for Sect. 2, we addressed this concern by showing that SOCAT is twice as dense, which we will add in the revised MS:

* SOCAT and LDEO contain >180,000 and ~90,000 data in the AS, respectively. SOCAT has more data because it includes multiple methodologies. As a result, SOCAT data were preferred , and LDEO observations were included for years where SOCAT data are unreported.*
* * *
>Line 103: "large overlap" how many data? you should be more quantitative and more
>precise in the words used.

As seen in the response above, the "large overlap" argument will be removed from the MS, and we appreciate the need to be more precise in our wording.
* * *
>Line 105: "... using reported sea surface temperature (SST) and S data included in
>the products" S= sea surface salinity? which equation do you use to convert fco2 to
>pCO2? Not clear how fugacity and partial pressure are related to SST and S

As addressed previously, S will become SSS for sea surface salinity. The equations to convert fugacity to $pCO_2$ are included below for your reference. Only SST is needed for the conversion from $fCO_2$ to $pCO_2$, whereas salinity enters the picture for the vapor correction in converting

pCO$_2$ to xCO$_2$. Since these equations are both somewhat long due to their thermodynamic nature and are not particularly illuminating for the reader, we will include a reference in the revised MS:

...reported SST and SSS data included in the products *using routines from the CO2SYS software package (Van Heuven et al., 2011).*
Here are the equations:

$$pCO_2 = fCO_2 \cdot e^{-(b+2\delta)\cdot P/(R\cdot(SST+273.15))} \tag{1}$$

where b is:

$$b = -1636.75 + 12.0408\cdot(SST+273.15) - 0.0327957\cdot(SST+2713.15)^2 + 3.16528\cdot10^{-5}\cdot(SST+273.15)^3 \tag{2}$$

and $\delta$ is $(57.7 - 0.18\cdot(SST+273.15))$. P is 1.01325 bar (1 atm), and R=83.1451 ml bar $K^{-1}mol^{-1}$.
* * *
>Line 106: why a value of 2 muatm yr-1. Should be justified.

We apologize for the wording, the anthropogenic effect on pCO$_2$ was calculated to be 2 by linear fit to the merged data; this value was not prescribed a priori. The revised version will be:

*The anthropogenic effect of increasing surface pCO2 was removed by calculating  a fit linear trend of 2 $\mu$atm yr$^{-1}$, slightly...*
* * *
>Line 109: "The gridded products" monthly?daily?annual grid? For which period?

We understand that each dataset used in this paper should contain information regarding its spatiotemporal resolution and scope. For example, both TK09 and L15 are global, monthly climatological products, but have different calibration years of pCO$_2$, as well as slightly different decadal spans. This information, all written out, becomes both hard to follow and tedious, which is why Table 1 is included. We will re-write this paragraph so that at the beginning the reader will be directed to Table 1 where the pertinent information (reference xCO$_2$ year, gridded or not, spatiotemporal resolution, use case, etc.) for each dataset is visible:

*Alternative pCO$_2$ products are used for comparison purposes. A complete list of these datasets and their characteristics is provided in Table 1. For all the comparison datasets, air-sea CO$_2$ flux is calculated from monthly values. $\Delta$pCO$_2$ values are calculated using Keeling curve values (downloaded from https://www.esrl.noaa.gov/gmd/ccgg/trends/gI_data.html, downloaded September 2019) of atmospheric xCO$_2$ for the respective calibrated year of each data set. The same climatological winds as used in the model (Sect. 2.2) are applied to the pCO$_2$ products. The gridded product TK09 is chosen because previous modeling studies in the AS use it as validation (see Introduction).  The L15 climatology, while based upon the same in situ data mentioned above, represents different processing methodologies, and as a high-resolution, global pCO$_2$ dataset, also serves to provide independent context to the model validation. pCO$_2$ is also calculated from DIC and TA provided by the statistical fits to JGOFS data by Sarma (2003) and to the gridded GLODAP climatological product. The statistical fits of Sarma (2003) were used twice, first using model monthly SST,SSS, and Chl-a, and second with World Ocean Atlas (WOA) 2009 monthly SST, SSS with SeaWifs Chl-a. GLODAP-derived pCO2 also uses WOA2009 monthly SST, SSS applied to the annual DIC, TA values. Calculations of pCO2 are performed using the*

*CO2SYS software package (Van Heuven et al., 2011). Since all calculations are conducted at the near-surface, differences between this software suite and Orr and Epitalon (2015) are minimal.* ~~*For air-sea flux calculations, all $\Delta pCO2$ values were calculated using Keeling curve values (downloaded from https://www.esrl.noaa.gov/gmd/ccgg/trends/gl_data.html, downloaded September 2019) of atmospheric xCO2 for the respective calibrated year of each data set (1995 for Sarma (2003), 2001 for L15, 2002 for GLODAP, 2005 for TK09). A summary of these datasets and their characteristics is provided in Table 1.*~~
* * *
>Line 115: Which products/datasets do you used to calculate the air-sea flux?

Please see the response above.
* * *
>2.2
>Line 121: "System-AGRIF (ROMS-AGRIF) 3.1.1." version 3.1.1? what means AGRIF?

AGRIF is an acronym for "Adaptive Grid Refinement In Fortran," which is a tool to create one and two-way nesting of model grids. Though this particular capability is not used in this study, we include it as part of the name. The full name, along with the explicit "version" will be added to the MS.
* * *
>Line122: "Previously used in the region (Lachkar et al., 2016), the..." should be by
>Lachkar et al.,

This change will be implemented in the revised MS.
* * *
>Line 125: "advection of tracers" in repeat twice in the same sentence

The repeated phrase will be deleted in the revision.
* * *
>Line 125: what is the K-profile?
>Line 126: what is KPP?

K-profile, and K-profile parameterization (KPP), is a vertical mixing scheme, described in Large et al., 1994. A real description of it is well beyond the scope of this forum, but the K stands for diffusivity, and the theory revolves around K's vertical profile and its relation to the gradient Richardson number. Since KPP is relatively well-established in the field, we leave it to the reader to look at the reference for further information.
* * *
>Line 127: 33° W or E?

We apologize, it is 33$^o$E, and this will be added to the MS.
* * *
>Line 127: which previous studies? Need references.

This choice is primarily to match with Sarma et al. (2013), which at one point in the analysis explicitly considers the AS north of the equator. This will explanation will be included in the revised version:
    *For the sake of comparison with* *Sarma et al., (2013), we will...*
* * *
>Line 127: sometimes you speak at present sometime in future→same comment for
>all text.

We understand that sometimes in the course of writing that tenses can be wrongly intermixed, which further confuses meaning. The revised MS will be re-read carefully to make sure there is consistency in grammatical structure.
* * *
>Line 130: what is the name of the OBGCM model?

The biological model has no colloquial name based off an acronym, such as PISCES or NE-MURO. Internally, we refer to it as $N_2PZD_2$ since there are two nutrient and detrital components, along with a single component each of phyto- and zooplankton.
* * *
>Line 131: "Biological parameters for the model are the same as those used in Gruber
>et al. (2011)." That is to say? Which parameters? Should be enumerated.

We sympathize with the reviewer that modeling papers often simply include a reference and expect the reader to wade through the details. To this end, we will add a Table that includes these values for reference.
* * *
>Line 136: what about nitrification and denitrification?

Both nitrification and denitrification are included in the model; this is implied by "removal and creation" or nitrate, but we will include them explicitly in the revised version:

*TA changes with the removal and creation of nitrate ($NO_3$), including nitrification and denitrification, as well as dissolution/precipitation of $CaCO_3$*
* * *
>Line 144: "climatological mode". Not clear, what is the period?

Apologies, "climatological mode" is short-hand for running the model with climatological, averaged forcing. Since it is confusing, this will be removed in the revised MS:

*The model was run  with 360-day years and interpolated , climatologically averaged monthly forcing.*
* * *
>Line 145: "surface salinity". In the text before it was S. Not clear

Similar to previous responses, this will now be SSS in the revised MS.
* * *
>Line 146: SST is from 1985-1997 but then you make a FCO2 comparison for
>another period. Still confused about the period of this study. What is the spatial
>resolution of SST?
>Line 147: Why do you need wind stress? The period of the wind stress is
>1999-2009, different than SST. Not clear again.

The questions posed are quite logical and pertinent. Indeed, the SST forcing is based from a climatology dating from 1985-1997, and the climatology for wind stress comes from a separate period, from 1999-2009. In the ideal, this makes no sense, and every source of forcing in the model should correspond to the same time period of study. Unfortunately, this is not the case as surely the reviewer is aware. However, in using these climatological forcing datasets, we understand the

need to explicitly state the underlying assumption that the dynamics in the forcing variables have not changed. This assumption is stronger for some variables and less so for others. For example, with ongoing climate change mean SSTs have probably changed (increased) in the time since 1997, whereas the strength and direction of summer monsoon winds have not changed much. In order to more directly state our assumption, in the revised MS we write (starting Line 144):

*... monthly forcing. The different climatological products derive from datasets spanning slightly different periods, and so here we assume in this study that the dynamics represented within them have not changed in the time since. Heat flux, ...*

The SST spatial resolution is $\sim 0.1^o \times 0.1^o$, less than 10km at the equator. Wind stress is the form in which wind forcing is implemented in ROMS (of course, u/v wind speeds can be converted to wind stress), and so it is necessary to run the model.
* * *
>Line 148: "T, S" same comment than before. SST or T .. not clear

Yes, thank you for keeping track, this will be remedied in the revision.
* * *
>2.3
>Line 157-158: need reference from the literature to support summer monsoon and
>enhanced biology.

Please find an added reference below:

*...but also enhanced biological productivity (Schott and McCreary Jr, 2001).*
* * *
>Line 162: same comment need reference for winter monsoon NPP

Please find an added reference below:

*...where the winter monsoon's primary productivity is most intense Kumar et al., 2001).*
* * *
>Figure 1: what is the period of the study?

The SeaWifs mission spanned from 1997 to 2010. As mentioned in a previous response regarding climatologies, the assumption inherent in this comparison is that average Chl-a does not change much outside this time period.
* * *
>2.4
>2.4.1
>Line 168: "The proximate variables impacting pCO2 in the" should be "..that affect
>pco2 change in the"

The wording will be changed to reflect this in the revised MS.
* * *
>Line 170: What is the depth of the water column where the pco2 decomposition into
>processes is performed in the model?

The decomposition is calculated for the topmost cells in the grid. Since ROMS has an sigma-coordinate system in the vertical, these cell depths vary but the average is 2.6m deep.
* * *
>Line 171: In equation 1, terms on the RHS are not described in the text and not clear
>when you said "deviation from an average."

Yes, you are quite right, the partial derivatives are not described. Additionally, we will clarify what we mean by "deviation from an average." The revision will be:

*...likewise express deviations from a prescribed value depending on whether the deviations are spatial or temporal in nature (see below). The coefficients of the $\Delta$ terms are partial derivatives of $pCO_2$ with respect to these variables, namely DIC, TA, SST, and SSS, and are calculated via centered differences described below. However, in order...*
* * *
>Line 181: how partial derivative are calculated? Which coefficient are used?

The description of how partial derivatives are calculated starts at line 190. To better illustrate the calculation, we will include an example calculation:

*...determined by Orr et al., (2018). For example, to calculate the monthly $pCO_2$ anomaly due to SST for a gridpoint with annual mean $pCO_2$ of 430 μatm, annual mean SST of 24º C and monthly SST of 26º C:*

$$\Delta pCO_2 \approx \frac{\partial pCO_2}{\partial SST} \Delta SST + ... \approx \frac{pCO_2(24 + 1 \cdot 10^{-4}, ...) - pCO_2(24 - 1e - 4)}{2 \cdot 1e - 4} \cdot (26 - 24) + ... \quad (3)$$

*where $1 \cdot 10^{-4}$ is the recommended SST deviation.*
* * *
>2.4.2
>Line 197-198: what about the river flux? Did you include the contributions from
>rivers?

Yes, indeed, riverine flux impacts salinity as well as carbon variables such as DIC and TA, and certainly there are freshwater inputs coming from India during the summer monsoon. However, these fluxes are not directly implemented as land/ocean boundary conditions. Instead, DIC and TA are altered by virtual fluxes that are proportional to SSS changes as prescribed by the COADS climatology. We will note this at the appropriate place in line 139:

*Surface fluxes of DIC and TA due to evaporation,  precipitation, and river input were included as virtual fluxes proportional to SSS forcing.*
* * *
>Line 202: Eq 6 what are "New" and "Reg" and why CaCO3precip.remin has the same
>sign than PPnew+reg (both with a -)?

"New" and "Reg" are the two forms of primary production, "new" production from nitrate assimilation and "regenerated" production from ammonium. Primary production and $CaCO_3$ production all remove inorganic carbon from the water column, so their contribution to the DIC budget is negative.
* * *
>2.4.3
>Line 226:"five-year average" why five? Not clear

The five-year average is necessary because of the definition of the Reynolds decomposition. In order for the equations in the decomposition to be exact, the mean needs to be defined against the entire timeseries. We use five years of model output to account for internal variability in the model, and as a result the mean needs to be calculated around the entire five years of output. We will make reference to this in the revised version:

*...where' indicates an anomaly and $\overline{x}$ is a five-year average of variable $x$, which are calculated at each grid point.* *The five-year average is necessary for exact closure in the Reynolds decomposition.* $F'_{CO_2}$ *is the...*
* * *
>3
>3.1
> Question: Why did you analyze the variables in depth in Fig. S1 and Fig. S2 and not
>just the spatial distribution at the sea surface as it is performed for Fig. S3? What is
>the reason for the choice to do in depth?

There was no conscious decision to show depth profiles in temperature and salinity but not with currents, but it is rather a consequence of data availability. Firstly, sea surface spatial distributions of temperature and salinity would reflect the surface forcing, whereas distributions of temperature and salinity at depth better demonstrate model performance and there is at least data available. The only measurements for currents stem from drifting floats that reflect the near-surface, and so this is the comparison we are forced to use.
* * *
>Question: I do not understand between figure 2 and figure 3 for the data spatial
>distribution. Are they the same data between both figures? It seems there is more
>surface covered by the data on figure 2 than when they are divided into seasons
>(figure 3)?

It is understandable that figures 2 and 3 might visually give the impression that these are not the same data. In figure 2, all available data are binned into a $1^o$x$1^o$ grid, which in itself inflates the sense of data coverage and may be the cause for the dissonance.
* * *
>Comment: Lots of imprecision on the form and problems with the numbering of the
>panels. e.g., Figure 3. "Seasonal surface pCO2 (muatm) from data (a-d)" panels are
>not ok between the figures and the caption.

Yes, many apologies, this caption is reflective of the initial distribution of subpanels, which was changed last-minute to maximize figure size in the submitted MS since .png files are not vectorized. The revised caption will reflect the updated placement of subpanels.
* * *
>Comment: the pco2 Seasonality has already been analyzed by the previous figures
>and compared to the SOCAT database spatially and temporally with a Taylor
>diagram. Why do a comparison to LA15 again? What more does this bring? not clear
>in the text

Figure 5 is included as part of the validation process because while Figures 3 and 4 show comparisons with the data (visual in Figure 3, quantitative in the Taylor diagram of Figure 4), the model's performance still needs further context. For example, Figure 3 visually demonstrates that the model reproduces summertime $pCO_2$ near Oman, while over-estimating $pCO_2$ most of the time elsewhere. Figure 4 shows that correlations are quite low and that model output's variability is less than the data's. Considering these weaknesses, does the model represent an improvement?

This is where Figure 5 shows that the model better captures the extremely high summer values seen in SOCAT than an alternative, popular pCO$_2$ product. We will revise the text to highlight the motivation behind this figure in the Results section:

*Direct comparisons between the in situ data and model output demonstrate the positive bias and middling correlations of the model with respect to the data, as well as the model's tendency to under-represent variability. As a result, it is necessary to investigate how these shortcomings compare with alternative pCO$_2$ estimates in the AS. Figure 5 shows monthly comparisons of the pCO$_2$*  *monthly probability distribution functions of pCO2 from in situ data, model output, and L15 demonstrate temporal variability between pCO2 products (Fig. 5). For most of the year, the data (Fig. 5a) stays within a relatively narrow range (375-425 µatm), except for the summer monsoon where values can exceed 500 µatm and the median value has its peak. In the model ...*
* * *
```
>Comment: I do not understand the added value of the Taylor diagram. What more
>does this figure add? moreover, you do not explain the various terms of the taylor
>diagram such as the Pearson coefficient. Why is it not done on the other variables,
>especially when you go directly to line 255: Since the model successfully replicates
>other tracers, physics, and biological processes". A little fast, no? In addition, the
>seasonality of these processes is not analyzed.
```

The Taylor diagram is added because it is a compact, visual way of representing multiple statistical quantities of interest to score a model's fidelity. As you have mentioned, it contains the Pearson correlation coefficient, as well as normalized standard deviation with model bias added as a color. In the revised MS we will include an explanation of the terms in the Taylor diagram:

*A Taylor diagram (Taylor, 2001) comparing in situ pCO2 data and model output shows the relative performance of the model (Fig. 4). The distance from the origin, r, is model variability normalized by standard deviation of in situ data. The angle created from the y-axis, θ, is the Pearson correlation coefficient between the model and in situ data. If the model were to perfectly reproduce the data, it would appear at the position (1,0), equivalent to a normalized standard deviation of 1, and correlation coefficient of 1. For the entire dataset...*

We think that doing a Taylor diagram on other variables, such as SOCAT SST and SSS, would be a wonderful addition. First, it would show the relatively high performance of the model in representing the physics, and provide comparison demonstrating the difficulty in reproducing pCO$_2$. As a result, the model's apparently lackluster performance in pCO$_2$ would better motivate further comparison with L15 in Figure 5. This way, we can provide better model validation of other tracers and physics without seeming to gloss it over, and also better motivate the three figures of validation done for pCO$_2$.
* * *
```
>Comment: You start with fig S1 to S3 discussing variables then you discuss about
>pco2 and then you discuss again with variables (line 255). Not clear. Should be first
>pco2 then variable or vice versa
```

Yes, we understand that the text should not move from one subject to another and back again. Jumping back to SST, SSS, TA, and DIC is part of an attempt to attribute the source of model bias. Understandably, this should all be part of the validation process, and the text will be moved before discussion of pCO$_2$ spatial and seasonal distributions.
* * *
```
>Line 230 "The implementation of ROMS-AGRIF presented here has been used in
>previous studies (Lachkar et al., 2016)." And? What is the conclusion of this
>sentence? Is it in the AS?
```

Yes, we should be more explicit. In saying "the implementation presented here", we meant this specific implementation in the AS that has been published in previous studies. We will be more precise in our wording in the revised MS:

*The implementation of ROMS-AGRIF presented here has been used in previous studies of the AS (Lachkar et al., 2016).*
* * *
>Line 233: Add panels with the mismatch (model minus data) in FIG S1 to S3 would
>be useful to highlight difference between model-data. Same comment for figure 3.

This is a good idea, especially since the data in Figure 3 can be crowded together and colors are difficult to directly compare between subpanels. As a result, we will add subpanels showing model minus data for Figs. S1-3 and Figure 3.
* * *
>Fig. S 2. "Transects, similar to Fig. S1, but with salinity instead of temeprature"
>should be temperature.

Thank you, the typographic error will be corrected in the revised MS.
* * *
>Lines 230-233: e.g. "Reproduce well the ..." need to add quantitative values.

Understandably, quantitative values are needed so that the reader can judge the validation for themselves apart from visual inspection of Figs. S1-S3. We will add values to the text in reference to the subpanels we will add showing model - data, as well as correlations and standard deviations from the Taylor diagrams to be added.
* * *
>Line 235: "Regarding pCO2, in situ data from the merged SOCAT/LDEO database
>shows that average binned pCO2 values in the region are positive for most of the AS
>(Fig. 2a). The ensemble of observations show that 90% of _pCO2 observations are
>positive, indicating positive flux to the atmosphere (Fig. 2a, inset)." These two
>sentences should be merged in one.

Understood, these two sentences will be combined as follows:

*Regarding $pCO_2$, in situ data from the merged SOCAT/LDEO database shows that ~90% of  $\Delta pCO_2$ values in the  AS are positive (Fig.2a, inset), indicating a positive flux to the atmosphere that is applicable geographically (Fig.2a). ~~for most of the AS (Fig. 2a). The ensemble of observations show that ~90% of $\Delta pCO_2$ observations are positive, indicating positive flux to the atmosphere (Fig. 2a, inset).~~*
* * *
>Line 239: "pCO2 value..." we do not know if it is modelled pco2 value or
>observational pco2 value. Maybe you could use "data-based pco2" and "modeled
>based pCO2" everywhere in the text.

Understood, as mentioned in an earlier response, efforts will be made in the revision to specify in situ $pCO_2$ data versus model output, including in this section.
* * *
>Line 242: Again, this paragraph needs quantitative values. In the pco2 model
>description, there is no value in the text. Use values instead of "Is lowest".

Yes, we understand the desire for more quantitative values. To this end, in addition to the subpanels suggested for Fig. 3, we will add a table that contains the means and standard deviations for both data and model so that there is more transparency in the results and so that the results section of the paper is not cluttered with repetition of values.
* * *
>Fig. 4 "Taylor diagram of modeled vs. observed surface pCO2, both in total and
>seasonal sub-sampling. Data are from merged SOCAT and LDEO databases,
>corrected to year 2005. Distance frmo origin .." should be from

This typographical error will be corrected in the revised MS.
* * *
>Line 260: "close to reported measurement error" give a value.

Yes, there needs to be more context. As stated in an earlier response, the best accuracy for SOCAT data is $\sim 2\mu$atm, and this quality applies to only 10% of the current database. We will revise this sentence to better reflect reality:

*...results in a $pCO_2$ shift of -6.8 and -3.5 $\mu$atm for SST and SSS, respectively, close in magnitude to the best-case*  *measurement error of sim2$\mu$atm.*
* * *
>Line 255: "tracers, physics, and biological processes" which are?

We will revise this to state:

*Since the model successfully replicates*  *temperature, salinity, surface currents, and seasonality of primary production (Fig. 1, Fig. S1-S3), we look for the source of bias...*
* * *
>Line 257: Why the SST from SOCAT is used here from the socat database while in
>Fig. S1 it derives from the WOA? Not clear

SST and SSS are used from SOCAT because 1) this is as close as possible to the conditions of in situ data collection, and 2) while it is possible some of these data are used in generating the WOA, it is somewhat independent because WOA is used to initialize the model (even though satellite data forcing at the surface is probably more important). Additionally, in Fig. S4 we are hunting for the bias in these parameters at the SURFACE, whereas in Fig. S1-S2 a better measure of model performance is at depth, necessitating the use of the WOA dataset.
* * *
>Line 258: "sst" sometime sst sometime T. Not clear. Same comment for every
>variable e.g. Fig S4, SAL instead of S or SSS. Be consistent all text and figures.

Yes, thank you for pointing another instance of the terminology not being consistent, in the revised version it will be SSS.
* * *
>Fig. S4. same comment SST or temperature... not clear

Thank you, this will become SST in the revised MS.
* * *
>Line 267: Figure 5, I suggest merging the three plots in one and use monthly
>boxplot for instance for median, 5th and 95th. Easier for the reader.

Thank you for the idea, we are willing to alter Figure 5 so that it is all in one plot and consisting of monthly boxplots for the three datasets.
* * *
>Line 277: "competiting product" Not clear. what that mean? Observational? Model?
>Not clear

The revised text will be more precise:

*...the model captures the high summer monsoon $pCO_2$ values better than the  alternative L15 climatology .*
* * *
>3.2
>3.2.1
>Comment: need to be more quantitative.

In order to be more quantitative, we propose to add a table for Sect. 3.1. For this section, we will add more values:

*Spatial $pCO_2$ anomalies calculated from the  annual mean highlight the geographic hotspots of $pCO_2$ inside the domain (Fig 6a). $pCO_2$ anomalies range from -89 to $+415\mu atm$, indicative of a positive skew in the distribution. Within the regions of analysis prescribed in this study, it is clear that Oman, the Indian coast, and the North AS including the Gulf of Oman host enhanced $pCO_2$ , with average positive anomalies of 8.6, 21.5, and $49\mu atm$, respectively. In contrast, both the oligotrophic central AS and Somalia regions have negative $pCO_2$ anomalies (-13.7 and -2.9μatm). The contributing factors to these $pCO_2$ anomalies, temperature, DIC, TA, and freshwater components, display differing distributions. Temperature (Fig. 6c) contributes toward negative $pCO_2$ anomalies in a southwest-to-northeast band along the coasts of east Africa and the Arabian peninsula, up to the coasts of Pakistan and northern coast of India near Gujarat. The cold SST structure contributes to a -20μatm effect on $pCO_2$, and largely overlaps the stronger summer monsoon winds (Fig. S5). The opposite trend is found in the central oligotrophic and Indian regions, where the average temperature contributi to $pCO_2$ is 20μatm  despite upwelling in the southern Indian coast. The distribution of DIC-induced anomalies (Fig. 6d) shows a positive influence near coastal regions and the western AS off the coast of Somalia ($+25\mu atm$), whereas a strong minimum is found in an oval region encompassing the central, open-ocean AS(average -36.6μatm effect). TA effects (Fig. 6e) show a north-south gradient with positive contributions to $pCO_2$ occurring in the north and negative towards the south, in a similar distribution to surface salinity gradients in the AS and with a +20 to -20μatm difference, similar to temperature. Freshwater contributions (Fig. 6f) show a similar distribution as TA, but weaker in magnitude ±10μatm .*
* * *
>Comment: You should be careful when you use the term "dpco2" for change in pco2
>spatial or temporally. You also use dpco2 for the air-sea CO2 gradient. Not clear
>e.g., Fig. 6 and Fig.2.

True, in the scientific literature $\Delta pCO_2$ refers to the air-sea $CO_2$ gradient used in the calculation of air-sea $CO_2$ flux, and the casual reader may completely misinterpret the meaning of Fig. 6 and 7. As a result, the figure labels will be changed to read $pCO_2$ anomaly instead of $\Delta pCO_2$.

>Comment: maybe specify in the text what corresponds to a positive anomaly and a
>negative anomaly on pco2

In asking what a postiive vs negative anomaly corresponds to, perhaps this means to better define what the averaging process is. The map of spatial anomalies is derived by subtracting the spatial average of pCO$_2$ from the map of temporal average (annual mean) pCO$_2$. Thus, a positive anomaly indicates a region has higher pCO$_2$ than the domain as a whole. Please see the proposed revision above for the specification of positive vs negative anomalies.
* * *
>Line 286: T, S alk are variable but FW is a process. Maybe salinity instead of FW?

We understand that the phrasing for freshwater is more evocative of a process as opposed to a variable. We have no objections to using salinity, but it is necessary to stress that a change in pCO$_2$ due to "salinity" includes effects where DIC and TA increase linearly with salinity.
* * *
>Line 284: In fig 6, maybe add numbers to regions for each box, then in panel b also
>add the number instead of name. Easier for the reader to say in panel b where is the
>regions in panel a

If the reviewer finds that numbers instead of the names to be useful for a quick reference, we as the authors have no strong objection to this and the change will be made in the revised figure.
* * *
>Line 286: "The cold SST structure largely overlaps the stronger summer monsoon
>winds." Not clear

Yes, in the text we simply state this, but do not show it. We will add a supplementary figure that shows the summer wind stress forcing used in this study to show that indeed, the colder SST matches with the strongest winds in the AS.
* * *
>Line 301: "The previous section outlines why certain geographic regions within the
>AS have overall high or low pCO2 values" This does not explain why. Just how the
>variables change within different regions but not in terms of process the section
>does not explain why. It DESCRIBES the pCO2 changes associated with the variables
>but not processes; pay attention to the word used. Question: what are the processes
>that explain the spatial distribution of pco2? Not clear

Yes, thank you, we take your point that these variables are descriptive, perhaps indicative, but not explanatory in the way of processes or mechanisms. We will re-word this so as not to mislead readers:

*The previous section outlines the geographic regions within the AS that have overall high or low pCO$_2$ values, but in order to investigate the strong seasonal monsoon  cycle in the AS, the decomposition of variables affecting monthly pCO2 values is calculated at each model grid point and averaged into each analysis region (Fig. 7).*

Reviewer 2 has also stressed the importance of highlighting processes. They have suggested that in addition to a detailed analysis of DIC fluxes, to look at TA and SST so that the overall impact of mechanisms such as advection/mixing, primary production, CaCO$_3$ calcification/dissolution, etc. can be linked first to SST, SSS, TA, and DIC, and then ultimately to pCO$_2$. Until this analysis is implemented in the revision, the answer to your question is not immediately clear.
* * *
>3.2.2
>Comment: Again, this section needs numbers to support the text. E.g., what is the
>typical amplitude of the respective seasonal signal?

Similar to section 3.2.1, numbers will be added to this section, such as the extrema of domain-wide $pCO_2$ anomalies (-9.3 to $20.3\mu$atm), DIC contributions to $pCO_2$ in Oman (-40.7 to $81.1\mu$atm), etc.
* * *
>Question: How do you explain that the important contribution of DIC in the
>upwelling regions is not seen on the seasonality of the entire domain (Fig. 7a)

Yes, considering how strong the upwelling signal in DIC is near the coast of Oman, it is curious that there is not more of an impact on the domain as a whole. The simplest answer is that the upwelling region both consists of a smaller area (Oman region is 5% of the domain total), and that excess DIC is not exported as much from the region (see Horz. Circ. in Fig. 8b).
* * *
>Line 302: "the decomposition of factors affecting monthly" should be "variable"
>instead of "factors"?

The word "factors" will become "variables" in the revised ms.
* * *
>Line 303: Figure 7. "Timeseries of pCO2 anomalies (muatm)" units should be muatm
>month-1

We understand that these are monthly anomalies, but they do not represent fluxes, and we disagree that the units should be changed to $\mu$atm month$^{-1}$, and suggest that it remains $\mu$atm.
* * *
>Line 304: "pCO2" should be called "pco2 anomaly"

This change will be implemented in the revised MS.
* * *
>Line 305: "DIC acts" DIC is not a process. Should be "change in pCO2 associated to
>DIC change"

Yes, we understand, the language will be changed to reflect this and not mislead the reader.
* * *
>Line 307: you first discuss about Fig.7a in the text for the entire domain. Then you
>discuss about the oligotrophic central region. Hence, Fig.7d should be Fig. 7b. The
>different panels in figure 7 should be in order of what they are called in the text.

Fair enough, in the revised MS we will change the order of subpanels to reflect when they are mentioned in the text, so that the order will be: Domain, Oligotrophic, Oman, India, Somalia, and the North AS.
* * *
>Lines 320-326 Maybe this section should be in the discussion section instead of the
>result section.

These lines were added to briefly summarize the section's main points, but we are amenable to move this to the Discussion section, especially considering that the Reviewer suggests reducing the Results section.
* * *
>Line 328: "is controlled by the physical processes of surface forcing, mixing and
>advection" not clear what means the term "surface forcing"?

Surface forcing refers to heating/cooling. We will use this wording in the revised MS:

> *Whereas SST and its effect on pCO$_2$ is controlled by the physical processes of surface heating and cooling, mixing ...*
* * *
>Line 330: Figure 8. "Timeseries of DIC fluxes (PgCyr-1)" why the DIC flux is in Pg C
>yr-1? Should be inPg C month-1? Why do you call this variable "DIC flux" instead of
>"delta DIC" such as "deltapco2" before?

The unit of DIC flux in PgCyr$^{-1}$ was chosen to match the presentation of monthly air-sea CO$_2$ flux in Sarma et al., (2013). However, since in Fig. 12 the implied rate is in month$^{-1}$, we will change this to be consistent throughout the MS.

We call this variable DIC flux because, well, it represents the amount of DIC, in units of mass, that are added or removed per unit time through the volume's 2D boundaries or in situ due to biogeochemical processes. Considering that $\Delta$pCO$_2$ is to be removed in favor of pCO$_2$ "effect" or "deviation," there is no need to use $\Delta$DIC.
* * *
>3.3
>Comment: Again, this section needs numbers and more precise.

Similar to Sects. 3.1 and 3.2, we understand this need and will add appropriate numbers to the presentation.
* * *
>Line 330: "Budgets of DIC fluxes in the upper 20 m (Fig. 8) show that two major
>processes dominate, vertical circulation (light blue lines) and net biological
>processes (green lines)." I'm not sure that the term "budget" is appropriate here.
>This figure shows the monthly decomposition of DIC change into processes which is
>not a budget. Also, it is not clear when you said, "the two major processes
>dominate". They do not dominate; they show the largest seasonal variations but
>they not dominate the final seasonal signal.

We understand that in some contexts a budget consists of accounting for all the reservoirs of a given variable to estimate its totality (e.g. finding total organic carbon in a pond by adding up all phytoplankton, macrophyte, zooplankton, fish biomass, etc.). Since the model already has total DIC, we instead tally up all the monthly fluxes that add and remove DIC in a given volume over the course of a year, such that they balance out to create a closed cycle and DIC "budget."
However, if the precise use of "budget" in this instance is deemed inappropriate, then in the revised MS is can be easily foregone:

> * DIC fluxes in the upper 20m...*
* * *
>Line 330: Fig. 8 complicated to compare one region over another. All panels should
>be the same yaxis scale.

We understand that for the sake of comparison the y-axis should be the same across all sub-panels. However, due to the differently sized areas that the regions represent, the y-axis scale for India is about two order of magnitude smaller than for the entire domain. As a compromise, we will express the DIC flux as a volume-specific rate, and include on the top-right of each panel a number to indicate the relative size of each volume to that of the domain.
* * *
```
>Line 332: "acts as a source of DIC" when? All time? Which month? Should be more
>precise.
```

The following revision will be in the final MS:

*In the entire domain and all sub-regions, and for all months, vertical circulation (advection and mixing) acts as a source of DIC, with the sum of all biological processes acting as a sink...*
* * *
```
>Line 341: "escape is mostly smaller than biological flux," not clear. Smaller of what?
>In term of seasonal amplitude? For a specific month? Again, need to be more precise
>in the text.
```

Once more numerical values are introduced into the revised MS, this will be more clear:

*DIC flux due to atmospheric escape, while reaching its maximum magnitude of $\sim 27 TgCmonth^{-1}$ in June and July for the whole domain (Fig. 8a), only surpasses  biological flux in May, when 19TgC is released to the atmosphere compared to 12.3TgC.*
* * *
```
>Line 351-357 Maybe this section should be in the discussion section instead of the
>result section.
```

Similar to the last paragraph of Sect. 3.2, these sentences are designed to briefly summarize the main results. We can also move this to the Discussion section in the revised MS.
* * *
```
>3.4
>Comment: Again, this section needs more value to support results than only a
>qualitative description.
```

As with all the other sections in the Results, more numbers will be added in the revised text.
* * *
```
>Comment: I propose to change the order of the different sections. First, what
>governs the variability of FCO2, then pCO2. So, I suggest to first start with 3.1, then
>3.4 then 3.2 and 3.3.
```

As per the suggestion earlier on to re-organize the methods section, we are willing to change the order in the revised MS to begin with air-sea $CO_2$ flux before presenting $pCO_2$ variability and DIC fluxes.
* * *
```
>Line 370: what about the contribution of atmospheric pco2?
```

As stated previously, the variability due to atmospheric $pCO_2$ is much less than that seen in the surface ocean. For example, if atmospheric $pCO_2$ were to remain constant in this model, the change to $\Delta pCO_2$ would be at most <5%. However, it is true that the term in Eqn. 8 is $\Delta pCO_2$, not $pCO_2$, so this will be amended in the MS:

*...can be attributed to the contributions of winds, $\Delta pCO_2$, and interacting cross-terms...*
* * *
```
>Line 372: Figure 10 "(a) Anomaly of air-sea CO2 flux during summer monsoon JJAS
>(molCm-2yr-1. Summer flux anomaly contributions due to (b) wind, (c) pCO2, and
>(d) cross-terms in Eqn. (8)". Could say what means a positive/negative anomaly.
>Also (molCm-2yr-1).
```

Thank you, we will include this information in the figure caption:

*Anomaly of air-sea $CO_2$ flux during summer monsoon JJAS ($molCm^{-2}yr^{-1}$). Summer flux anomaly contributions due to (b) wind, (c) $pCO_2$, and (d) cross-terms in Eqn.(8). Anomalies are calculated relative to the five-year average.*
* * *
>Line 374: "The _pCO2 contribution to" dpco2 or pco2? Not clear. What about the
>atmospheric CO2 contribution?

Yes, this will be corrected to be $\Delta pCO_2$ in the revised MS.
* * *
>Line 378: Figure 11. "Seasonal CO2 flux anomaly (purple) for winter DJFM monsoon
>(top-left), springAM(top- right), summer monsoon JJAS (bottomleft), and fall ON
>(bottom-right). Contributors to the flux are solubility/winds (k_,blue), pCO2 (red),
>and cross-terms (yellow)." Results are shown for entire domain and sub- regions."
>Problem with panel "top left" "top right" in the caption and the figure.

Yes, many apologies again, similar to Figure 3, the organization of this figure was changed to maximize its size in the submitted MS, but the caption missed being updated. It will become:

*Seasonal $CO_2$ flux anomaly (purple) for winter DJFM monsoon (top), spring AM (second from top), summer monsoon JJAS (third from top), and fall ON (bottom). Contributors to the flux are solubility/winds (k$\alpha$,blue), $pCO_2$ (red), and cross-terms (yellow). Results are shown for entire domain and sub-regions.*
* * *
>Line 378: Figure 11. It is complicated based on this figure to see which process
>controls the seasonal flux variability. This figure represents the budget for each
>season. It would be easier to represent the seasonal flux variability into processes
>using the same strategy than for the seasonal pCO2 anyalsis with seasonal profiles
>as realized for pco2 (Fig 7 and 8).

We understand your concern, we have no objection to recasting Figure 11 to follow Figures 7 and 8 for better presentation of the model output.
* * *
>Line 378: "The seasonal flux anomalies for all regions throughout the year are
>displayed in Fig. 11" should be "aggregated by season" instead of "throughout the
>year".

Yes, thank you, this smooths out the prose:

*The seasonal flux anomalies for all regions  are aggregated by season and displayed in Fig. 11*
* * *
>Line 393-395: Maybe this section should be in the discussion section instead of the
>result section.

Similar to previous sections, we are amenable to shifting these brief summaries to the discussion instead of results.
* * *
>4
>Comment: Lots of repetition of results in the discussion section. Should be reduced.
>I also propose to reorganize the discussion section in 1) model vs data, 2) spatial
>analysis (FCO2 and pCO2) and processes, 3) seasonal analysis (FCO2 and pCO2) and
>processes and include in 3) the point 4.3

We appreciate that, especially with the inclusion of more values in the Results section, that the Discussion can be pared down. Additionally, as with the Methods and Results, the revised MS will be organized as proposed.
* * *
>Question: according to your results, does the seasonal FCO2 and pCO2 variabilities
>and processes diverge between the open ocean and the coastal domain in the AS?

For air-sea $CO_2$ flux, the open ocean and coastal regions in the AS have the same seasonal signal, with a summertime peak. the $pCO_2$, however, diverges between the coastal seas and open ocean. In the open ocean, $pCO_2$ (and $\Delta pCO_2$) has a peak in the spring whereas coastal $pCO_2$ has its seasonal peak in the summer. Hopefully, this result is clear from Figures 7 and 9.
* * *
>Line 406: what is the order of magnitude of the bias of Valsala? same order as you?
>Need quantitative values.

In Valsala and Maksyutov (2010), in their Figure 4 (see their "North Indian" subpanel) the free-run model is (from visual estimation) $\sim$5-15$\mu$atm above TK09. By comparison, our model has an overall bias of 2.1$\mu$atm with respect to *in situ* data, with spring having the largest seasonal bias of 48.4$\mu$atm. We shall add these numbers to the revised MS:

*For instance, in the global data assimilation study of Valsala and Maksyutov (2010), they found an overall positive bias in the North Indian ocean, $\sim$+5 to +15$\mu$atm above TK09 compared to -3.1 to +48.4$\mu$atm with respect to in situ data. Additionally, that study found  a similar...*
* * *
>Line 428: "Since upwelling regions are limited in geographic extent near the coast,
>capturing their high pCO2 values can be difficult for other approaches, such as
>TK09 with its coarse grid". Same comment than before. Why did you use the TK09
>climatology instead of others finer spatial resolution climatologies. Also, since the
>upwelling is close to the coast, what about the coastal climatology of Laruelle et al.
>(2017)?

As we have mentioned previously, TK09 was included because many previous modeling studies use it, and while it is not part of our model validation, it serves as a useful baseline to compare to those studies. However, since the suggestion to use Laruelle et al., (2017) has repeatedly entered discussion, find below a figure comparing the monthly pdf distributions of the in situ data, the L15 climatology, and the combined L15 and Laruelle climatology published in Landschützer et al. (2020) doi:10.5194/essd-12-2537-2020, and downloaded Jun 2021 from NOAA NCEI.

[Figure]

Fig. R1-2. Monthly PDFs showing 5-25-50-75-95 percentile $pCO_2$ values of (top-left, mid-right) in situ SOCAT/LDEO data, (top-right) L15 climatology, (mid-left) L15+Laruelle climatology, and (bottom) the 25-50-75 percentile $pCO_2$ values of L15 and L15+Laruelle.

With the coastal climatology merged into L15, we can see that outside of summer the $pCO_2$ is slightly higher and probably more accurate. During the summer, however, $pCO_2$ is underestimated even more so than in L15. As the figure shows, including the new, higher-resolution climatology does not a priori increase skill in the AS. This exercise should further underscore the need for more *in situ* data in the region.
* * *
>4.2
>Line 434-445: should be in section 4.1

Fair enough, this section of text does compare the model to previous studies with regard to the bias and higher $pCO_2$ values, and would make sense in Sect. 4.1.
* * *
>Line 434: "from previous studies" observational or modelled studies? Not clear

The references in the following text contain a mix of both modeling and data-based studies. As a result, we will clarify as follows:

*The distribution of model $pCO_2$ is both similar to and different from previous data-based and modeling studies.*
* * *
>Line 446: what are the consequences on this seasonal mismatch on your
>decomposition into processes?

The consequences of seasonal mismatch vis à vis our decomposition into processes in the AS depends on the magnitude. For instance, if maximum $pCO_2$ is in June instead of August near Oman, this does not rule out upwelling as a process, but rather the wind forcing used in the model should be investigated to determine its timing. The greatest mismatch in our study is the springtime $pCO_2$ maximum caused by heating; there is no plausible reason to doubt that heating occurs (and as far as we can tell from the SOCAT data the model is not over-estimating SST), and multiple models presented show this feature. However, the fact that the data do not reflect a springtime maximum requires an explanation that is not readily available. This specific discrepancy, which does reflect an unresolved issue in the region, is explored in the following paragraphs.
* * *
>Line 462: "influence of outflows from marginal seas" how?

In this sentence, we merely mention that part of the reason for high salinity in the north AS is due to outflow from the Persian/Arabian Gulf. We will add references in the revised MS:

*...with strong tropical heating and the influence of outflows from marginal seas Prasad et al., 2001; L'Hégaret et al., 2015).*
* * *
>Line 470: "the dominance of temperature is clear" spatially or seasonally?

In this section we are discussion the seasonal $pCO_2$ anomalies, and so in the revised MS we will stress the seasonal aspect:

*Zooming out from the upwelling regions and looking at the whole AS, the dominance of temperature on the seasonal $pCO_2$ cycle is clear.*
* * *
>Line 471: Be careful when you compared processes on different time scale (seasonal
>vs interannual)

True, just because temperature is important for the annual $pCO_2$ cycle does not translate into its inter-annual variability being the determining variable in the AS. To avoid further confusion, this sentence will be removed:

*...temperature is clear.  In the domain average, ...*
* * *
>4.3
>Line 497: "and so no biologically-induced decrease of pCO2 occurs". Maybe add "in
>the final pCO2 signal".

In the revised MS, this changed will be made:
*and so no biologically-induced decrease of pCO2 occurs textcolorbluein the final $pCO_2$ signal.*
* * *
```
>4.4
>4.4.1
>Line 512: "where uptake of CO2 takes" should be "atmospheric CO2 uptake takes"
```

Thanks, this will be re-written in the revised version:

*...where uptake of $CO_2$ takes...*
* * *
```
>Line 518: Fig. 9d instead of Fig. 10d
```

Thank you, this will be corrected in the revised MS.
* * *
```
>Line 521: "The fact that model CO2 flux peaks in summer despite a wide-ranging
>spring peak in pCO2" For the entire region? Which region? Not clear
```

Yes, in this sentence we were referring to the entire region; we will be more precise in the revised MS:

*The fact that model $CO_2$ flux for the entire domain peaks in summer despite a  spring peak in $pCO_2$ for the domain as a whole, along with the Somalia and oligotrophic regions, is the first sign that...*
* * *
```
>Line 535: need more reference for the use of different k-parametrization and wind
>speed on the flux calculation (e.g., wanninkhof, Ho, Roobaert, ...)
```

True, there has been much more recent work on improving the air-sea flux calculation, and these studies should be highlighted:

*...the possibility that incomplete representation of winds and the various parameterizations of piston velocity must be considered in addition to $pCO_2$, especially in light of recent work in the field (Roobaert et al., 2018; Wanninkhof 2014; Ho et al., 2006).*
* * *
```
>Line 521-535: Need more comparison to other studies. e.g, you could compare your
>results with the study of Roobaert et al. (2019) that decompose the CO2 flux
>seasonality into processes in the region.
```

Yes, thank you for pointing out the recent global studies that decompose $CO_2$ flux and its constituents in a manner comparable (and in ways superior) to our own study. We will add Roobaert et al. (2019) to our discussion, since judging from their Figure 7 $\Delta pCO_2$ and cross-terms are more important than wind near the Horn of Africa, and in Figure 8 winds do not factor largely into FCO'$_2$ for 0-10$^o$N or 10-40$^o$N (although, granted, this is a global result so perhaps the AS signal is not clear):

*to the point that wind contribution to the domain's summer anomaly was 90.8% in magnitude relative to the total. We should keep in mind, however, that these results conflict with the analysis of Roobaert et al. (2019). In their global study of coastal waters, while seasonal $CO_2$ flux variability in the AS is relatively high compared to other regions (their Figure 6), the largest contributions come from $\Delta pCO_2$ and cross-terms (their Figure 7), especially near the Horn of Africa. As a result, further work should be conducted to reduce uncertainty in sea surface $pCO_2$ values to determine whether winds, $\Delta pCO_2$, or cross-terms are significant drivers of air-sea flux. Additionally, when considering the inconsistencies of models in estimating air-sea $CO_2$*

*flux (Sarma et al., 2013),  uncertainties from  incomplete representation...*

As an aside, we would be curious to see where the data come from that allow for estimation of $pCO_2$ and flux coming from the Persian/Arabian Gulf...
* * *
>4.4.2
>Question: What is the real purpose of this section 4.4.2? is the objective to calculate
>an uncertainty on the flux? if so, why only focused on the choice of pco2? why you
>chose TK09 instead of that of Rodenbeck (2014) which is finer in terms of spatial
>resolution and more recent. What about the uncertainty associated with the other
>terms involved in the flux calculation? the wind? its spatial and temporal resolution?
>the k-parametreization? Etc...

The point of Section 4.4.2 is not to calculate the total uncertainty in flux. Rather, it is to highlight that despite the fact that in our analysis winds appear to control the timing and magnitude of $CO_2$ flux, there is non-negligible variability in our estimate due to the uncertainty in $pCO_2$, and so we show that variability by keeping constant the wind product, wind parameterization, etc. At some level, this is an important point to make considering that as discussed above Roobaert et al. (2019) show that $\Delta pCO_2$ is more important than winds in determining flux!

We understand that different wind products and parametrizations would lead to different flux magnitudes, but that is not the point of our study (plus it would be numerically quite onerous to run a complete sensitivity analysis with the full model). Again, TK09 was chosen for comparison with previous studies in the region, and the L15 climatology is both recent and has a finer resolution.
* * *
>Question: This study highlights that the effect of the wind governs the seasonal
>variability of the flux. Why is it important to do, as it is performed here, a finer
>pCO2 decomposition into processes?

True, at face value if the $CO_2$ flux was the sole variable of concern, then the importance of winds should obviate the need to enter further detail into the other components. However, since many other studies (including the Roobaert et al.2019 analysis above) stress $\Delta pCO_2$ and surface $pCO_2$ dynamics in general, it is included here. This is both for the sake of completeness but also so that if future studies contradict ours, then perhaps the weakness in our study (whether it is the wind product or the $pCO_2$ model output) will be more clear.
* * *
>Line 540: "Considering the important role of winds" on the seasonality? Spatiality?
>Not clear

In this sentence the stress is on seasonality:
*Considering the important seasonal role of winds...*
* * *
>Line 542: Fig 12: should be "Tg C month-1" instead of "TgC"

Yes, formally the unit is a flux, and so it will be changed to $TgC month^{-1}$.
* * *
>Line 546: "Despite differing pCO2 seasonality". In this figure only LA15 is referred.
>Not the other products. So you can not refer as something common between all
>pco2 products

Fair enough, the reference to Figure 5 was to show an example of how $pCO_2$ seasonality is different, without haveing to provide a figure showing the $pCO_2$ seasonality of ALL the $pCO_2$ products. In order to be consistent, the reference to Fig. 5 will be removed in the revised MS.
* * *
>Line 560: "1) gridded data-based pCO2 products will under-estimate the upwelling
>zone maxima of pCO2 and CO2 flux during the summer" how can you conclude that
>since you said before that your model produces higher pCO2 values compared to
>other studies. What is the value/study for this conclusion? Not clear

The conclusion that $pCO_2$ products under-estimate summertime $pCO_2$ should be clear from Fig. 5; if necessary, a similar plot can be made for all the other $pCO_2$ products considered in this study in the Supplementary material. The model produces an overall positive bias, true, but the model still has a slight *negative* bias during the *summer*.
* * *
>5
>Line 573: 160 while in line 550 it is between 57-120. Not clear.

The true model estimate is $160\mathrm{TgCyr}^{-1}$. This discrepancy arises because the model, with its high resolution, has a greater total surface area, whereas the more coarse products do not. To this end, model output was interpolated to the coarser grid for a more fair comparison. This needs to be clarified in the methods section (end of Sect. 2.1):

*differences between this software suite and Orr and Epitalon (2015) are minimal. Furthermore, for air-sea $CO_2$ flux intercomparison purposes, all $pCO_2$ values except for TK09 are interpolated to the same $1^o x 1^o$ grid already shared by GLODAP, WOA, and L15. This also means model $CO_2$ flux in the comparison context is reduced due to the model's high spatial resolution.*
* * *
>Line 574: need references.

In the revised MS we will add these references:

*with the range of other findings (Sarma, 2003; Naqvi et al., 2006; Sarma et al. 2013, ).*

---

## Author Comment (AC2)

**Response to Reviewer 2's Comments**

Alain de Verneil      Zouhair Lachkar      Marina Lévy      Shafer Smith

June 16, 2021

We all would like to thank Reviewer 2 for providing their comments in reviewing this manuscript (MS). It is very clear from the detail and thoughtfulness of the comments and questions that a lot of time and effort were spent in the review, which is both fair and will help produce a better paper. Below, we reproduce the review (lines preceded with >), with our responses interspersed within along with updated text from the MS in italics. Changes made to the original text show red and blue for deletions and additions, respectively.

>The authors used a high-resolution regional model to quantify the annual mean
>and the seasonal cycle of air-sea fluxes of CO2 in the Arabian Sea. The model
>results showed that monsoon-driven sea surface temperature variations strongly
>influence the seasonal cycle of air-sea fluxes of CO2 in the Arabian Sea, except
>in upwelling regions. Here the supply of DIC seems to exert a main control on
>CO2 fluxes across the air-sea interface. Overall, the model results imply that the
>Arabian Sea acts as a CO2 source to the atmosphere, since the biological
>drawdown of CO2 in surface waters failed to overcompensate the physical CO2
>supply. Additionally, strong winds increased the CO2 fluxes into the atmosphere
>especially during the upwelling season in summer. The annual mean CO2 flux
>into the atmosphere amounted to 160 TgCyr-1.

>In principle the paper is well structured and nicely written even though it includes
>quite extensive data descriptions, which could be shortened. However, I have
>three significant overarching objections due to which I recommend a major
>revision. The first objection regards the novelty of the presented work, the
>second point of criticisms is the way in which temperature and especially DIC and
>TA are discussed, while the last one refers to the model validation.

>Novelty: The results obtained by the model were expected and not new, except
>the magnitude of the CO2 flux into the atmosphere. This estimate by far exceeds
>estimates derived from field data and other models of up to approximately 90
>TgCyr-1. To me it remained elusive whether the presented estimate of 90
>TgCyr-1 is reliable or a model artefact. The authors used a high-resolution
>model to study the carbon cycle and the resulting air sea fluxes of CO2 in the
>Arabian Sea. This is a new approach that according to the authors, eliminates
>shortcomings of coarse-resolution models and helps to overcome uncertainties
>caused by the low density of field data. In so far, the high CO2 flux seems to be a
>new result showing that CO2 fluxes from the Arabian Sea into the atmosphere
>have been underestimated in the past. However, due to differences between
>model data and field data (DIC and especially TA), it appeared to me that the
>authors even considered the high CO2 flux, at least partly, as an artefact. This
>aspect needs to be clarified since it reduces the novelty of the study and
>additionally raises doubts regarding the advantage of the used high-resolution
>model over previously used coarse-resolution models.

Indeed, the Reviewer is perceptive and has focused on the primary tension in the paper. Regarding the base claim, that the high-resolution model is superior to a more coarse-resolution model, at the very least this is true if one glances at Figures 3, 4, and 7 from the metanalysis for RECCAP by Sarma et al. (2013). In that study, we see how the coarse-resolution models under-estimate flux. By comparison, the estimate in this study, despite its limitations and high flux value, we argue is more in line with observational studies. For example, in Sarma (2003), the statistical fits for DIC and TA from the JGOFS data are quite impressive, and the ultimate $CO_2$ flux estimate is $90TgCyr^{-1}$ for the Arabian Sea (AS) *north of $10^o N$*. Since our domain also includes the AS from the equator to $10^o$N, which by all accounts is also a net outgassing region, arriving at a $CO_2$ flux $>100TgCyr^{-1}$ does not seem quite as unreasonable.

For our part, the greatest uncertainty is the balance between slightly under-estimating $pCO_2$ in the summer, and over-estimating it the rest of the year. First, our under-estimation of $pCO_2$ is not as bad as the climatological products such as L15, and so estimated summer fluxes will be better. Second, for the rest of the year, when we over-estimate $pCO_2$, winds are not as strong, so excess $CO_2$ flux is limited. What the overall balance is, whether the over-estimate of $CO_2$ flux outside of summer is larger than the under-estimate during the summer, remains our greatest uncertainty.

Furthermore, an unappreciated factor to consider is the model's high-resolution, which increases the total area considered. For example, when we restricted our model output to the $1^o$x$1^o$ grid of the World Ocean Atlas, GLODAP, and the L15 climatology, this reduced the flux from 160 to 120 $TgCyr^{-1}$. Reviewer 1 commented on the discrepancy between these two values (and it's commented on below, as well), and so in the revised MS this will also be highlighted.

In summary, all this is to say that, despite our hedges, we are still confident that previous $CO_2$ flux estimates are probably too low, and that the AS from the equator northward probably outgasses $>100TgCyr^{-1}$.

The revised MS, then, will clarify that despite sources of uncertainties, the model result should be considered seriously as an indication that previous $CO_2$ estimates are too low.
* * *
>Discussion: The authors discussed temperature, TA and especially DIC changes
>as processes but these changes are the result of the interplay of different
>physical and in the case of DIC and TA, also biological processes. To my
>understanding, disentangling the role of these processes on the CO2 flux should
>be the main aim of data evaluation and discussion. For instance, I would have
>expected a discussion about the impact of the marine carbon pumps on the CO2
>emissions into the atmosphere. Since changes in DIC, TA and temperature are
>the result of their interplay, a discussion, which is largely restricted to changes of
>parameters, circumvents the discussion on driving forces and that is what, to my
>opinion, matters.

The wish to discuss processes such as primary production, surface heating, and upwelling advection, more than "parameters" or "variables" such as TA and DIC, is understandable and is also mentioned by Reviewer 1, as well. As written in our response to that review, we are amenable to conducting a similar analysis for TA, temperature, etc. as has been done for DIC, so that all aspects of how underlying processes impact the parameters determining $pCO_2$ can be ascertained. The complete analysis will have to be completed as part of the revision, but below please find a figure similar to Fig. 8 but for TA:
* * *
[Figure]

Fig. R2-1. TA budget similar to Fig. 8 for domain (top-left) and the five sub-regions.

>Model validation: Model validation is a crucial aspect especially in the presented
>work as the model includes a variety of processes and parameterization, which to
>my opinion, are problematic. In the following I will present two examples to
>underpin this statement.

>1) The model is based on nitrogen, and a fixed C/N ratio of 106/16 is applied to
>convert nitrogen into carbon and vice versa. In contrast to many other regions,
>denitrification and nitrogen fixation in the Arabian Sea strongly influence the
>nitrogen cycle and cause deviations from classical Redfield ratio. In previous
>studies, some of the co-authors used a similar or even the same model to
>investigate the nitrogen cycle in the Arabian Sea. These aspects should also be
>included into the current work as changes in the Redfield ratio and the
>availability of nitrogen both affect the carbon cycle and the resulting fluxes of
>CO2 into the atmosphere.

Yes, the Arabian Sea is an interesting region because denitrification in low-oxygen environ-
ments and nitrogen fixation (mostly at the surface) impact the nitrogen cycle. The biological
model uses a nitrogen currency, with a fixed C/N ratio, and explicitly includes denitrification, but
unfortunately not nitrogen fixation. Since this remark is included as an example regarding model
validation, we will include a supplementary figure comparing model $NO_3$ (which is impacted by
denitrification) with *in situ* data (see below):

[Figure]

Fig. R2-2. (top-left) $NO_3$ profiles from GLODAP database (black) and ROMS model output (red) interpolated to same lon/lat/depth/month, and (top-right) the same but zoomed into the top 1000m. (bottom) Model output plotted against data, with 1:1 line (red dashes) and colored according to sampling depth.

It's true that processes such as denitrification and nitrogen fixation will impact Redfield ratios in the water column, especially the N:P ratio. Unfortunately, the model does not resolve Phosphate, and so aspects such as a change of limiting nutrient is not possible. C is not likely to be a limiting nutrient, as reflected in the inorganic pools, DIC and DIN ($NO_3+NO_2$), where typical deep values of DIC vs $NO_3$ (the major component of DIN) are 2300:35$\mu$M, giving a C:N ratio closer to 65:1 than the Redfield 6.6:1. If the reviewer has a specific dataset or quantity with which to help better validate the model, we are more than happy to include it as part of the Supplementary material.
* * *
```
>2) Production, export and dissolution of CaCO3 are further issues, which, to my
>understanding, need to be reconsidered especially if one considers the TA
>problem as mentioned before. The production of CaCO3 was linked via a fixed
>ratio to primary production. Primary production rates were compared to satellite
>data but what about the production rates of CaCO3? Furthermore, it was
>assumed that CaCO3 sinks with a velocity of 20 m day-1 and dissolves with a
>rate of 0.0057 day-1 in the water column and 0.003 day-1 in surface sediments.
>Does this approach reflect the distribution of carbonate in shelf sediments? In
>regions where oxygen-depleted mid-waters flushes the shelf, primary production
```

```
>is high and thus also the carbonate production, but low CaCO3 concentrations
>within the underlying sediments indicate a high CaCO3 dissolution. Does the
>model represent such processes? Furthermore, how does the constant carbonate
>dissolution rate agree to observations showing that the entire upper water
>column is oversaturated with respect to calcite and aragonite and how do forams
>fit into the modelling approach? They are assumed to be major CaCO3 producers
>in the Arabian Sea and their shells can sink with a speed of several hundred
>meters per day!
```

We appreciate this focus on $CaCO_3$, especially since resolving issues here can help with the model's bias in total alkalinity. The reviewer notes that primary production rates were compared to satellite data but not $CaCO_3$ production rates. Having looked into the literature, we were able to find $CaCO_3$ production rates that date from the JGOFS years in the database located at https://doi.org/10.1594/PANGAEA.888182. The median production rate for this dataset is 0.03 $mmolCm^{-3}day^{-1}$. Let's extend this rate to the top 50m of the water column, which is close to the measured euphotic zone depth ($\sim$1% surface PAR in the $CaCO_3$ dataset). This results in $1.5mmolCm^{-2}day^{-1}$, or $6.57gCm^{-2}yr^{-1}$. Using the 7% fixed ratio to NPP, this means a median primary production rate of $94gCm^{-2}yr^{-1}$. Most of the sampling locations (being JGOFS) are near the Arabian peninsula and moving slightly into the central AS (see figure below):

[Figure]

Fig. R2-3. Locations of $CaCO_3$ carbon production from the Poulton et al., 2018 dataset.

Judging from the NPP rates in Figure 1, a median primary production rate of $94gCm^{-2}yr^{-1}$ might be hard to justify for the central oligotrophic region, but in the productive zones this would be not difficult to achieve. As a result, the ratio is not inconsistent with the implied primary production that the $CaCO_3$ rate measurements suggest.

Still, it is true that the model is too simple in other ways, and perhaps can be better calibrated

to *in situ* data. The current formulation of the model has constant rates for many processes, and does not take into account ambient conditions such as $CaCO_3$ solubility in acidic waters. If the reviewer has a relevant dataset at hand with carbonate sediment distributions and concentrations, it would be much appreciated for further model refinement and validation should the step be taken to explicitly include these processes.

Also, with regard to Forams and their particular sinking speeds, it may be worthwhile to investigate the magnitude of these fluxes and impact on TA to judge whether including them specifically has merit. We the authors have no particular qualm with forams, but as it is, the model has only one component each for all the complexity of the phytoplankton and zooplankton assemblages in the AS, so each additional component's numerical expense must be compared to its relative value in resolving biogeochemical processes.
* * *
```
>Even though the data density is low over the last decades, it is high during the
>JGOFS field studies and this data can be used to dispel the majority of my doubts
>regarding processes discussed before in the two examples. For instance Millero
>et al. (1998) presented water column profiles showing parameters characterizing
>the carbonate system such as saturation states of calcite and aragonite, TA, DIC,
>pH and AOU. Morrison et al. (1998) provided associated nutrient data including
>data on the distribution of dissolved oxygen in the water column. I strongly
>recommend to include data obtained from the entire water column into the
>model validation and show profiles of field and model data in one plot. See e.g.
>Figure A1 C in Segschneider et al. (2018). Such plots draw a clear picture and
>allow all readers to asses the reliability of model results.

>Segschneider, J., Schneider, B., Khon, V., 2018. Climate and marine
>biogeochemistry during the Holocene from transient model simulations.
>Biogeosciences, 15, 3243-3266.
```

Thank you for providing a reference from which we could produce similar Supplementary Figures to aid in the better validation of our model. Please find in the three figures below comparisons between the GLODAP dataset (which includes the available JGOFS data for DIC, TA, and $O_2$) and our model output, which will be added as Supplementary Figures to our revised MS.

[Figure]

Fig. R2-4. (top-left) DIC profiles from GLODAP database (black) and ROMS model output (red) interpolated to same lon/lat/depth/month, and (top-right) the same normalized to S=35.. (bottom-left) Model output plotted against data, with 1:1 line (red dashes) and colored according to sampling depth, and (bottom-right) same but for normalized data.

[Figure]

Fig. R2-5. (top-left) TA profiles from GLODAP database (black) and ROMS model output (red) interpolated to same lon/lat/depth/month, and (top-right) the same normalized to S=35.. (bottom-left) Model output plotted against data, with 1:1 line (red dashes) and colored according to sampling depth, and (bottom-right) same but for normalized data.

[Figure]

Fig. R2-6. (top-left) $O_2$ profiles from GLODAP database (black) and ROMS model output (red) interpolated to same lon/lat/depth/month, and (top-right) the same but zoomed into the top 1000m. (bottom) Model output plotted against data, with 1:1 line (red dashes) and colored according to sampling depth.

Hopefully showing the model output at depth in comparison with the available *in situ* data helps allay some of the reviewers' fears regarding some of the model's weaknesses in TA and DIC.
* * *
```
>More specific comments:
>Line 9-10: the authors wrote 'In the seasonal pCO2 cycle, temperature plays the
>major role in determining surface pCO2, except where DIC delivery is important
>in summer upwelling areas.' The first part of the sentence is correct in so far as
>temperature influences the solubility of CO2 in water, but the sentence is
>confusing since temperature is no process. It is a physical quantity and results
>from the interplay of different physical processes. These processes control the
>surface pCO2 via their impact on temperature and I would have expected to learn
>something about processes controlling the pCO2.
```

Yes, as already mentioned Reviewer 1 also raised the concern that the word "process" is used inappropriately, and towards that end, again we propose to extend our DIC analysis to TA, SST

and sea surface salinity (SSS) to explicitly account for how all the "processes" translate into changes of variables impacting $pCO_2$.
* * *
>Line 11: the authors wrote: 'We find that primary productivity during both
>summer and winter monsoon blooms, but also generally, is insufficient to off-set
>the physical delivery of DIC to the surface, resulting in limited biological control
>of CO2 release.' To my understanding, it is the export production rather than
>primary production, which should at least partially offsets the physical delivery of
>DIC to the surface.

Yes, in the global sense of the entire water column, export production is what removes fixed C from the DIC pool and moves it to deeper depths. Since our control volume for the decomposition of $pCO_2$ variables and DIC processes are entirely in the surface layer we focused on the first step, that of fixation via primary production, and not the export component from the euphotic zone. Of course, some of this production gets remineralized before it can get exported, which is included in the DIC budget presented in the MS. In the revised MS, we will to include particulate C export in the discussion to provide an estimate of how much primary productivity is ultimately exported in the model.
* * *
>Line 17: Please clarify the term 'Reynolds decomposition'. In line 223 the term
>was also used and Doney et al. (2009) was cited but the name 'Reynolds' was not
>mentioned in this paper.

Thank you for pointing this out, we should more explicitly explain. Reviewer 1 also had a question about the "five-year average," which is necessary because in Reynolds averaging fluctuations are around a specific time-average, and and as a result in order for the equations to be exact so we used the entire model timeseries. We will add a quick addition to the revised MS outlining what we mean:

*...here we use a Reynolds decomposition (Doney et al., 2009). Briefly, a Reynolds decomposition takes a timeseries and divides it into a temporal mean and fluctuating component. When applied correctly, multiple terms can be produced in isolation showing their fluctuating contribution to the total.*
* * *
>Line 19 - 23: the author wrote: 'Since summer monsoon winds are critical in
>determining flux both directly and indirectly through temperature, DIC, TA,
>mixing, and primary production effects on pCO2, studies looking to predict CO2
>emissions in the AS with ongoing climate change will need to correctly resolve
>their timing, strength, and upwelling dynamics.' Please clarify how wind relates
>to parameters such temperature, DIC, TA and processes such as mixing and
>primary production.

Yes, we did not explicitly state how winds play a role in the multiple variables impacting not only air-sea $CO_2$ flux, but also SST, DIC, etc. While in lines 352-355 we suggest that wind forcing both upwells subsurface DIC and provides mixing, also enhancing DIC. However, it must be stated that these same processes (upwelling and enhanced mixing) will entrain colder waters from below and result in reduced SST and increased SSS.
* * *
>Line 197- 198: What about upwelling and eddies? To my understanding the
>advantage of the high-resolution model was to resolve such processes.

Yes, it is true that one of the purposes of having a high resolution model is to be able to include phenomena such as coastal upwelling and mesoscale eddies. The "vertical circ." term in our budget encompasses the coastal upwelling, and its impact should be visible for the Oman sub-region in Figure 8.

As for eddies, a previous study cited in our MS, Resplandy et al. (2011), found that eddies are an important export mechanism of nutrients into the central, oligotrophic region of the AS. In order to treat eddy flux explicitly, we would have to choose another analysis method to identify mesoscale structures and follow them in time, possibly across multiple years in our 5-year time-series. However, since the horizontal advection term of the DIC budget (see Fig. 8) was never the largest magnitude flux, focusing on mesoscale export did not figure into the present analysis, and eventually in lines 504-507 we hypothesize that possibly the timescale for DIC uptake and removal is short.
* * *
>Line 200 – 205: Since NPP stands for 'net primary production' (see e.g. line 231),
>I would suggest to replace 'PPNew+Reg' by NPP and to rename the term NPP
>Remin. If I understood it correctly, the term 'NPP-Remin' represents the soft
>tissue and the alkalinity pump.

Thank you for the note, it will be clearer to the reader if a consistent notation of NPP is used, and this will be changed in the revised MS. The NPP-Remin (Biology) grouping in Eqn. 6 does include the soft tissue pump, as well as the alkalinity ($CaCO_3$ component) pump.
* * *
>Line 206: Please name the two detrital pools. I guess that Det-remin represents
>bacterial respiration, is this correct?

The two detrital pools are designated "large" and "small," primarily to accommodate for faster sinking of larger particles, as well as for aggregation of smaller particles into larger ones. Reviewer 1 has requested that we include the model parameters used, and so the differences between these two detrital pools will be clearer in the revised MS. The reviewer is correct, Det-remin represents bacterial respiration of biomass, which will be clarified in the revised MS:

and $Det_{remin}$ is remineralization of both detrital pools, *i.e. bacterial respiration.*
* * *
>Line 213 – 215: Perhaps I misunderstood this part but deep DIC feeds the CO2
>emission into the atmosphere. Its annual cycle largely controls the air sea fluxes
>and without describing the deep DIC cycle there is no way to see whether the
>surface DIC dynamic operates correctly.

The reviewer did not misunderstand, ultimately some part of deep DIC will find its way to the surface and feed $CO_2$ emission. However, determining how deep, and from where, is a worthwhile follow-on study in itself that would probably require using Lagrangian methods. The reviewer is right to be curious about the ultimate sources of AS $CO_2$ emission, but at present this is beyond the scope of this study.

Hopefully, with the addition of deep *in situ* data as part of our validation, the reviewer can make an educated judgment regarding whether the surface DIC dynamics operates correctly.
* * *
>Line 220: Why the authors used this old 'K0' parameterization ?

True, at this point the Wanninkhof 1992 parameterization is dated, and in other studies the 2014 formulation from the same author is presently being used. Similar to our response to Reviewer 1's questioning of our use of older climatological datasets, we chose the old K0 representation so that our model results could be compared to previous studies in the region. In particular, Sarma et al. (2003) included the 1992 K0 formulation. We will add an explanatory sentence in the revised MS:

*...positive outward from the ocean.* *The choice of Wanninkhof (1992) for the solubility parameterization was for direct comparison with previous studies, despite the fact more recent formulations are available, such as Wanninkhof (2014).* *The objective...*
* * *
>Line 249 - 250: Since Biogeoscience has a wide readership and many scientists
>are interested in the topic presented by the authors, I suggest to present simple
>x-y plots in addition or instead of Taylor diagrams. Furthermore, I suggest to
>comment the results and say that the model results and field data reveal a weak
>correlation in spring and summer and do not correlate in winter and fall!

The reviewer is correct, Biogeosciences has a wide readership, and so accommodations must be made sometimes to make results digestible for scientists from other specialties. Due to the comments from Reviewer 1, we are already adding more Taylor diagrams, but including another subpanel showing data on an x-y plot would not be difficult; we will do this for the revised MS. Also, the reviewer makes a good point, we should comment on how the Taylor diagram shows weak correlations for some seasons but not others. In the revised MS will include this in the Results, including especially the number of observations, etc.
* * *
>Line 255: This statement regarding the model performance refers only to surface
>data including primary production which poorly constrains the carbon cycle as it
>includes the regenerated production. As mentioned before, without getting an
>impression about the distribution of DIC, TA, and nitrate and oxygen in the water
>column, it is to my understanding impossible to judge the model performance.

Yes, thank you, we hope that by adding the plots shown in our responses above that we have addressed this concern.
* * *
>Line 280: What is the role of sediments in this shallow water and acidic
>environments?

We presume that the author is referring to the west coast of India, since most of the other coastlines in the region have limited continental shelves. It is true, a lot of remineralization must occur in the sediments, and is a source of DIC (considering how shallow, a large part of the "vertical circ" is probably the mixing of sediment-derived DIC to the surface. We will both add an estimate of total annual DIC production in the sediments within the "India" subregion of our model analysis and include it in the discussion of the revised MS.
* * *
>Line 328 - 330: Please rephrase this sentence and include all processes of
>relevance.

Yes, very well, in the revised MS this sentence will read:

*Whereas SST and its effect on pCO2 is controlled by the physical processes of surface forcing, mixing and advection, DIC reflects both physical and biological processes because it is also impacted by photosynthesis, respiration, remineralization, and shell calcification.*

* SST's  effect on pCO$_2$  reflects physical processes  like surface heating and cooling, mixing and advection. DIC, by contrast, reflects both physical and biological processes because in addition it is also impacted by photosynthesis, CaCO$_3$ shell formation and dissolution, zooplankton respiration, detritus remineralization (bacterial respiration), and  air-sea exchange.*
* * *
>Line 332: and that is why the deep DIC cycle has to be included.

Duly noted, we understand why the reviewer would like to see a budget that includes these deeper processes. As mentioned previously, analysis of the entire water column may serve as a follow-up paper of the entire Carbon cycle of the AS, but begins to draw away from the focus of air-sea CO$_2$ flux, which is the primary reason for this study.
* * *
>Line 338: This estimate agrees quite well to a result obtained from field data,
>which implies that organic carbon and CaCO3 export removed 30 - 70% of the
>DIC introduced into the surface layer via physical processes. See Rixen, T.,
>Guptha, M.V.S., Ittekkot, V., 2005. Deep ocean fluxes and their link to surface
>ocean processes and the biological pump.Progress In Oceanography, 65, 240 -
>259.

We thank you for mentioning this reference, we will include it in our revised MS, and hint towards follow-up work including the entire water column.
* * *
>Line 430: Yes, this is to my understanding the advantage of using a high
>repulsion model but apart from stating it, it should also be demonstrated.

We undestand that we should explicitly demonstrate how increased resolution results in a better representation of upwelling. The correct comparison would be with a model of $\sim$1$^o$x1$^o$. Unfortunately, at these scales the mesoscale is completely absent, and it becomes necessary to parameterize their impact on tracer mixing. Since ROMS as a regional model is not designed to do this, it would require either coding up the paramterization, or using another more traditional GCM model. We know this is not satisfying, but all we can suggest is to compare with the model outputs presented in Fig. 4 of Sarma et al. (2013), which we will again emphasize in our revised MS.
* * *
>Line 435: please consider also
>Körtzinger, A., Duinker, J.C., Mintrop, L., 1997. Strong CO2 emissions from the
>Arabian Sea during south-west monsoon. Geophysical Research Letters, 24,
>1763-1766.
>Goyet, C., Millero, F.J., O'Sullivan, D.W., Eischeid, G., McCue, S.J., Bellerby, R.G.J.,
>1998. Temporal variations of pCO2 in surface seawater of the Arabian Sea in
>1995. Deep Sea Research I, 45, 609-623.
>Millero, F.J., Degler, E.A., O'Sullivan, D.W., Goyet, C., Eischeid, G., 1998. The
>carbon dioxide system in the Arabian Sea. Deep Sea Research Part II: Topical
>Studies in Oceanography, 45, 2225-2252.

Thank you for these references to demonstrate this point; while we use both Goyet et al. 1998 and Millero et al. 1998 elsewhere, we are also happy to include Kötzinger et al. 1997 as well.
* * *
```
>Line 550: Is 120 correct? What about the 162.6 TgCyr-1 as mentioned in line
>393? Papers of relevance, which should have also been cited in this context, are:
>Somasundar, K., Rajendran, A., Dileep Kumar, M., Sen Gupta, R., 1990. Carbon
>and nitrogen budgets of the Arabian Sea. Marine Chemistry, 30, 363-377.
>George, M.D., Kumar, M.D., Naqvi, S.W.A., Banerjee, S., Narvekar, P.V., de Sousa,
>S.N., Jayakumar, A., 1994. A study of the carbon dioxide system in the northern
>Indian Ocean during the premonsoon. Marine Chemistry, 47, 243-254.
```

Reviewer 1 also noted the 120 vs 160 $TgCyr^{-1}$. The reason we report 120 $TgCyr^{-1}$ in this part is because in order to compare between $pCO_2$ sources, we had to use the same grid/surface area. Therefore, when we interpolate the model from the natural grid to the $1^o$x$1^o$ WOA/GLODAP/Landschützer grid, it results in a net loss of almost $40TgCyr^{-1}$! Again, thank you for the relevant citations, we will include them in the revised MS.

---

## Author Response (AR2)

**Response to Reviewer's Comments**

Alain de Verneil        Zouhair Lachkar        Marina Lévy        Shafer Smith

November 22, 2021

We are pleased that the changes made to the manuscript were found to be sufficient for Reviewer 1 and mostly satisfactory for Reviewer 2. We thank both reviewers and the editor for the time and work put into improving this manuscript. As in the previous iteration, below we reproduce the reviewer's suggestions (lines preceded with >), with our responses in plain text and updated text in italics. Changes will be in red and blue for deletions and additions, respectively.

>Comments:

>Abstract
>Line 16: I would suggest to avoid the term 'Reynolds decomposition' in the
>abstract as this term is not familiar to many readers and requires an
>explanation which the authors provide in line 199 only.

True, and considering that the emphasis within this paper is not specifically this technique, we agree to remove this from the abstract:

> … *versus **6%** for pCO$_2$* . *In comparison with other...*
* * *
>Introduction
>Line 32: was Lachkar et al., 2016 this first and only one who discovered that
>the world's thickest oxygen minimum zone was located in the Arabian Sea? See
>e.g. Acharya and Panigrahi, 2016, in Deep Sea Research Part I, Vol. 115 Pages
>240-252.

We did not mean to imply through the reference that this was the first time that it was stated that the Arabian Sea's OMZ is the world's thickest. We will add these references:

> *…in addition to unique features such as the world's thickest oxygen minimum zone (OMZ) (Morrison et al., 1999; Acharya and Panigrahi, 2016; Lachkar et al.,2016) and corresponding Carbon Maximum Zone (CMZ) (Paulmier et al., 2011).*
* * *
>Line 51 : please clarify the meaning of 'similar story'

Understood, by 'similar story' we were trying to make the point that surface pCO$_2$ data in the AS is also dated, with most data (>98%) before 2000. We will add this clarification in the revision:

> *…more recent than 1998, with a similar  >98% of data predating 2000 for pCO$_2$.*
* * *
>Methods
>Line 117: is this consistent with the latest IPCC report?

The reviewer is right, in CMIP6 (the most recent iteration of the models used in the IPCC) the historical models now run until 2014, as opposed to 2005 in CMIP5. We will correct the MS to reflect this detail:

*The year 2005 is chosen for the model's $xCO_2$ concentration because it is the end of the historical period for the Intergovernmental Panel of Climate Change (IPCC) models used in its 5th report published in 2014.*
* * *
>Line 118/119: please rephrase this sentence which is really difficult to
>understand

We will rephrase the sentence as follows:
*As a result, we assume there is  a baseline seasonal cycle of pCO2 and air-sea CO2 flux  which has held  stable over the past decades.*
To read:
*As a result, we assume there is a baseline seasonal cycle of $pCO_2$ and air-sea $CO_2$ flux which has held stable over the past decades.*
* * *
>Line 153: Due to high sinking speeds of foram-shells and their significant
>contribution to the carbonate fluxes in AS sediment traps (e.g. 1) , I would
>suggest to include at least a short note saying that this low sinking is a
>simplification. Since the authors, furthermore, state that the TA bias cause
>the overall positive pCO2 bias (see line 340) and enhanced carbonate export
>via foram-shells might have solved these issues, I would also recommend to
>include the aspect of too low sinking speeds into the discussion of the TA
>bias in line 340.

>[1] Curry, W. B., Ostermann, D. R., Guptha, M. V. S., and Ittekkot, V. (1992)
>Foraminiferal production and monsoonal upwelling in the Arabian Sea: evidence
>from sediment traps, In Upwelling Systems: Evolution Since the Early Miocene
>(Summerhayes, C. P., Prell, W. L., and Emeis, K. C., Eds.), pp 93-106,
>Geological Society Special Publication.

We will first add a note to line 153 as suggested:

*In addition to usual physical transport and mixing, $CaCO_3$ is allowed to vertically sink at 20m $day^{-1}$. The chosen sinking rate is a simplification in that it does not include the faster rates observed for foraminifera shells (Curry et al., 1992), which as a biological group are not resolved by the biological model due to numerical constraints. Organic carbon...*

We will also add to line 340:

*As a result, while the DIC model bias lowers $pCO_2$, the stronger bias in TA is the most likely cause for the model's overall positive $pCO_2$ bias, which may in part be due to the unresolved fast sinking rates of foraminifera in the model.*
* * *
>Line 167 Please delete 'in this study' and change 'chnaged' into 'changed'.

Thanks for spotting the typographic errors, the changes have been made in the MS.
* * *
>Line 150 delete 'i.e. remineralization'

This deletion has been made in the MS to 'i.e. remineralization', although this occurs (in the marked-up version that appears to be used so far) at line 259:

*...is remineralization in both detrital pools*
* * *
>Line 188: Please describe this in more detail. For instance, Figures R2-3 and
>2-4 show that model data reflects general trends but there are also
>deviations between GLODAP and model data. These deviations are of great
>relevance as they are most pronounced in water-depths of approximately <500m.
>Processes within this depth-range strongly influence the pCO2 in the surface
>waters.

We assume the reviewer is referring to line 288 (in the latest mark-up version), which references depth profiles of $NO_3$, $O_2$, DIC, and TA. We will add the following:

*Depth profiles of nitrate, oxygen, DIC, and TA are similarly conserved (Fig.S3-S6). Nitrate, DIC, and TA all show their usual nutrient-like profiles, while oxygen is its minimum within the OMZ. The deviations seen between in situ data and model output are greatest at depths less than 500m. Deviations in near-surface $NO_3$ (Fig. S3) can be large for intermediate values (5-20μM) but overall do not show a systematic bias. DIC (Fig. S5) also has large deviations (∼50μM) in the top 500m and with a slight positive bias. It is in TA (Fig. S6) that deviations, while similarly ∼50μM-eq, show a consistent near-surface underestimation. Thesurface currents in the model also demonstrate...*
* * *
>Results
>In general: please check the grammar and the readability. In many cases
>additional information has been squeezed into sentences which reduces the
>readability of the manuscript and especially the results section.

Yes, we understand that in trying to add all the quantitative values reviewer 1 requested that the readability of the MS has suffered. Please find changes in the updated MS and mark-up version.
* * *
>Line 294 please check grammar

Thank you for the pointer, we checked the following sentence for issues on grammarly.com, and while there do not seem to be specific grammar issues, the sentence is certainly not concise. We propose the following:

*Sampling dates for $pCO_2$  (Fig. 2b)  show that ~~the majority of data (~70%) come,and that mostcoming from the years 1995 and 1997 alone.~~*
* * *
>Line 404 -418. Please rephrase this paragraph. It is difficult to follow and
>to understand from where the numbers are coming from.

Please find our preposed rephrasing below:

*All calculations have their peak $CO_2$ flux sometime in the summer, confirming the role of winds in $CO_2$ flux timing.  This study's model consistently produced on of the higher estimates with*

 120 TgCyr$^{-1}$  (reduced from 162.6 due to re-gridding ) and 57 TgCyr$^{-1}$  north of 10°N. The only estimate higher than the model is GLODAP data in the region north of 10°N with 65 TgCyr$^{-1}$ possibly driven by summer monsoon sampling bias. The high model estimate is perhaps unsurprising, considering the pCO$_2$ bias. The  range in estimates of total  CO$_2$ flux is 57-120 TgCyr$^{-1}$  , resulting in a ratio of 2.1x variability. In the reduced domain of the AS north of 10°N, estimates range from 12.3  to 65.6  , resulting in a ratio of 5.3x variability. The 5.3x ratio is quite high, and is in part driven by the low estimates from the Sarma (2003) model, which are 12.3 and 17.6 using tracer data from WOA and ROMS, respectively. Indeed, the Sarma (2003) model estimates have negative CO$_2$  flux for some months, which is not observed in the original publication,  and the fluxes are quite smaller than the 70 TgCyr$^{-1}$ reported  from Wanninkhof (1992).  in the original publication. If the two lower estimates are removed, the range in air-sea CO$_2$ flux in the domain north of 10°N is 41-65 TgCyr$^{-1}$, providing a ratio of 1.6  similar to 2.1 for the whole domain. Even considering the model's pCO$_2$  bias, as mentioned the GLODAP estimate supersedes it in the region north of 10°N, as does the original Sarma (2003)  estimate with 70 TgCyr$^{-1}$ . Thus, while  we may think the model over-estimates flux, it is still within the range of previous studies in the AS.
* * *
>Discussion
>Line 559: heading is missing

In the mark-up version we are referring to, it is not clear if there is a heading missing. However, it is possible that in the mark-up process of "latexdiff" the appearance of headings may not reflect the final version. For example, on line 705 the section number is 4.3.1, whereas in the updated MS it is 4.3.3 (line 594), as it should be.
* * *
>Line 655 'in' ?

This sentence will be rewritten to be more clear:
Indeed, in a scenario where the  cross-term contribution is at its maximum amplitude, the  Omani upwelling region during  summer (Fig. 8b),  the cross-term is not strong enough to sway the direction of the flux anomaly.
* * *
>Line 684: please clarify this sentence

Yes, the way the sentence is written, it reads as if vertical circulation and biological processes are not important. Rather, we meant to simply say that they are not as dominant as in the DIC cycle. We will re-write it as follows:
A preliminary TA budget of the model (Fig. S12) shows that  while vertical circulation and biological processes strongly dominate the seasonal cycle of near-surface DIC, TA  has multiple forces influencing its time evolution. However,  the magnitude of the...
* * *
>Conclusion
>Line 822 Please clarify of what is a 'pH relevant biological threshold'.

We will change the sentence to include an example of a biological threshold related to pH:
...*whereas other important indicators such as pH and aragonite saturation, $\Omega_a$, which at important thresholds have deleterious impacts for various biological taxa (Doney et al., 2009; Bednarsek et al., 2019; Bednarsek et al., 2021) and its relevant biological thresholds will be less so.*